# PAC-Bayes Analysis for Recalibration in Classification

**Masahiro Fujisawa** [* 1 2]  **Futoshi Futami** [* 1 2]

## Abstract

Nonparametric estimation using uniform-width binning is a standard approach for evaluating the calibration performance of machine learning models. However, existing theoretical analyses of the bias induced by binning are limited to binary classification, creating a significant gap with practical applications such as multiclass classification. Additionally, many parametric recalibration algorithms lack theoretical guarantees for their generalization performance. To address these issues, we conduct a generalization analysis of calibration error using the *probably approximately correct* Bayes framework. This approach enables us to derive the first *optimizable* upper bound for generalization error in the calibration context. On the basis of our theory, we propose a generalization-aware recalibration algorithm. Numerical experiments show that our algorithm enhances the performance of Gaussian process-based recalibration across various benchmark datasets and models.

## 1. Introduction

Increasing the reliability of machine learning models is crucial in risk-sensitive applications such as autonomous driving (Chen et al., 2015). Recently, the concept of *calibration* has become a significant measure of reliability, especially in classification tasks. In this context, the calibration performance is evaluated by how well predictive probabilities provided by our model align with the actual frequency of true labels. A close correspondence between them indicates that the model is well-calibrated (Dawid, 1982; Widmann et al., 2021). To evaluate the calibration performance, a *calibration error* (Gupta and Ramdas, 2021; Roelofs et al., 2022) such as the top-label calibration error (TCE) (Kumar et al., 2019; Gruber and Buettner, 2022) is often used. This

evaluates the disparity between the predicted probability of a model and the conditional probability of the label frequency given by the model prediction. Analytically computing the TCE is, however, challenging because the conditional probabilities of the labels are intractable. Among various methods proposed to address this issue, constructing its estimator called the *expected calibration error* (ECE) using *uniform-width binning* (UWB) (Zadrozny and Elkan, 2001; Naeini et al., 2015) is one of the most widely adopted. This study centers on this approach.

If the ECE is small, we consider the model well-calibrated and its predictions highly reliable. Unfortunately, as has already been made evident in recent studies, models such as neural networks are not necessarily well-calibrated (Guo et al., 2017). If a model shows poor calibration performance, it is common to apply a post-processing technique known as *recalibration* (Guo et al., 2017; Zadrozny and Elkan, 2001). This technique adjusts predicted probabilities using a recalibration function—a parametric function trained separately from the original model—on a dataset independent of both training and test data. Recalibration aims to train a separate function that, when composed with the original predictor, yields a low ECE. Numerous methodologies have been proposed recently, including temperature scaling (Guo et al., 2017) and recalibration based on variational inference (VI) with a Gaussian process (GP) (Wenger et al., 2020).

Given that the ECE is an estimator of the TCE, it is important for reliable uncertainty evaluation to understand the extent of bias introduced between them *before and after* recalibration. Nevertheless, the theoretical understanding of this remains limited. In the context of bias analysis *before* recalibration, many studies have focused exclusively on *binary classification* and have shown that the number of bins used in binning significantly affects both the estimated ECE and the bias (Gupta and Ramdas, 2021; Sun et al., 2023; Futami and Fujisawa, 2024). Such insights remain unclear in the *multiclass classification* context. Moreover, regularization techniques have been proposed to reduce the ECE on training data (Kumar et al., 2018; Popordanoska et al., 2022; Wang et al., 2021a). However, a low ECE on training data does not guarantee a similar low TCE. Verifying this requires a theoretical analysis of the gap between these, which, to our knowledge, has not yet been explored.

[*]Equal contribution [1]The University of Osaka, Osaka, Japan [2]RIKEN Center for Advanced Intelligence Project, Tokyo, Japan. Correspondence to: Masahiro Fujisawa <fujisawa@ist.osaka-u.ac.jp>, Futoshi Futami <futami.futoshi.es@osaka-u.ac.jp>.

*Proceedings of the $42^{nd}$ International Conference on Machine Learning*, Vancouver, Canada. PMLR 267, 2025. Copyright 2025 by the author(s).

This situation remains the same even *after* recalibration. Moreover, in many recalibration methods, postprocessing is performed to achieve a low ECE only on the recalibration data. However, in practice, it is desirable to perform recalibration such that the *ECE evaluated on test data is reduced*—that is, to enhance the model's generalization performance from the ECE perspective through recalibration. Although existing studies have confirmed this point only through numerical experiments (Platt, 1999; Zadrozny and Elkan, 2002; Kull et al., 2017; Naeini et al., 2015; Guo et al., 2017; Wenger et al., 2020), recalibration methods with theoretical guarantees for improving generalization performance in terms of ECE have not been explored thus far.

Given this background, we conduct a comprehensive analysis of the bias of the ECE in *multiclass classification*. To achieve this, we develop a statistical learning theory focusing on the ECE before and after recalibration by applying the *probably approximately correct* (PAC) Bayes theory (McAllester, 2003; Alquier et al., 2016), a general method for deriving generalization bounds across various models. Since PAC-Bayes theory allows for the algorithm-dependent analysis of generalization performance, the obtained upper bounds are often optimizable and useful for deriving new generalization-aware algorithms. In applying PAC-Bayes analysis to the ECE, we face the following two challenges: (i) Owing to the nonparametric estimation of conditional probabilities, the ECE computed on the test dataset is *not a sum of independently and identically distributed (i.i.d.) random variables* and thus cannot be applied using the existing PAC-Bayes bounds derived under the i.i.d. assumption, and (ii) some bins may *not contain samples* because UWB divides the probability interval $[0, 1]$ into equal widths, making it difficult to apply the concentration inequality used in the PAC-Bayes bound derivation.

Our main contribution is a novel analysis of ECE in *multiclass classification*, which addresses the abovementioned limitations. We begin by presenting a decomposition of the bias into *binning bias* and *finite-sample estimation bias*. We then derive a novel concentration inequality that enables theoretical analysis in the binning-based ECE setting with UWB. This framework reveals that the bias in ECE estimation converges at a notably slow rate. Furthermore, by deriving the PAC-Bayes bound, we have successfully formulated a new generalization-aware recalibration algorithm, which is expected to improve the generalization performance of calibration and reduce the estimation bias in TCE. Numerical experiments confirm a correlation between the Kullback–Leibler (KL) regularization terms in our bounds and the generalization performance, show that our method can improve GP-based recalibration, and reveal the instability of ECE-based calibration performance evaluation due to the slow convergence rate. Finally, we discuss related work in light of our theoretical findings in Section 5, providing a

view of how our contributions relate to existing literature.

## 2. Preliminaries

In this section, we summarize the basic notations, problem setup (Section 2.1), calibration metric (Section 2.2), and postprocessing by recalibration (Section 2.3).

### 2.1. Notations and Problem Setting

For a random variable denoted in capital letters, we express its realization with corresponding lowercase letters. Let $P(X)$ denote a probability distribution of $X$, and let $P(Y|X)$ represent the conditional probability distribution of $Y$ given $X$. We express the expectation of a random variable $X$ as $\mathbb{E}_X$. Let $\mathrm{KL}(P\|Q)$ be the KL divergence of $P$ from $Q$, where $Q$ is a probability distribution and $P$ is absolutely continuous with respect to $Q$.

Let $\mathcal{Z} = \mathcal{X} \times \mathcal{Y}$ be the domain of data, where $\mathcal{X}$ and $\mathcal{Y}$ are the input and label spaces, respectively. Let the label space be $\mathcal{Y} := \{0, \ldots, K-1\}$, where $K \in \mathbb{N}$ is the number of classes. For the label $Y \in \mathcal{Y}$, we define the one-hot encoding of the label $Y$ as $\mathbf{e}_Y \in \mathbb{R}^K$. Suppose $\mathcal{D}$ represents an *unknown* data distribution, and let $S_{\mathrm{tr}} := \{(X_m, Y_m)\}_{m=1}^{n_{\mathrm{tr}}}$ denote the training dataset consisting of $n_{\mathrm{tr}}$ samples drawn i.i.d. from $\mathcal{D}$. We also define the test dataset comprising $n_{\mathrm{te}}$ samples drawn i.i.d. from $\mathcal{D}$. Let $f_w : \mathcal{X} \to \Delta^K$ be a probabilistic classifier, where $\Delta^K$ represents the $K-1$-dimensional simplex. This classifier is parameterized by $w \in \mathcal{W} \subset \mathbb{R}^d$. For example, such a simplex is obtained by the final softmax layer in neural networks. Under these settings, $f_w$ predicts the label from $C := \mathrm{argmax}_k f_w(X)_k$, where $f_w(X)_k$ is the $k$-th dimension of $f_w(x)$, which represents the model's confidence of the label $k \in K$.

We evaluate the performance characteristics of the trained predictor $f_w$, such as its accuracy using the classification loss function $l_{\mathrm{acc}} : \mathcal{Y} \times \Delta^K \to \mathbb{R}$, where $l_{\mathrm{acc}}(y, f_w(x))$ denotes the loss incurred by the prediction $f_w(x)$ for the target $y$. We define the training loss as $\hat{L}(w, S_{\mathrm{tr}}) := \frac{1}{n_{\mathrm{tr}}} \sum_{(X,Y) \in S_{\mathrm{tr}}} l_{\mathrm{acc}}(Y, f_w(X))$ and the expected loss as $L(w, \mathcal{D}) := \mathbb{E}_{(X,Y)} l_{\mathrm{acc}}(Y, f_w(X))$. Under this setting, one of the major purposes of the learning algorithms is to achieve the small generalization gap for loss function $l_{\mathrm{acc}}$, which is defined as $|L(w, \mathcal{D}) - \hat{L}(w, S_{\mathrm{tr}})|$.

### 2.2. Calibration Metrics for Multiclass Classification

In the context of calibration, it is expected that not only $f_w$ has high accuracy but also its predictive probability aligns well in the actual label frequency $P(Y = k|f_w(x))$ for $k = 0, \ldots, K-1$. Hereinafter, given $(w, x)$, we treat $P(Y|f_w(x))$ as the $K$-dimensional vector by regarding its $k$-th element as $P(Y = k|f_w(x))$. Then, a model is *well-*

*calibrated* (Widmann et al., 2021; Gupta et al., 2020) when $P(Y|f_w(X)) = f_w(X)$ holds. However, it is difficult to confirm this in practice. Instead, a frequently used measure is the **top-label calibration error** (TCE) (Kumar et al., 2019; Gruber and Buettner, 2022), which uses the highest prediction probability in $f_w$:

$$\text{TCE}(f_w) := \mathbb{E}|P(Y = C|f_w(X)_C) - f_w(X)_C|.$$

Unfortunately, evaluating the TCE is still infeasible because $P(Y = C|f_w(X)_C)$ is intractable. To resolve this issue, we evaluate the calibration performance by constructing the estimator of the TCE. The widely used estimator is the *expected calibration error* (ECE), where *binning* is used to estimate $P(Y = C|f_w(X)_C)$ (Guo et al., 2017; Zadrozny and Elkan, 2001; 2002). The ECE is often computed using UWB, which partitions the predictive probability range $[0, 1]$ into $B$ equal-width intervals $\mathcal{I} = \{I_i\}_{i=1}^{B}$ (called *bins*) and averaging within each bin using an evaluation dataset $S_e := \{z_m\}_{m=1}^{n_e} \in \mathcal{Z}^{n_e}$. For simplicity, we refer to UWB simply as *binning*. Accordingly, we define the ECE based on binning as follows:

$$\text{ECE}(f_w, S_e) := \sum_{i=1}^{B} p_i|\bar{f}_{i,S_e} - \bar{p}_{i,S_e}|, \qquad (1)$$

where $|I_i| := \sum_{m=1}^{n_e} \mathbb{1}_{f_w(x_m)_C \in I_i}$, $p_i := \frac{|I_i|}{n_e}$, $\bar{f}_{i,S_e} := \frac{1}{|I_i|} \sum_{m=1}^{n_e} \mathbb{1}_{f_w(x_m)_C \in I_i} f_w(x_m)_C$, and $\bar{p}_{i,S_e} := \frac{1}{|I_i|} \sum_{m=1}^{n_e} \mathbb{1}_{f_w(x_m)_C \in I_i} \mathbb{1}_{y_m = C}$. Here, bins are set by dividing the interval $[0, 1]$ into $B$ bins of the same width: $I_1 = (0, 1/B], I_2 = (1/B, 2/B], \dots, I_B = ((B-1)/B, B]$.

Since the ECE is an estimator of TCE, it is important to understand the bias defined as

$$\text{Bias}(f_w, S_e, \text{TCE}) := |\text{TCE}(f_w) - \text{ECE}(f_w, S_e)|,$$

and we refer to this as the **total bias**. As discussed in Section 1, one of our goals is to evaluate this bias theoretically in the multiclass classification setting.

### 2.3. Postprocessing by Recalibration

The trained predictor $f_w(x)$ can be poorly calibrated (Guo et al., 2017; Kumar et al., 2019). One common approach to address this problem involves *recalibrating* $f_w(x)$ by postprocessing using the parametric function $\eta_V : \Delta^K \to \Delta^K$, where $V \in \mathcal{V} \subset \mathbb{R}^{d'}$ is the parameter learned by the recalibration dataset $S_{\text{re}} \sim \mathcal{D}^{n_{\text{re}}}$ at fixed $w$ and $S_{\text{re}}$ is independent of $S_{\text{tr}}$. In this procedure, the overall dataset is split into the training data $S_{\text{tr}}$ for learning $W$, the recalibration data $S_{\text{re}}$ for learning $V$, and the test data $S_{\text{te}}$ for the evaluation of the ECE. We define the recalibrated model as $\eta_v \circ f_w$.

The output after recalibration, $\eta_v \circ f_w$, is expected to yield a sufficiently small $\text{ECE}(\eta_v \circ f_w, S_{\text{re}})$. From a general-

ization perspective, it is important to theoretically investigate conditions under which $\eta_v \circ f_w$ also achieves low $\text{ECE}(\eta_v \circ f_w, S_{\text{te}})$. To this end, we define the following error term, resembling the standard generalization error typically defined via a loss function.

$$\begin{aligned}&\text{gen}(\eta_v \circ f_w, S_{\text{re}}, \text{ECE})\\&:= |\mathbb{E}_{S_{\text{te}}}\text{ECE}(\eta_v \circ f_w, S_{\text{te}}) - \text{ECE}(\eta_v \circ f_w, S_{\text{re}})|.\end{aligned}$$

We refer to this quantity as **the generalization error of the ECE** under $S_{\text{re}}$. In addition, the **total bias for the recalibration**, defined below, is required to be small:

$$\begin{aligned}&\text{Bias}(\eta_v \circ f_w, S_{\text{re}}, \text{TCE})\\&:= |\text{TCE}(\eta_v \circ f_w) - \text{ECE}(\eta_v \circ f_w, S_{\text{re}})|.\end{aligned}$$

Another goal of our study is to provide a theoretical understanding of these errors and biases by deriving their upper bounds using PAC-Bayes theory.

## 3. Analysis of ECE for Multiclass Classification

Here, we present a comprehensive analysis of the ECE *before* applying recalibration.

### 3.1. Total Bias Analysis for the ECE

In this section, we present our main analysis for the total bias. Under the smoothness assumptions commonly used in nonparametric estimation (Tsybakov, 2008), we show the following theorem:

**Assumption 1.** *Conditioned on $W = w$, the $L$-Lipschitz continuity holds for $P(Y = C|f_w(x)_C)$.*

**Theorem 1.** *Given $W = w$, for any positive constant $\varepsilon \in (0, 1)$ and $\lambda > 0$, with probability $1 - \varepsilon$ with respect to $S_{\text{te}}$, we have*

$$\text{Bias}(f_w, S_{\text{te}}, \text{TCE}) \leq \frac{1 + L}{B} + \frac{B \log 2 + \log \frac{1}{\varepsilon} + \frac{2\lambda^2}{n_{\text{te}}}}{\lambda}.$$

The complete proof is shown in Appendix B.5. The first term on the right-hand side is referred to as the *binning bias*, whereas the second term is called the *estimation bias* (Futami and Fujisawa, 2024). According to this theorem, the number of bins shows a trade-off relationship as follows. As we increase $B$, the binning bias decreases because we estimate the label frequency more precisely. On the other hand, the statistical bias increases since the number of samples allocated to each bin decreases. From this observation, we derive the optimal number of bins that minimize the total bias. By setting $\lambda = \sqrt{Bn_{\text{te}}}$, we can minimize the upper bound of Eq. (3) with respect to $B$. This yields $B = \mathcal{O}(n_{\text{te}}^{1/3})$, and

under this bin size, the order of the estimation bias becomes

$$\text{Bias}(f_w, S_{\text{te}}, \text{TCE}) = \mathcal{O}(1/n_{\text{te}}^{1/3}). \qquad (2)$$

This optimal order matches the result of the binary classification provided by Futami and Fujisawa (2024). This is natural, given that the binning ECE divides the top-1 predicted probabilities equally into $B$ bins in both cases.

The order of the total bias in Eq. (2) is tight from the non-parametric regression viewpoint. As discussed by Futami and Fujisawa (2024), the TCE measures the error between two functions, namely, $P(Y = C|f_w(x)_C)$ and $f_w(x)$, and we estimate $P(Y = C|f_w(x)_C)$ by binning, which is a nonparametric method. Thus, this is a problem of non-parametric regression on $[0, 1]$ under Lipschitz continuity. According to Tsybakov (2008), the error in nonparametric regression cannot be smaller than $\mathcal{O}(1/n_{\text{te}}^{1/3})$. Achieving an order smaller than this requires additional assumptions about the data distribution. Thus, the order of our bound is convincing under the current assumptions. We note that assuming Hölder continuity instead of Assumption 1 does not improve the order of the estimation bias owing to the bias caused by binning (see Appendix B.7 for details).

### 3.2. Total Bias Under the Training Dataset

Some recent algorithms (Kumar et al., 2018; Popordanoska et al., 2022; Wang et al., 2021a) focus on minimizing $\text{ECE}(f_w, S_{\text{tr}})$ without guaranteeing that the gap between $\text{TCE}(f_w)$ and $\text{ECE}(f_w, S_{\text{tr}})$ will remain small. Therefore, in this section, we theoretically investigate the conditions that an algorithm must satisfy to minimize this gap and to achieve a well-calibrated model. To this end, we conduct a theoretical analysis focusing on the following bias:

$$\text{Bias}(f_w, S_{\text{tr}}, \text{TCE}) \coloneqq |\text{TCE}(f_w) - \text{ECE}(f_w, S_{\text{tr}})|.$$

The following theorem presents the first PAC-Bayes bound for the generalization error of the ECE.

**Theorem 2.** *Under Assumption 1, for any fixed prior $\pi$, which is independent of $S_{\text{tr}}$, and posterior distribution $\rho$ over $W$, and for any positive constant $\varepsilon \in [0, 1]$ and $\lambda > 0$, with probability $1 - \varepsilon$ with respect to $S_{\text{tr}}$, we have*

$$\mathbb{E}_\rho \text{Bias}(f_W, S_{\text{tr}}, \text{TCE}) \qquad (3)$$

$$\leq \frac{1 + L}{B} + \frac{\text{KL}(\rho\|\pi) + B\log 2 + \log\frac{1}{\varepsilon} + \frac{2\lambda^2}{n_{\text{tr}}}}{\lambda}.$$

The complete proof is shown in Appendix B.3. Assuming that $\text{KL}(\rho\|\pi)$ is sufficiently smaller than $n_{\text{tr}}$, e.g., $\mathcal{O}(\log n_{\text{tr}})$ and independent of $B$, and setting $\lambda = \sqrt{Bn_{\text{tr}}}$, we can minimize the upper bound of Eq. (3) with respect to $B$. This yields $B = \mathcal{O}(n_{\text{tr}}^{1/3})$ and we have $\text{Bias}(f_w, S_{\text{tr}}, \text{TCE}) = \mathcal{O}(\log n_{\text{tr}}/n_{\text{tr}}^{1/3})$.

**Comparison with existing work:** The most closely related analysis is that performed by Futami and Fujisawa (2024); they evaluated a similar bias using information-theoretic generalization error analysis (IT analysis) (Xu and Raginsky, 2017). Their obtained bound and ours achieved the same order with respect to $n$, which improves previous results, such as those obtained by Gupta and Ramdas (2021). However, our result has two advantages over that obtained by Futami and Fujisawa (2024). The first advantage is that ours can treat the multiclass setting and the second advantage is that our theory builds on the PAC-Bayesian theory and the upper bound is optimizable, which leads to our novel recalibration algorithm, as shown in Section 4. On the other hand, Futami and Fujisawa (2024) used the supersample setting of the IT analysis, with which the optimizable upper-bound is difficult to obtain.

We also note that it seems difficult to use the approach of deriving tighter PAC-Bayes bounds on the basis of the KL divergence of the Bernoulli random variable $kl$ (Maurer, 2004; Foong et al., 2021). This is because the ECE is no longer a sum of i.i.d. random variables as it is in a test loss.

### 3.3. Analysis for the Top-$K$ Calibration Metric

In applications such as medical diagnosis (Jiang et al., 2012), it is often important to calibrate the prediction probabilities for any $k \in [K]$, not simply focusing on the top-1. In this case, the alternative metric

$$\text{CE}_K(f_w) \coloneqq \mathbb{E}\|P(Y|f_w(X)) - f_w(X)\|_1,$$

has been explored instead of TCE (Gruber and Buettner, 2022), where $\|\cdot\|_p$ is the $L^p$ distance in $\mathbb{R}^K$. The $\text{CE}_K$ metric measures the calibration performance of models across all classes. Here, we show that the direct estimation of $\text{CE}_K$ by binning results in a slow convergence due to the curse of dimensionality.

Let us consider the following binning scheme to estimate $\text{CE}_K$. Since we want to estimate the $K$-dimensional conditional probability in $[0, 1]^K$ by binning, we split each dimension $[0, 1]$ into $B'$ bins of the same width. After doing so, there are $B = (B')^K$ bins, and $[0, 1]^K$ is split into $B$ small regions. We refer to these regions as $\mathcal{I} \coloneqq \{I_i\}_{i=1}^B$. In this case, the ECE of $\text{CE}_K$ can be defined as

$$\text{ECE}_K(f_w, S_e) \coloneqq \sum_{i=1}^B p_i\|\bar{f}_{i,S_e} - \bar{p}_{i,S_e}\|_1, \qquad (4)$$

where $|I_i| \coloneqq \sum_{m=1}^{n_e} \mathbb{1}_{f_w(x_m) \in I_i}$, $\hat{p}_i \coloneqq |I_i|/n_e$, $\bar{f}_{i,S_e} \coloneqq \frac{1}{|I_i|}\sum_{m=1}^{n_e} \mathbb{1}_{f_w(x_m) \in I_i} f_w(x_m)$, and $\bar{p}_{i,S_e} \coloneqq \frac{1}{|I_i|}\sum_{m=1}^{n_e} \mathbb{1}_{f_w(x_m) \in I_i} \mathbf{e}_{y_m}$. The definition of Eq. (4) is similar to that of Eq. (1). The difference is that Eq. (1) only focuses on the top label. Under these settings, the upper bound of the total bias in $\text{ECE}_K$ is given as follows.

**Theorem 3.** *Assume that $f_w(x)$ has a probability density over $[0,1]^K$ and $\mathbb{E}[\mathbf{e}_Y|f_w(x)]$ satisfies the $L$-Lipschitz continuity. Then, for any fixed prior $\pi$, which is independent of $S_{\mathrm{tr}}$, and the posterior distribution $\rho$ over $W$, and for any positive constant $\varepsilon \in [0,1]$ and $\lambda > 0$, with probability $1 - \varepsilon$ with respect to $S_{\mathrm{tr}}$, we have*

$$\mathbb{E}_\rho|\mathrm{CE}_K(f_w) - \mathrm{ECE}_K(f_W, S_{\mathrm{tr}})| \tag{5}$$
$$\leq \frac{K(1+L)}{B^{\frac{1}{K}}} + \frac{\mathrm{KL}(\rho\|\pi) + BK\log 2 + \log\frac{1}{\varepsilon} + \frac{K^2\lambda^2}{2n_{\mathrm{tr}}}}{\lambda}.$$

By minimizing the upper bound of Eq. (5), we obtain the optimal bin size as $B = \mathcal{O}(n_{\mathrm{tr}}^{\frac{1}{K+2}})$, leading to an estimation bias of $\mathcal{O}(K^{\frac{3}{2}}\log n_{\mathrm{tr}}/n_{\mathrm{tr}}^{\frac{1}{K+2}})$. We can similarly show the result of the total bias as

$$\mathrm{Bias}(f_w, S_{\mathrm{te}}, \mathrm{CE}_K) := |\mathrm{CE}_K(f_w) - \mathrm{ECE}_K(f_w, S_{\mathrm{te}})|$$
$$= \mathcal{O}(1/n_{\mathrm{te}}^{1/(K+2)}).$$

The proofs are provided in Appendix B.8. This order aligns with the lower bound of $K$-dimensional nonparametric regression with Lipschitz continuity (Tsybakov, 2008), and it is much larger than that of the TCE ($\mathcal{O}(\log n_{\mathrm{tr}}/n_{\mathrm{tr}}^{1/3})$) owing to the curse of dimensionality. Thus, this result clarifies the pros and cons of the TCE; the TCE successfully circumvents the curse of dimensionality as described in Theorem 2 by focusing only on the top-1 predicted label, although it does not necessarily guarantee good calibration performance for all classes (Gruber and Buettner, 2022).

We note that our proof technique can be extended to cases where we focus on the calibration of any $K' \in [1, K]$ classes, resulting in a total bias of $\mathcal{O}(1/n_{\mathrm{te}}^{1/(K'+2)})$ (see Appendix B.10 for the details). This finding suggests that in scenarios with a large number of classes, it is crucial to selectively choose which classes to evaluate for calibration, especially when additional classes beyond the top label require calibration evaluation, rather than evaluating all classes. This approach is essential for achieving sample-efficient calibration evaluations.

## 4. PAC-Bayes bounds under Recalibration

In this section, we analyze the recalibration using PAC-Bayesian theory, leading to our novel recalibration algorithm in Section 4.2. All the proofs are provided in Appendix C.

### 4.1. Generalization and Total Bias under Recalibration

The following result shows the PAC-Bayes bound of the ECE under the recalibration context.

**Corollary 1.** *Assume that $n_{\mathrm{te}} = n_{\mathrm{re}} = n$. For both binary and multiclass classifications, conditioned on $W = w$, for*

*any fixed prior $\tilde{\pi}$, which is independent of $S_{\mathrm{re}}$, and posterior $\tilde{\rho}$ over $V$, and for any $\varepsilon \in (0,1)$ and $\lambda > 0$, with probability $1 - \varepsilon$ with respect to $S_{\mathrm{re}}$, we have*

$$\mathbb{E}_{\tilde{\rho}}\mathrm{gen}(\eta_V \circ f_w, S_{\mathrm{re}}, \mathrm{ECE}) \tag{6}$$
$$\leq \frac{\mathrm{KL}(\tilde{\rho}\|\tilde{\pi}) + B\log 2 + \log\frac{1}{\varepsilon} + \frac{4\lambda^2}{n}}{\lambda}.$$

This result highlights the importance of KL regularization in the parameter space—similar to the standard PAC-Bayes bound over $S_{\mathrm{tr}}$ (McAllester, 2003; Alquier et al., 2016)—in preventing overfitting and improving generalization. Since the recalibration data is available and we have fixed $w$, we only consider the posterior $\tilde{\rho}$ over $V$. Instead of fixing $W = w$, we can also obtain the PAC-Bayes bound by taking expectation over $W$ (see Appendix C.2).

**Discussion about the assumption of $n_{\mathrm{te}} = n_{\mathrm{re}} = n$:** Although a single data point can be used to evaluate loss in a standard generalization error analysis, multiple data points are required to evaluate the ECE, as it is a nonparametric estimator. Therefore, for a fair comparison, the same number of data points should be used in the analysis of the ECEs based on $S_{\mathrm{re}}$ and $S_{\mathrm{te}}$. We also remark that this assumption is not very restrictive since we split all the available data into training, recalibration, and test datasets, and most data are allocated to the training dataset in practice.

Next, we present our PAC-Bayes bound on the estimation bias of the ECE and TCE in the recalibration context after introducing the necessary assumption similar to Assumption 1.

**Assumption 2.** *Conditioned on $W = w$ and $V = v$, the $L$-Lipschitz continuity holds for $P(Y = C|\eta_v \circ f_w(X)_C)$ in the multiclass classification.*

**Corollary 2.** *Under Assumption 2, conditioned on $W = w$, for any fixed prior $\tilde{\pi}$, which is independent of $S_{\mathrm{re}}$, and posterior distribution $\tilde{\rho}$ over $V$, and for any positive constant $\varepsilon \in [0,1]$ and $\lambda > 0$, with probability $1 - \varepsilon$ with respect to $S_{\mathrm{re}}$, we have*

$$\mathbb{E}_{\tilde{\rho}}\mathrm{Bias}(\eta_V \circ f_w, S_{\mathrm{re}}, \mathrm{TCE})$$
$$\leq \frac{1+L}{B} + \frac{\mathrm{KL}(\tilde{\rho}\|\tilde{\pi}) + B\log 2 + \log\frac{1}{\varepsilon} + \frac{2\lambda^2}{n_{\mathrm{re}}}}{\lambda}. \tag{7}$$

Similarly to Eq. (3), there is a trade-off relationship with respect to $B$ in the above bound. By minimizing this upper bound with respect to $B$, we obtain the optimal bin size as $B = \mathcal{O}(n_{\mathrm{re}}^{1/3})$. For practical purposes, it is possible to set $B = \lfloor n_{\mathrm{re}}^{1/3} \rfloor$, where $\lfloor x \rfloor := \max\{m \in \mathbb{Z} : m \leq x\}$.

### 4.2. Proposed Recalibration Algorithm Based on PAC-Bayes Bounds

From Corollary 2, we can see that as long as $\mathrm{KL}(\tilde{\rho}\|\tilde{\pi})$ is regularized and the ECE under the recalibration data is small, the recalibrated function could exhibit a small TCE. This leads us to propose a *generalization-aware* recalibration method by minimizing our bound.

However, minimizing this function is difficult since the ECE is neither smooth nor differentiable. To avoid this issue, we focus on the following relation:

$$\mathrm{ECE}(\eta_v \circ f_w, S_{\mathrm{re}}) \le \mathbb{E}_{(X,Y) \sim S_{\mathrm{re}}} \|\mathbf{e}_Y - \eta_V \circ f_w(X)\|_2^2,$$

in the multiclass classification, where $\mathbb{E}_{(X,Y) \sim S_{\mathrm{re}}}$ denotes the expectation by the empirical distribution of $S_{\mathrm{re}}$, see Appendix C.3 for its derivation. This upper bound is the *Brier score* (Gruber and Buettner, 2022), which is a continuous and *differentiable* calibration metric. Therefore, under the posterior $\tilde{\rho}(v) = \tilde{\rho}(v;\theta)$ parametrized by $\theta$, we can minimize the following PAC-Bayes-based objective by gradient-based optimization:

$$\mathbb{E}_{\tilde{\rho}(v;\theta)} \underbrace{\mathbb{E}_{(X,Y) \sim S_{\mathrm{re}}} \|\mathbf{e}_Y - \eta_V \circ f_w(X)\|_2^2}_{\text{Brier score}} + \alpha \frac{\mathrm{KL}(\tilde{\rho}\|\tilde{\pi})}{n_{\mathrm{re}}}. \quad (8)$$

We expect the recalibrated models to achieve not only a low TCE or CE but also a high accuracy in practice. We thus consider minimizing the following objective with the classification loss $l_{\mathrm{acc}}$:

$$\mathbb{E}_{\tilde{\rho}(v;\theta)} \underbrace{\mathbb{E}_{(X,Y) \sim S_{\mathrm{re}}} l_{\mathrm{acc}}(Y, \eta_V \circ f_w(X))}_{\text{Expected classification loss}} \quad (9)$$

$$+ \underbrace{\mathbb{E}_{(X,Y) \sim S_{\mathrm{re}}} \|\mathbf{e}_Y - \eta_V \circ f_w(X)\|_2^2}_{\text{Brier score}} + \alpha \frac{\mathrm{KL}(\tilde{\rho}\|\tilde{\pi})}{n_{\mathrm{re}}}.$$

$$(10)$$

Since this objective function is derived from the PAC-Bayes bound for $l_{\mathrm{acc}}$ (Theorem 4 in Appendix B) and Corollary 2 via a union bound, it remains within the generalization error bound (See Corollary 5 in Appendix C.3 for details).

We refer to the recalibration achieved by minimizing Eqs. (8) and (10) as *PAC-Bayes recalibration* (PBR) and summarize these procedures in Algorithm 1. As an example of a recalibration model in PBR, we use the Gaussian process (GP) (Rasmussen and Williams, 2005). In this case, $\tilde{\rho}(v;\theta)$ is set as the multivariate Gaussian distribution, $\mathcal{N}(v; \mu, \Sigma)$, where $\mu$ and $\Sigma$ are the mean and covariance matrix, respectively. Furthermore, we can construct a *data-dependent* GP prior $\tilde{\pi}$ using the $M$ outputs $\{f_w(x,i)\}_{i=1}^M$ ($x \in S_{\mathrm{re}}$) obtained during training as inducing points, which is expected

---

**Algorithm 1** PAC-Bayes recalibration (PBR)

1: **INPUT:** All dataset $S_{\mathrm{all}}$, model $f_w$, recalibration model $\eta_v$, variational posterior $\tilde{\rho}(v;\theta)$
2: Splitting all dataset $S_{\mathrm{all}}$ to $S_{\mathrm{tr}}, S_{\mathrm{re}}, S_{\mathrm{te}}$.
3: Training $f_w$ using $S_{\mathrm{tr}}$.
4: Updating $\theta$ by minimizing Eq. (8) or (10) using $S_{\mathrm{re}}$ at fixed $w$.
5: Obtaining $V$ by taking the mean of $J$ samples $\{V_j\}_{j=1}^J \overset{\text{i.i.d.}}{\sim} \tilde{\rho}(v;\theta)$
6: (Option) Calculate the test ECE: $\mathrm{ECE}(\eta_v \circ f_w, S_{\mathrm{te}})$ by setting $B = \lfloor n_{\mathrm{re}}^{1/3} \rfloor$.
7: **RETURN:** $\eta_v \circ f_w$, $(\mathrm{ECE}(\eta_v \circ f_w, S_{\mathrm{te}}))$

---

to yield a small KL value. This is because $w$ is trained using $S_{\mathrm{tr}}$, and since $S_{\mathrm{tr}}$ and $S_{\mathrm{re}}$ are independent, this does not violate the independence of $S_{\mathrm{re}}$ from $\tilde{\pi}$ as assumed in Corollary 1.

A GP-based recalibration method was also proposed by Wenger et al. (2020), and, as far as we know, it is one of the methods that achieve *state-of-the-art* performance. Their recalibration was conducted via variational inference (VI), i.e., minimizing the following objective function derived from the evidence lower bound (ELBO):

$$\mathbb{E}_{\tilde{\rho}(v;\theta)} \mathbb{E}_{(X,Y) \sim S_{\mathrm{re}}} l_{\mathrm{acc}}(Y, \eta_V \circ f_w(X)) + \frac{\mathrm{KL}(\tilde{\rho}\|\tilde{\pi})}{n_{\mathrm{re}}},$$

where $l_{\mathrm{acc}}$ is defined as the softmax loss. Eq. (10) corresponds to this ELBO objective when $\alpha = 1$ and the Brier score is removed. This fact indicates that our PBR extends the GP-based recalibration to have flexible KL regularization and Brier score loss to control calibration performance on the basis of the PAC-Bayes notion. In this paper, we adopt in PBR the same computational strategy for GP-based recalibration as proposed by Wenger et al. (2020), in order to reduce the computational cost from $\mathcal{O}(N^3)$ to $\mathcal{O}(N^2 M)$. While this cost is not negligible, it remains justifiable in practice, as the size of $S_{\mathrm{re}}$ is significantly smaller than that of $S_{\mathrm{tr}}$. In Section 6.2, we compare PBR and GP-based recalibration using the same recalibration model settings.

## 5. Related work

Several studies have applied PAC-Bayes theory to classification metrics beyond accuracy. Ridgway et al. (2014) addressed AUC, Morvant et al. (2012) studied the confusion matrix in multiclass settings, and Sharma et al. (2024) examined conformal prediction as a tool for uncertainty quantification. In this work, we extend the PAC-Bayesian framework to the calibration metric.

Gupta and Ramdas (2021) and Kumar et al. (2019) examined the ECE in binary classification, focusing on *uniform mass binning* (UMB), which partitions the probability space into

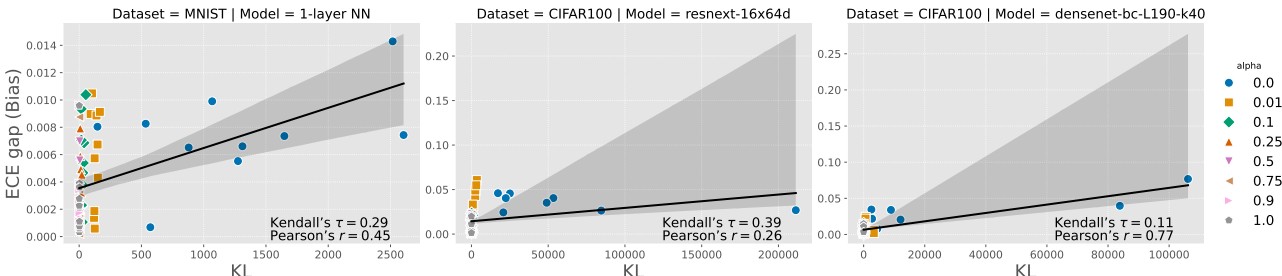

*Figure 1.* KL vs ECE gap on MNIST (Multiclass; 1-layer NN) and CIFAR-100 (Multiclass; ResNeXt-29 (Xie et al., 2017) and DenseNet-BC-190 (Huang et al., 2017)) dataset for each of various regularize parameters ($\alpha$).

*Table 1.* Results of multiclass classification experiments on MNIST and CIFAR-100. The symbols ($\downarrow$) and ($\uparrow$) refer to lower and higher values indicating a higher performance, respectively. We set $n_{\text{re}} = 1000$.

| Data | Model | Uncalibrate | | GP (Wenger et al., 2020) | | PBR (Ours; Eq. (8)) | | PBR (Ours; Eq. (10))) | | Temp. Scaling | |
|---|---|---|---|---|---|---|---|---|---|---|---|
| | | ECE ($\downarrow$) | Accuracy ($\uparrow$) | ECE ($\downarrow$) | Accuracy ($\uparrow$) | ECE ($\downarrow$) | Accuracy ($\uparrow$) | ECE ($\downarrow$) | Accuracy ($\uparrow$) | ECE ($\downarrow$) | Accuracy ($\uparrow$) |
| MNIST | XGBoost | .0036 ± .0003 | .9785 ± .0005 | .0038 ± .0006 | .9786 ± .0007 | **.0035 ± .0004** | .9785 ± .0007 | .0037 ± .0004 | .9785 ± .0006 | .0055 ± .0017 | .9785 ± .0005 |
| | random forest | .1428 ± .0007 | .9659 ± .0005 | .0313 ± .0056 | .9663 ± .0011 | .0065 ± .0017 | .9639 ± .0018 | **.0057 ± .0028** | .9656 ± .0017 | .0062 ± .0009 | .9659 ± .0005 |
| | 1layer NN | .0164 ± .0004 | .9760 ± .0005 | .0101 ± .0023 | .9740 ± .0008 | .0137 ± .0014 | .9751 ± .0009 | .0112 ± .0019 | .9740 ± .0016 | **.0062 ± .0026** | .9760 ± .0005 |
| CIFAR100 | alexnet | .2548 ± .0007 | .4374 ± .0008 | .2548 ± .0008 | .4373 ± .0009 | .2548 ± .0007 | .4372 ± .0009 | .2548 ± .0007 | .4373 ± .0009 | **.0216 ± .0037** | .4374 ± .0008 |
| | WRN-28-10-drop | .0568 ± .0009 | .8131 ± .0011 | .0567 ± .0010 | .8129 ± .0011 | .0504 ± .0054 | .8129 ± .0011 | **.0349 ± .0093** | .8129 ± .0011 | .0374 ± .0028 | .8131 ± .0011 |
| | resnext-8x64d | .0401 ± .0010 | .8229 ± .0013 | .0400 ± .0010 | .8229 ± .0014 | .0320 ± .0050 | .8228 ± .0014 | **.0310 ± .0082** | .8230 ± .0014 | .0401 ± .0019 | .8229 ± .0013 |
| | resnext-16x64d | .0405 ± .0012 | .8231 ± .0013 | .0403 ± .0013 | .8230 ± .0014 | **.0305 ± .0056** | .8231 ± .0013 | .0321 ± .0071 | .8230 ± .0014 | .0420 ± .0025 | .8231 ± .0013 |
| | densenet-bc-L190-k40 | .0639 ± .0008 | .8230 ± .0007 | .0639 ± .0008 | .8229 ± .0007 | .0630 ± .0045 | .8228 ± .0006 | .0618 ± .0054 | .8229 ± .0007 | **.0222 ± .0013** | .8230 ± .0007 |

intervals of equal mass. They also analyzed a recalibration method based on UMB. Sun et al. (2023) further analyzed binning-based recalibration for binary classification under a similar Lipschitz assumption. Futami and Fujisawa (2024) investigated the generalization of the ECE using IT analysis under a comparable setting, and derived a total bias of order $\mathcal{O}(1/n^{1/3})$. In contrast, our work provides an analysis of the binning ECE in the *multiclass* setting—a direction that has not yet been explored. Furthermore, our bound is optimizable, which allows us to derive a novel recalibration algorithm for the ECE criterion that explicitly considers generalization. A natural direction for future work is to extend our analysis to settings that employ UMB.

The analysis of multiclass calibration remains limited compared to that of binary classification. For instance, Zhang et al. (2020) investigated the kernel-based $\text{CE}_K$; however, their analysis does not address generalization or recalibration, which fundamentally distinguishes our approach. Moreover, while they noted that the binning-based $\text{CE}_K$ suffers from the curse of dimensionality, no formal justification was provided. In contrast, our Theorem 3 formally validates this phenomenon. Gruber and Buettner (2022) discussed various calibration metrics and clarified that the TCE is a weaker calibration metric than $\text{CE}_K$. Theorem 3 further demonstrates that the naive estimator of $\text{CE}_K$ suffers more severely from the curse of dimensionality than the TCE.

# 6. Experiments

In Section 6.1, we verify the correlation between our bounds in Corollary 1 and the generalization performance of recalibration. We then evaluate the effectiveness of our PBR described in Section 4.2. Due to page limitations, we primarily report results from multiclass classification experiments on the MNIST (LeCun et al., 1989) and CIFAR-100 (Krizhevsky, 2009) datasets. Additional experimental results are provided in Appendix E, and details of our experimental settings, including model specifications, are summarized in Appendix D. For ECE evaluation, we set $B = \lfloor n_{\text{re}}^{1/3} \rfloor$ based on our findings.

## 6.1. Verification of Our Bounds

In this section, we empirically investigate the correlation between the KL divergence term in Eq. (6) and the *ECE gap*, which we define as the absolute difference between the ECEs of the test and training datasets. Our goal here is to assess whether the KL divergence in our bound can adequately explain the observed generalization performance. To investigate this, we first conducted recalibration experiments using PBR (Eq. (8)) with various KL constraints ($\alpha$). We then measured the KL values and the ECE gap after recalibration using 10-fold cross-validation, setting $n_{\text{re}} = 1000$. As evaluation metrics, we adopted Pearson's and Kendall's rank-correlation coefficients, which are widely used to assess the consistency between generalization metrics and actual generalization performance (Jiang et al., 2020; Wang et al., 2021b; Kawaguchi et al., 2023).

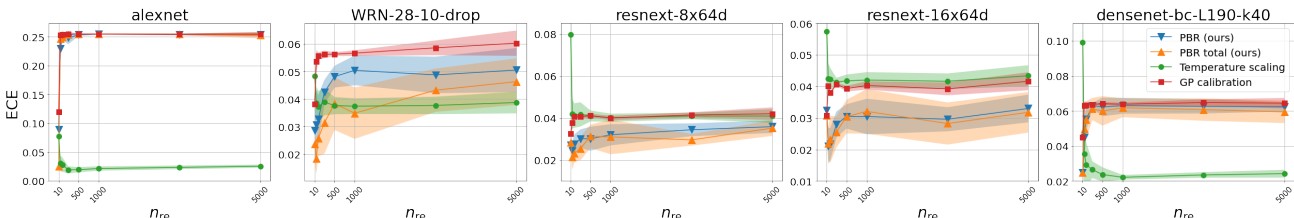

*Figure 2.* $n_{\mathrm{re}}$ vs ECE on the test dataset on the experiments with CIFAR-100 dataset. PBR (ours) and PBR total (ours) correspond to the results obtained by optimizing Eqs. (8) and (10), respectively.

A portion of the results is presented in Figure 1. These results confirm the positive correlation between the KL divergence and the ECE generalization gap. This confirms that our bounds are valid in explaining the generalization performance. It is also evident that the ECE gap is noisy. This is due to the slow convergence rate of the binning ECE estimator, which scales as $\mathcal{O}(1/n_{\mathrm{re}}^{1/3})$ even under the optimal $B$. With $n_{\mathrm{re}} = 1000$, the estimation bias can contribute noise on the order of $\mathcal{O}(1/10)$, meaning that when models achieve an ECE below 0.1, the correlation could be obscured by noise. A straightforward approach to reducing noise is increasing the size of the recalibration dataset. However, due to the $\mathcal{O}(1/n_{\mathrm{re}}^{1/3})$ convergence rate, this is impractical, highlighting the inherent difficulty of precise ECE evaluation.

### 6.2. Empirical Comparison of Recalibration Methods

In this section, we compare our proposed PBR and existing standard or state-of-the-art recalibration methods, temperature scaling (Guo et al., 2017) and GP calibration (Wenger et al., 2020). The optimizable parameter $\alpha$ in PBR is selected via grid search from $\{0., 0.01, 0.1, 0.25, 0.5, 0.75, 0.9, 1.0\}$. Wenger et al. (2020) also reported results for recalibration methods originally developed for binary classification, such as Platt scaling (Platt, 1999), isotonic regression (Zadrozny and Elkan, 2002), Beta calibration (Kull et al., 2017), and Bayesian binning into quantiles (BBQ) (Naeini et al., 2015), by adapting these methods to the multiclass setting via one-vs-all conversion. To ensure a fair comparison, however, we included only methods that are directly compatible with multiclass classification as baselines. A comparison of these methods in the binary classification setting can be found in Appendix E.

We summarize the results in Table 1. The results show that our method consistently improves calibration performance over existing approaches, particularly outperforming the GP-based recalibration method. Additionally, we found that, contrary to the findings of Wenger et al. (2020), *temperature scaling sometimes performs better than GP-based recalibration* in terms of ECE when using the optimal $B$.

One possible reason for this discrepancy is that Wenger et al. (2020) reported the ECE values with $B = 100$. According to our analysis, this setting minimizes estimation bias when $n_{\mathrm{te}} = \mathcal{O}(100^3)$. However, since the actual $n_{\mathrm{te}}$ used in their experiments is significantly smaller (see Section 4.1 of Wenger et al. (2020)), their reported results may have been affected by severe bias. This finding underscores the fundamental importance of selecting an appropriate number of bins based on a well-grounded theoretical framework for conducting reliable empirical evaluations.

These findings also indicate that the GP recalibration sometimes yield results that are not significantly different from the uncalibrated setting. In contrast, our PBR shows substantial improvements over both uncalibrated and GP-based methods. It is worth mentioning that our additional experiments in Appendix E confirm that GP-based methods, including PBR, do not consistently outperform standard recalibration techniques such as temperature scaling. These observations suggest that the use of GP for recalibration methods does not guarantee superior performance and emphasize the importance of carefully selecting a recalibration strategy that aligns with the characteristics of the dataset and model.

We also conducted additional experiments using $n_{\mathrm{re}} = \{10, 50, 100, 250, 500, 1000, 3000, 5000\}$ to further examine how the performance of each recalibration method varies with different values of $n_{\mathrm{re}}$. The parameter $\alpha$ in PBR was selected following the same procedure as described earlier.

We show the results in Figure 2. From these results, we can see that our PBR achieves relatively good performance under various settings of $n_{\mathrm{re}}$, consistently improving the performance of GP-based recalibration methods. However, in experiments with AlexNet, GP-based recalibration including PBR fails except when $n_{\mathrm{re}}$ is small. Given AlexNet's lower accuracy (see the *Uncalibrated* column in Table 1), the following explanation may be plausible. All of our recalibration methods construct the GP prior using the outputs of the pretrained model $f_w$ as inducing points. If $f_w$ fails to classify the training data accurately, the resulting GP prior will be misaligned with the true data distribution. Consequently, the recalibrated posterior is regularized toward

this inappropriate prior, which can degrade overall performance. The temperature scaling appears to perform stable and effective recalibration as $n_{re}$ increases, regardless of accuracy, and it even achieves a higher performance than PBR in some experimental settings; however, it is considered to cause overfitting when $n_{re}$ is small because the temperature scaling lacks regularization terms such as the KL divergence used in PBR. This underscores the importance of carefully selecting recalibration methods based on the characteristics of the model and dataset as denoted previously. Summarizing these results, in practical applications—at least for multiclass classification—it would be advisable to try both temperature scaling and our PBR, verify their performance using validation data, and choose the method that contributes more to improving calibration performance.

## 7. Conclusion and Limitations

In this paper, we analyzed the generalization error and estimation bias of the ECE in multiclass classification for the first time, resulting in several non-asymptotic bounds and the practical optimal bin size. Moreover, we developed a new generalization-aware recalibration algorithm based on our PAC-Bayes bound. Our analysis also revealed the slow convergence of the ECE through binning, which is a fundamental limitation. As discussed in Section 3.2, the binning method cannot leverage the underlying smoothness of the data distribution. Our numerical results suggest that the limited number of test data points makes the numerical evaluation of the ECE unstable in practice. We believe that other nonparametric methods for estimating the CE or TCE may address the limitations identified in this study. Investigating the effectiveness of such methods constitutes an important direction for future work. Another key avenue for future research is to extend the theoretical analysis to cases where alternative methods for estimating the TCE, such as UMB, are employed.

## Acknowledgments

MF was previously supported by the RIKEN Special Post-doctoral Researcher Program and JST ACT-X Grant Number JPMJAX210K, Japan. MF is currently supported by KAKENHI Grant Number 25K21286. FF was supported by JSPS KAKENHI Grant Number JP23K16948. FF was supported by JST, PRESTO Grant Number JPMJPR22C8, Japan.

## Impact Statement

This paper presents work whose goal is to advance the field of Machine Learning. There are many potential societal consequences of our work, none which we feel must be specifically highlighted here.

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

## A. Remark about the binary setting

Here we introduce the calibration for binary classification. Although, the primary goal of our study is the multiclass setting, our theory can handle for both the binary and multicalss settings.

Let $\mathcal{Y} = \{0, 1\}$ and let $f_w : \mathcal{X} \to [0, 1]$ be a model, where the output corresponds to the model's confidence that the label is 1. In the context of calibration, it is expected that not only $f_w$ has high accuracy, but also its predictive probability aligns well in the actual label frequency $P(Y|f_w(X))$. A model is *well-calibrated* (Widmann et al., 2021; Gupta et al., 2020) when $P(Y|f_w(X)) = f_w(X)$ holds; however, it is difficult to confirm this. Therefore, an alternative metric called as the calibration error (CE) (Widmann et al., 2021; Gupta et al., 2020) is typically used. Given $W = w$, the CE is defined as

$$\mathrm{CE}(f_w) \coloneqq \mathbb{E}|P(Y = 1|f_w(X)) - f_w(X)| = \mathbb{E}|\mathbb{E}[Y|f_w(X)] - f_w(X)|.$$

Unfortunately, evaluating the CE is still infeasible because $\mathbb{E}[Y|f_w(X)]$ is intractable. To resolve this issue, we evaluate the calibration performance by constructing the estimator of the CE. The widely used estimator is the *expected calibration error* (ECE), where *binning* is used to estimate $\mathbb{E}[Y|f_w(X)]$ (Guo et al., 2017; Zadrozny and Elkan, 2001; 2002). The ECE is calculated by partitioning the range of the predictive probability $[0, 1]$ into $B$ intervals $\mathcal{I} = \{I_i\}_{i=1}^B$ (called *bins*) and averaging within each bin using an evaluation dataset $S_e \coloneqq \{z_m\}_{m=1}^{n_e} \in \mathcal{Z}^{n_e}$, where we assume $n_e \geq 2B$. That is, the ECE is defined as

$$\mathrm{ECE}(f_w, S_e) \coloneqq \sum_{i=1}^B p_i |\bar{f}_{i,S_e} - \bar{y}_{i,S_e}|,$$

where $|I_i| \coloneqq \sum_{m=1}^{n_e} \mathbb{1}_{f_w(x_m) \in I_i}$, $p_i \coloneqq \frac{|I_i|}{n_e}$, $\bar{f}_{i,S_e} \coloneqq \frac{1}{|I_i|} \sum_{m=1}^{n_e} \mathbb{1}_{f_w(x_m) \in I_i} f_w(x_m)$, and $\bar{y}_{i,S_e} \coloneqq \frac{1}{|I_i|} \sum_{m=1}^{n_e} \mathbb{1}_{f_w(x_m) \in I_i} y_m$. In the above, bins are set by dividing the interval $[0, 1]$ into $B$ bins of the same width: $I_1 = (0, 1/B], I_2 = (1/B, 2/B], \ldots, I_B = ((B-1)/B, B]$. Setting $S_e = S_{\mathrm{te}}$, for instance, corresponds to evaluating the ECE on the test dataset.

Similarly to the multiclass setting, we define the total bias of CE as $\mathrm{Bias}(\eta_v(f_w), S_e, \mathrm{CE}) \coloneqq |\mathrm{CE}(\eta_v(f_w)) - \mathrm{ECE}(\eta_v(f_w), S_e)|$ for binary classification.

## B. Proofs and discussion of Section 3

For the latter purpose, we define $[n] = \{1, \ldots, n\}$ for $n \in \mathbb{N}$ and $\lfloor x \rfloor = \max\{m \in \mathbb{Z} : m \leq x\}$.

### B.1. Additional preliminary

We use the following lemma repeatedly in our proofs.

**Lemma 1** (Used in the proof of McDiarmid's inequality). *(Boucheron et al., 2013) We say that a function $f : \mathcal{X} \to \mathbb{R}$ has the bounded difference property if for some nonnegative constants $c_1, \ldots, c_n$,*

$$\sup_{x_1, \ldots, x_n, x_i' \in \mathcal{X}} |f(x_1, \ldots, x_n) - f(x_1, \ldots, x_{i-1}, x_i', x_{i+1}, \ldots, x_n)| \leq c_i, \quad 1 \leq i \leq n.$$

*If $X_1, \ldots, X_n$ are independent random variables taking values in $\mathcal{X}$ and $f$ has the bounded difference property with constants $c_1, \ldots, c_n$, then for any $t \in \mathbb{R}$, we have*

$$\mathbb{E}\left[ e^{t(f(X_1, \ldots, X_n) - \mathbb{E}[f(X_1, \ldots, X_n)])} \right] \leq e^{\frac{t^2}{8} \sum_{i=1}^n c_i^2}.$$

The following theorem is the most basic form of the PAC-Bayes generalization bounds (Alquier et al., 2016).

**Theorem 4.** *For any fixed prior $\pi$, which is independent of $S_{\mathrm{tr}}$, and posterior distribution $\rho$ over $W$ and for any $\varepsilon \in (0, 1)$ and $\lambda > 0$, with probability $1 - \varepsilon$, we have*

$$\mathbb{E}_\rho \mathrm{gen}(f_W, S_{\mathrm{tr}}, l) \leq \frac{\mathrm{KL}(\rho\|\pi) + \log \mathbb{E}_{\pi, S_{\mathrm{tr}}} e^{\lambda(L(w,\mathcal{D}) - \hat{L}(w,S_{\mathrm{tr}}))} + \log \frac{1}{\epsilon}}{\lambda}. \tag{11}$$

## B.2. Reformulations of the ECE to loss functions

Here, we introduce the reformulations of the ECE, which is the essential technique in our proof. Note that when focusing on the TCE of the multiclass setting, its properties are almost the same as those of the binary classification. This is because TCE only considers the top label and by regarding the top label corresponding to label 1 in the binary classification and other labels as 0 in the binary classification, then it is clear that the ECE of the multiclass and binary setting are almost identical.

BINARY CLASSIFICATION

We first focus on the binary setting. The training ECE can be reformulated as the empirical mean of $Y - f_w(X)$ in each bin as follows

$$\text{ECE}(f_w, S_e) := \sum_{i=1}^{B} |\mathbb{E}_{(X,Y) \sim S_e}(Y - f_w(X)) \cdot \mathbb{1}_{f_w(X) \in I_i}|, \tag{12}$$

which follows immediately from the definitions. Here $\mathbb{E}_{S_e}$ is the empirical expectation by the evaluation dataset.

Recall that the conditional expectation of $f_w$ given bins is defined as

$$f_{\mathcal{I}}(x) := \sum_{i=1}^{B} \mathbb{E}[f_w(X)|f_w(X) \in I_i] \cdot \mathbb{1}_{f_w(X) \in I_i}.$$

The following relation holds

$$\text{CE}(f_{\mathcal{I}}) = \sum_{i=1}^{B} |\mathbb{E}_{(X,Y) \sim \mathcal{D}}(Y - f_w(X)) \cdot \mathbb{1}_{f_w(X) \in I_i}|, \tag{13}$$

where $\text{CE}(f_{\mathcal{I}})$ means the CE of the function $f_{\mathcal{I}}$. We can derive this as follows; By definition, we have

$$\sum_{i=1}^{B} |\mathbb{E}_{(X,Y) \sim \mathcal{D}}(Y - f_w(X)) \cdot \mathbb{1}_{f_w(X) \in I_i}|$$

$$= \sum_{i=1}^{B} |\mathbb{E}_{(X,Y) \sim \mathcal{D}}[(Y - f_w(X)) \cdot \mathbb{1}_{f_w(X) \in I_i}]|$$

$$= \sum_{i=1}^{B} P(f_w(X) \in I_i)|\mathbb{E}|\mathbb{E}[Y|f_w(X) \in I_i] - \mathbb{E}[f_w(X)|f_w(X) \in I_i]|,$$

where we used the definition of the conditional expectation. On the other hand, We have

$$\text{CE}(f_{\mathcal{I}}) = \mathbb{E}|\mathbb{E}[Y|f_{\mathcal{I}}(x)] - f_{\mathcal{I}}(x)|$$

$$= \sum_{i=1}^{B} \mathbb{E}\left[|\mathbb{E}[Y|f_{\mathcal{I}}(x)] - f_{\mathcal{I}}(x)| \cdot \mathbb{1}_{f_{\mathcal{I}}(X) \in I_i}\right]$$

$$= \sum_{i=1}^{B} P(f_{\mathcal{I}}(x) \in I_i)\mathbb{E}\left[|\mathbb{E}[Y|f_{\mathcal{I}}(x)] - f_{\mathcal{I}}(x)|f_{\mathcal{I}}(X) \in I_i\right]$$

$$= \sum_{i=1}^{B} P(f_w(X) \in I_i)\mathbb{E}|\mathbb{E}[Y|f_w(X) \in I_i] - \mathbb{E}[f_w(X) \in I_i]|,$$

where we used the tower property. This concludes the proof.

Thus, we can transform the loss and ECEs by Eqs. (12) and (13).

MULTICLASS CLASSIFICATION

Next, we consider the ECE for the multiclass setting. Let $C := \arg\max_k f_w(X)_k$. Then by definition of ECE, we have

$$\mathrm{ECE}(f_w, S_e) = \sum_{i=1}^{B} \frac{|I_i|}{n_e} |\bar{f}_{i,S_e} - \bar{p}_{i,S_e}| = \sum_{i=1}^{B} |\mathbb{E}_{(X,Y) \sim S_e}(\mathbb{1}_{Y=C} - f_w(X)_C) \cdot \mathbb{1}_{f_w(X)_C \in I_i}|, \tag{14}$$

where $|I_i| := \sum_{m=1}^{n_e} \mathbb{1}_{f_w(x_m)_C \in I_i}$, $\bar{f}_{i,S_e} := \frac{1}{|I_i|} \sum_{m=1}^{n_e} \mathbb{1}_{f_w(x_m)_C \in I_i} f_w(x_m)_C$, and $\bar{p}_{i,S_e} := \frac{1}{|I_i|} \sum_{m=1}^{n_e} \mathbb{1}_{f_w(x_m)_C \in I_i} \mathbb{1}_{y_m=C}$. As for the conditional function, we define

$$f_{\mathcal{I}}^C(x) := \sum_{i=1}^{B} \mathbb{E}[f_w(X)_C | f_w(X)_C \in I_i] \cdot \mathbb{1}_{f_w(x)_C \in I_i}.$$

Then by repeating the exactly same proof for Eq. (13), we obtain

$$\mathrm{TCE}(f_{\mathcal{I}}^C(x)) = \sum_{i=1}^{B} |\mathbb{E}_{(X,Y) \sim \mathcal{D}}(\mathbb{1}_{Y=C} - f_w(X)_C) \cdot \mathbb{1}_{f_w(X)_C \in I_i}|, \tag{15}$$

We can see that the ECE of the binary and multiclass classification essentially represents the same quantity; in the multiclass setting, by re-labeling $C$ as 1 and others 0, they represent the same quantity.

### B.3. Proof of Theorem 2 (Bias under the training dataset)

Here we present a generalized version of Theorem 2 that includes the binary classification. After that, we show the proof of Theorem 1, which can be directly obtained from Theorem 2.

To simplify the notation, we express $n_{\mathrm{tr}} = n$.

**Theorem 5.** *Under Assumption 1, for any fixed prior $\pi$, which is independent of $S_{\mathrm{tr}}$, and posterior distribution $\rho$ over $W$ and any positive constant $\varepsilon \in [0, 1]$ and $\lambda > 0$, with probability $1 - \varepsilon$ with respect to $S_{\mathrm{tr}}$, we have*

$$\mathbb{E}_\rho \mathrm{Bias}(f_w, S_{\mathrm{tr}}, \mathrm{CE}) \leq \frac{1+L}{B} + \frac{\mathrm{KL}(\rho \| \pi) + B \log 2 + \log \frac{1}{\varepsilon} + \frac{2\lambda^2}{n}}{\lambda} \quad \textit{(Binary classification)},$$

$$\mathbb{E}_\rho \mathrm{Bias}(f_w, S_{\mathrm{tr}}, \mathrm{TCE}) \leq \frac{1+L}{B} + \frac{\mathrm{KL}(\rho \| \pi) + B \log 2 + \log \frac{1}{\varepsilon} + \frac{2\lambda^2}{n}}{\lambda} \quad \textit{(Multiclass classification)}.$$

*Proof.* BINARY SETTING

We start from the binary classification setting. To analyze the bias caused by binning, following Futami and Fujisawa (2024), we define the conditional expectation of $f_w$ given bins as $f_{\mathcal{I}}(x) := \sum_{i=1}^{B} \mathbb{E}[f_w(X) | f_w(X) \in I_i] \cdot \mathbb{1}_{f_w(x) \in I_i}$. With this definition, we decompose the estimation bias as follows:

$$\mathrm{Bias}(f_w, S_{\mathrm{tr}}, \mathrm{CE}) = |\mathrm{CE}(f_w) - \mathrm{ECE}(f_w, S_{\mathrm{te}})| \leq |\mathrm{CE}(f_w) - \mathrm{CE}(f_{\mathcal{I}})| + |\mathrm{CE}(f_{\mathcal{I}}) - \mathrm{ECE}(f_w, S_{\mathrm{te}})|.$$

where the first term is called as the *binning bias*, which arises from nonparametric estimation via binning, and the latter as the *statistical bias* caused by estimation on finite data points. We then evaluate these terms separately.

As for the first term, we show that

$$|\mathrm{CE}(f_w) - \mathrm{CE}(f_{\mathcal{I}})| \leq \mathbb{E}||\mathbb{E}[Y|f_w(X)] - \mathbb{E}[Y|f_{\mathcal{I}}(X)]| + \mathbb{E}|f_w(X) - f_{\mathcal{I}}(X)|. \tag{16}$$

holds. Note that $P(Y = 1 | f_w(x)) = \mathbb{E}[Y|f_w(x)]$ holds for the binary classification. First, we can show that

$$\mathrm{CE}(f_{\mathcal{I}}) \leq \mathrm{CE}(f_w),$$

by Jensen inequality with respect to the conditional expectation. This has been proved in Proposition 3.3 in Kumar et al. (2019). Next, we upper bound $\text{CE}(f_w)$ as follows

$$
\begin{aligned}
\text{CE}(f_w) &= \mathbb{E}\left[|\mathbb{E}[Y|f_w(X)] - f_w(X)|\right] \\
&= \sum_{i=1}^{B} \mathbb{E}[\mathbb{1}_{f_w(X) \in I_i} \cdot |\mathbb{E}[Y|f_w(X)] - f_w(X)|] \\
&= \sum_{i=1}^{B} P(f_w(X) \in I_i)\mathbb{E}[|\mathbb{E}[Y|f_w(X)] - f_w(X)||f_w(X) \in I_i] \\
&= \sum_{i=1}^{B} P(f_w(X) \in I_i)\mathbb{E}[|\mathbb{E}[Y|f_w(X)] - \mathbb{E}[f_w(X)|f_w(X) \in I_i] \\
&\quad + \mathbb{E}[f_w(X)|f_w(X) \in I_i] - f_w(X)||f_w(X) \in I_i] \\
&\leq \sum_{i=1}^{B} P(f_w(X) \in I_i)\mathbb{E}[|\mathbb{E}[Y|f_w(X)] - \mathbb{E}[Y|f_w(X) \in I_i]| \\
&\quad + \sum_{i=1}^{B} P(f_w(X) \in I_i)\mathbb{E}|\mathbb{E}[Y|f_w(X) \in I_i] - \mathbb{E}[f_w(X)|f_w(X) \in I_i]| \\
&\quad + \sum_{i=1}^{B} P(f_w(X) \in I_i)\mathbb{E}[|\mathbb{E}[f_w(X)|f_w(X) \in I_i] - f_w(X)||f_w(X) \in I_i],
\end{aligned}
$$

where the second term is $\text{CE}(f_{\mathcal{I}})$, we finished the proof of Eq. (16).

Then we can show that

$$
|\text{CE}(f_w) - \text{CE}(f_{\mathcal{I}})| \leq \mathbb{E}||\mathbb{E}[Y|f_w(X)] - \mathbb{E}[Y|f_{\mathcal{I}}(X)]| + \mathbb{E}|f_w(X) - f_{\mathcal{I}}(X)| \leq \frac{L}{B} + \frac{1}{B}, \tag{17}
$$

where we used the Lipschitz continuity of the function and the fact that we split the function with equal width $1/B$. Now, we have

$$
\begin{aligned}
\mathbb{E}_{\rho}|\text{CE}(f_w) - \text{ECE}(f_w, S_{\text{tr}})| &\leq \mathbb{E}_{\rho}|\text{CE}(f_w) - \text{CE}(f_{\mathcal{I}})| + \mathbb{E}_{\rho}|\text{CE}(f_{\mathcal{I}}) - \text{ECE}(f_w, S_{\text{tr}})| \\
&\leq \frac{L}{B} + \frac{1}{B} + \mathbb{E}_{\rho}|\text{CE}(f_{\mathcal{I}}) - \text{ECE}(f_w, S_{\text{tr}})|,
\end{aligned}
$$

We will bound the second term. For this purpose, we use Theorem 6 in Appendix B.4, this concludes the proof for the binary setting. □

*Proof.* MULTICLASS SETTING

Next, for the multiclass setting, similarly to the binary case, Similarly to the binary setting, we define the conditional expectation of $f_w$ given bins as $f_{\mathcal{I}}^C(x) := \sum_{i=1}^{B} \mathbb{E}[f_w(X)_C|f_w(X)_C \in I_i] \cdot \mathbb{1}_{f_w(x)_C \in I_i}$. With this definition, we decompose the estimation bias as follows:

$$
\text{Bias}(f_w, S_{\text{tr}}, \text{TCE}) = |\text{TCE}(f_w) - \text{ECE}(f_w, S_{\text{te}})| \leq |\text{TCE}(f_w) - \text{TCE}(f_{\mathcal{I}}^C)| + |\text{TCE}(f_{\mathcal{I}}^C) - \text{ECE}(f_w, S_{\text{te}})|.
$$

As for the multiclass setting, similarly to the binary case, we have

$$
|\text{TCE}(f_w) - \text{ECE}(f_w, S_{\text{tr}})| \leq |\text{TCE}(f_w) - \text{TCE}(f_{\mathcal{I}})| + |\text{TCE}(f_{\mathcal{I}}) - \text{ECE}(f_w, S_{\text{tr}})|. \tag{18}
$$

Then similarly to Eq. (16) in the binary classification, we can show that

$$
|\text{TCE}(f_w) - \text{TCE}(f_{\mathcal{I}}^C)| \leq \mathbb{E}|P(Y = C|f_w(X)_C) - P(Y = C|f_{\mathcal{I}}^C(X)| + \mathbb{E}|f_w(X)_C - f_{\mathcal{I}}^C(X)|
$$

by the almost identical proof. We then upper-bound them

$$|\mathrm{TCE}(f_w) - \mathrm{TCE}(f_{\mathcal{I}}^C)| \leq \mathbb{E}|P(Y = C|f_w(X)_C) - P(Y = C|f_{\mathcal{I}}^C(X)| + \mathbb{E}|f_w(X)_C - f_{\mathcal{I}}^C(X)| \leq \frac{L}{B} + \frac{1}{B}$$

using the Lipschitz continuity and the property of the binning. We substitute this into Eq. (18), we have

$$
\begin{aligned}
&\mathbb{E}_\rho|\mathrm{TCE}(f_w) - \mathrm{ECE}(f_w, S_{\mathrm{tr}})| \\
&\leq \mathbb{E}_\rho|\mathrm{TCE}(f_w) - \mathrm{TCE}(f_{\mathcal{I}})| + \mathbb{E}_\rho|\mathrm{TCE}(f_{\mathcal{I}}) - \mathrm{ECE}(f_w, S_{\mathrm{tr}})| \\
&\leq \frac{L}{B} + \frac{1}{B} + \mathbb{E}_\rho|\mathrm{TCE}(f_{\mathcal{I}}) - \mathrm{ECE}(f_w, S_{\mathrm{tr}})|
\end{aligned}
$$

Then using Theorem 6 in Appendix B.4 for the second term, this concludes the proof. $\qquad\square$

### B.4. Auxiliary result about the statistical bias

Here we present the auxiliary result, which is used in Appendix B.3.

First, we introduce the assumption, which is used in existing analysis Gupta and Ramdas (2021) and **not necessarily in our proof**;

**Assumption 3.** *Given $W = w$, $f_w(x)_C$ is absolutely continuous with respect to the Lebesgue measure.*

As for the binary setting, we assume that $f_w(x)$ is absolutely continuous with respect to the Lebesgue measure. This assumption means that $f_w(x)$ has a probability density, it is satisfied without loss of generality as elaborated in Appendix C in Gupta and Ramdas (2021). Although **this assumption is not required for our main results**, if we assume this, we can improve the coefficient of the upper bound. Thus in the below, we provide the proof with and without this assumption simultaneously.

Thanks to this assumption, $f_w(X)$ takes distinct values almost surely, so for example, the situation when all training samples take the same predicted probability will be circumvented. When considering the multiclass setting with $\mathrm{CE}_K$, this property plays an important role in eliminating the curse of dimensionality, see Appendix B.9.

**Theorem 6.** *Assume that $n_{\mathrm{tr}} = n$. For both binary and multiclass settings, for any fixed prior $\pi$, which is independent of $S_{\mathrm{tr}}$, and posterior distribution $\rho$ over $W$ and any positive constant $\varepsilon \in (0, 1)$ and $\lambda > 0$, with probability $1 - \varepsilon$ with respect to $S_{\mathrm{tr}}$, we have*

$$\mathbb{E}_\rho|\mathrm{CE}(f_{\mathcal{I}}) - \mathrm{ECE}(f_W, S_{\mathrm{tr}})| \leq \frac{\mathrm{KL}(\rho\|\pi) + B\log 2 + \log\frac{1}{\varepsilon} + \frac{2\lambda^2}{n}}{\lambda} \quad \textit{(Binary classification)},$$

$$\mathbb{E}_\rho|\mathrm{TCE}(f_{\mathcal{I}}) - \mathrm{ECE}(f_W, S_{\mathrm{tr}})| \leq \frac{\mathrm{KL}(\rho\|\pi) + B\log 2 + \log\frac{1}{\varepsilon} + \frac{2\lambda^2}{n}}{\lambda} \quad \textit{(Multiclass classification)}.$$

*Furthermore, in addition to the above assumption, when Assumption 3 holds, we have*

$$\mathbb{E}_\rho|\mathrm{CE}(f_{\mathcal{I}}) - \mathrm{ECE}(f_W, S_{\mathrm{tr}})| \leq \frac{\mathrm{KL}(\rho\|\pi) + B\log 2 + \log\frac{1}{\varepsilon} + \frac{\lambda^2}{2n}}{\lambda} \quad \textit{(Binary classification)},$$

$$\mathbb{E}_\rho|\mathrm{TCE}(f_{\mathcal{I}}) - \mathrm{ECE}(f_W, S_{\mathrm{tr}})| \leq \frac{\mathrm{KL}(\rho\|\pi) + B\log 2 + \log\frac{1}{\varepsilon} + \frac{\lambda^2}{2n}}{\lambda} \quad \textit{(Multiclass classification)}.$$

*Proof.* To simplify the notation, we express $S_{\mathrm{tr}}$ as $S$ in this proof.

BINARY SETTING

To derive the PAC Bayes bound, we transform the loss using Eq. (12) and (13), we have

$$
\begin{aligned}
&|\mathrm{CE}(f_{\mathcal{I}}) - \mathrm{ECE}(f_W, S)| \\
&= \left| \sum_{i=1}^{B} \left| \mathbb{E}_{Z'=(X',Y')} \left[ (Y' - f_W(X')) \cdot \mathbb{1}_{f_W(X') \in I_i} \right] \right| - \sum_{i=1}^{B} \left| \frac{1}{n} \sum_{m=1}^{n} (Y_m - f_W(X_m)) \cdot \mathbb{1}_{f_W(X_m) \in I_i} \right| \right| \\
&\leq \sum_{i=1}^{B} \left| \mathbb{E}_{Z'=(X',Y')} \left[ (Y' - f_W(X')) \cdot \mathbb{1}_{f_W(X') \in I_i} \right] - \frac{1}{n} \sum_{m=1}^{n} (Y_m - f_W(X_m)) \cdot \mathbb{1}_{f_W(X_m) \in I_i} \right| \\
&\leq \sum_{i=1}^{B} \left| \mathbb{E}_{Z'} l_i(Z') - \frac{1}{n} \sum_{m=1}^{n} l_i(Z_m) \right|,
\end{aligned}
\tag{19}
$$

where we used the triangle inequality $||a| - |b|| \leq |a - b|$ for the first inequality and set $l_i(z) = (y - f_W(x)) \cdot \mathbb{1}_{f_w(X)_C \in I_i}$. We then evaluate the exponential moment as

$$
\mathbb{E}_{S,\pi} e^{t|\mathrm{CE}(f_{\mathcal{I}}) - \mathrm{ECE}(f_w, S)|} \leq \mathbb{E}_{\pi} \mathbb{E}_S e^{t \sum_{i=1}^{B} \left| \mathbb{E}_{Z'} l_i(Z') - \frac{1}{n} \sum_{m=1}^{n} l_i(Z_m) \right|}.
\tag{20}
$$

By setting $g(i, S) := \mathbb{E}_{Z'} l_i(Z') - \frac{1}{n} \sum_{m=1}^{n} l_i(Z_m)$, we have

$$
\begin{aligned}
\mathbb{E}_S e^{t \sum_{i=1}^{B} |g(i,S)|} &= \mathbb{E}_S \prod_{i=1}^{B} e^{t|g(i,S)|} \\
&\leq \mathbb{E}_S \prod_{i=1}^{B} \left( e^{tg(i,S)} + e^{-tg(i,S)} \right) \\
&\leq \mathbb{E}_S \sum_{v_1,\dots,v_B=0,1} e^{t \sum_{i=1}^{B} (-1)^{v_i} g(i,S)} \\
&= \sum_{v_1,\dots,v_B=0,1} \mathbb{E}_S e^{t \sum_{i=1}^{B} (-1)^{v_i} g(i,S)} \\
&= \sum_{v_1,\dots,v_B=0,1} \mathbb{E}_S e^{t \sum_{i=1}^{B} (-1)^{v_i} \left[ \mathbb{E}_{Z'} l_i(Z') - \frac{1}{n} \sum_{m=1}^{n} l_i(Z_m) \right]},
\end{aligned}
\tag{21}
$$

where $\sum_{v_1,\dots,v_B=0,1}$ is all the combinations that will be generated by expanding $\prod_{i=1}^{B}$ in Eq. (21) and it has $2^B$ combinations.

We would like to upper bound $\mathbb{E}_S e^{t \sum_{i=1}^{B} (-1)^{v_i} \left[ \mathbb{E}_{Z'} l_i(Z') - \frac{1}{n} \sum_{m=1}^{n} l_i(Z_m) \right]}$ using Lemma 1. For that purpose, here we evaluate

$c_i$s of Lemma 1. By focusing on the exponent, we can estimate $c_i$s by

$$\sup_{\{z_m\}_{m=1},\tilde{z}_m\in\mathcal{Z}} \sum_{i=1}^{B} t(-1)^{v_i} \cdot \left[\mathbb{E}_{Z'}l_i(Z') - \frac{1}{n}\sum_{m=1}^{n} l_i(z'_m)\right]$$

$$- t(-1)^{v_i} \cdot \left[\mathbb{E}_{Z'}l_i(Z') - \frac{1}{n}\sum_{m\neq m'}^{n} l_i(z'_m) - \frac{1}{n}l_i(\tilde{z}_{m'})\right]$$

$$= \sup_{z'_m,\tilde{z}_{m'}\in\mathcal{Z}} \sum_{i=1}^{B} \frac{t(-1)^{v_i}}{n} \cdot [-l_i(z'_{m'}) + -l_i(\tilde{z}_{m'})]$$

$$= \sup_{z'_m,\tilde{z}_{m'}\in\mathcal{Z}} \frac{t(-1)^{v_1}}{n} \Big( - \big((y'_{m'} - f_W(x'_{m'}))\cdot\mathbb{1}_{f_W(x'_{m'})\in I_1}\big) + \big((\tilde{y}_{m'} - f_W(\tilde{x}_{m'}))\cdot\mathbb{1}_{f_W(\tilde{x}_{m'})\in I_1}\big)\Big) +$$

$$\vdots$$

$$+ \frac{t(-1)^{v_B}}{n} \cdot \Big( - \big((y'_{m'} - f_W(x'_{m'}))\cdot\mathbb{1}_{f_W(x'_{m'})\in I_B}\big) + \big((\tilde{y}_{m'} - f_W(\tilde{x}_{m'}))\cdot\mathbb{1}_{f_W(\tilde{x}_{m'})\in I_B}\big)\Big) \tag{22}$$

$$\leq \frac{2t}{n}, \tag{23}$$

The last inequality is derived as follows: By definition of the binning, each data point is assigned to exactly one bin. Consequently, for the input $x'_{m'}$, one indicator from the set $\{\mathbb{1}_{f_W(x'_{m'})\in I_i}\}_{i=1}^{B}$ is non-zero, we denoted the corresponding index $b$. Thus, $\mathbb{1}_{f_w(x'_{m'})\in I_b} \neq 0$, and $\mathbb{1}_{f_w(x'_{m'})\in I_{b'\neq b}} = 0$. A similar discussion applies to the input $\tilde{x}_{m'}$ with the non-zero index denoted as $\tilde{b}$, indicating $\mathbb{1}_{f_W(\tilde{x}_{m'})\in I_{\tilde{b}}} \neq 0$ and $\mathbb{1}_{f_w(x'_{m'})\in I_{b'\neq\tilde{b}}} = 0$. Note that $b$ and $\tilde{b}$ can be either identical or different. Therefore, although Eq. (22) contains $2B$ indicator functions, at most only two are non-zero.

Combined with the fact that $|y'_{m'} - f_w(x'_{m'})| \leq 1$, we obtain Eq. (23). Note that by Assumption 3, $\{f_w(x_m)\}_{m=1}^{n}$ in $x_m \in S$ takes the distinct values almost surely and in the above discussion, we do not consider the case when $b/B = f_w(x_m)$ for some $b$ holds, which means that the predicted probability is just the value of the boundary of bins.

Combined with Lemma 1, we have that

$$\mathbb{E}_S e^{t\sum_{i=1}^{B}|g(i,Z'_m)|} \leq \sum_{v_1,\ldots,v_B=0,1} \prod_{m=1}^{n} \mathbb{E}_{Z'_m} e^{t\sum_{i=1}^{B}(-1)^{v_i}\left[\mathbb{E}_{Z''}l_i(Z'') - \frac{1}{n}\sum_{m=1}^{n}l_i(Z'_m)\right]}$$

$$\leq \sum_{v_1,\ldots,v_B=0,1} e^{(t^2/8)n\left(\frac{2}{n}\right)^2}$$

$$= 2^B e^{\frac{t^2}{2n}}.$$

When we do not assume that Assumption 3, there may be a possibility that $b/B = f_w(x_m)$ for some $b$ holds, which means that the predicted probability is just the value of the boundary of bins. Then in Eq. (22), at most only four indicator functions are not zero. This results in a worse bound

$$\sup_{\{z_m\}_{m=1},\tilde{z}_m\in\mathcal{Z}} \sum_{i=1}^{B} t(-1)^{v_i} \cdot \left[\mathbb{E}_{Z'}l_i(Z') - \frac{1}{n}\sum_{m=1}^{n} l_i(z'_m)\right]$$

$$- t(-1)^{v_i} \cdot \left[\mathbb{E}_{Z'}l_i(Z') - \frac{1}{n}\sum_{m\neq m'}^{n} l_i(z'_m) - \frac{1}{n}l_i(\tilde{z}_{m'})\right]$$

$$\leq \frac{4t}{n},$$

and results in

$$\mathbb{E}_S e^{t\sum_{i=1}^{B}|g(i,Z'_m)|} \leq \sum_{v_1,\ldots,v_B=0,1} \prod_{m=1}^{n} \mathbb{E}_{Z'_m} e^{t\sum_{i=1}^{B}(-1)^{v_i}\left[\mathbb{E}_{Z''}l_i(Z'') - \frac{1}{n}\sum_{m=1}^{n}l_i(Z'_m)\right]} \leq 2^B e^{\frac{2t^2}{n}}.$$

We define the upper bound of the exponential moment as

$$f(t,n) := \begin{cases} \log 2^B e^{\frac{2t^2}{n}} & \text{without Assumption 3,} \\ \log 2^B e^{\frac{t^2}{2n}} & \text{with Assumption 3.} \end{cases} \tag{24}$$

From Eq. (20), for any $w$ we have

$$\mathbb{E}_S e^{t|\mathrm{CE}(f_{\mathcal{I}}) - \mathrm{ECE}(f_W, S)| - f(t,n)} \le 1. \tag{25}$$

Then we take the expectation with respect to the prior

$$\mathbb{E}_\pi \mathbb{E}_S e^{t|\mathrm{CE}(f_{\mathcal{I}}) - \mathrm{ECE}(f_W, S)| - f(t,n)} \le 1.$$

Using the Fubini theorem, we change the order of expectations,

$$\mathbb{E}_S \mathbb{E}_\pi \left[ e^{t|\mathrm{CE}(f_{\mathcal{I}}) - \mathrm{ECE}(f_W, S)| - f(t,n)} \right] \le 1.$$

and use the Donsker–Varadhan lemma, we have

$$\mathbb{E}_S \left[ e^{\sup_{\rho \in \mathcal{P}(\mathcal{W})} \mathbb{E}_\rho t|\mathrm{CE}(f_{\mathcal{I}}) - \mathrm{ECE}(f_W, S)| - \mathrm{KL}(\rho\|\pi) - f(t,n)} \right] \le 1.$$

where $\mathcal{P}(\mathcal{W})$ be the set of all probability distributions on $\mathcal{W}$.

By using the Markov inequality with Chernoff-bounding technique, we have

$$P\left( \sup_{\rho \in \mathcal{P}(\mathcal{W})} \mathbb{E}_\rho t|\mathrm{CE}(f_{\mathcal{I}}) - \mathrm{ECE}(f_W, S)| - \mathrm{KL}(\rho\|\pi) - f(t,n) > \log \frac{1}{\varepsilon} \right) \le \varepsilon.$$

By rearranging the inequality

$$P\left( \exists \rho \in \mathcal{P}(\mathcal{W}), \mathbb{E}_{w \sim \rho} |\mathrm{CE}(f_{\mathcal{I}}) - \mathrm{ECE}(f_W, S)| > \frac{\mathrm{KL}(\rho\|\pi) + f(t,n) + \log \frac{1}{\varepsilon}}{t} \right) \le \varepsilon. \tag{26}$$

By substituting Eq. (24) when without Assumption 3, we have

$$P\left( \exists \rho \in \mathcal{P}(\mathcal{W}), \mathbb{E}_{w \sim \rho} |\mathrm{CE}(f_{\mathcal{I}}) - \mathrm{ECE}(f_W, S)| > \frac{\mathrm{KL}(\rho\|\pi) + B\log 2 + \frac{2t^2}{n} + \log \frac{1}{\varepsilon}}{t} \right) \le \varepsilon.$$

MULTICLASS SETTING

As for the ECE of the multiclass setting, from Eq. (14) and Eq. (15), we can proceed the proof exactly the same way as the binary setting. By using the triangular inequality similarly to Eq. (19) and setting $l_i(Z) = (\mathbb{1}_{Y=C} - f_w(X)_C) \cdot \mathbb{1}_{f_w(X)_C \in I_i}$, we have

$$|\mathrm{TCE}(f_w) - \mathrm{ECE}(f_w, S_{\mathrm{tr}})| \le \sum_{i=1}^{B} |\mathbb{E}_{Z \sim \mathcal{D}} l_i(Z) - \mathbb{E}_{Z \sim S_{\mathrm{tr}}} l_i(Z)|.$$

By using McDiarmid's inequality to evaluate the exponential moment, we obtain

$$\mathbb{E}_{S,\pi} e^{t|\mathrm{TCE}(f_{\mathcal{I}}) - \mathrm{ECE}(f_W, S)|} \le \mathbb{E}_{S,\pi} e^{\lambda \sum_{i=1}^{B} |\mathbb{E}_{Z \sim \mathcal{D}} l_i(Z) - \mathbb{E}_{Z \sim \hat{S}_{\mathrm{tr}}} l_i(Z)|} \le 2^B e^{\frac{2\lambda^2}{n}}$$

Then by using the Fubini theorem and Donsker-Valadhan inequality as in the case of the binary classification, we get the bound. □

### B.5. Proof of Theorem 1 (Total bias analysis)

Now we present the proof of Theorem 1. The proof is almost identical to that of Theorem 2 shown in Appendix B.3.

To analyze the bias caused by binning, following Futami and Fujisawa (2024), we define the conditional expectation of $f_w$ given bins as $f_{\mathcal{I}}^C(x) := \sum_{i=1}^B \mathbb{E}[f_w(X)_C | f_w(X)_C \in I_i] \cdot \mathbb{1}_{f_w(x)_C \in I_i}$. With this definition, we decompose the estimation bias as follows:

$$|\text{TCE}(f_w) - \text{ECE}(f_w, S_{\text{te}})| \leq |\text{TCE}(f_w) - \text{TCE}(f_{\mathcal{I}}^C)| + |\text{TCE}(f_{\mathcal{I}}^C) - \text{ECE}(f_w, S_{\text{te}})|.$$

where the first term is called as the *binning bias*, which arises from nonparametric estimation via binning, and the latter as the *statistical bias* caused by estimation on finite data points. We then evaluate these terms separately.

As for the binning bias, using Assumption 1, it can be bounded by $(1 + L)/B$ because prepared bins divide the interval $[0, 1]$ into equal widths.

Next, we evaluate the statistical bias. We first derive the following reformulation:

$$\text{ECE}(f_w, S_{\text{te}}) = \sum_{i=1}^B |\mathbb{E}_{(X,Y) \sim S_{\text{te}}}(\mathbb{1}_{Y=C} - f_w(X)_C) \cdot \mathbb{1}_{f_w(X)_C \in I_i}|,$$

where $\mathbb{E}_{S_{\text{te}}}$ is the empirical expectation for $S_{\text{te}}$. This reformulation is derived in Eq. (14). As for the TCE, we get a similar relation from Eq. (15).

By using the triangular inequality similarly to Eq. (19) and setting $l_i(Z) = (\mathbb{1}_{Y=C} - f_w(X)_C) \cdot \mathbb{1}_{f_w(X)_C \in I_i}$, we have

$$|\text{TCE}(f_w) - \text{ECE}(f_w, S_{\text{tr}})| \leq \sum_{i=1}^B |\mathbb{E}_{Z \sim \mathcal{D}} l_i(Z) - \mathbb{E}_{Z \sim S_{\text{te}}} l_i(Z)|. \tag{27}$$

By using McDiarmid's inequality to evaluate the exponential moment of Eq. (27), we obtain

$$\mathbb{E}_{S_{\text{te}}} e^{\lambda \sum_{i=1}^B |\mathbb{E}_{Z \sim \mathcal{D}} l_i(Z) - \mathbb{E}_{Z \sim S_{\text{te}}} l_i(Z)|} \leq 2^B e^{\frac{2\lambda^2}{n_{\text{te}}}}$$

for any $\lambda \in \mathbb{R}$ (see Eq. (24)). Combining this result with the Markov inequality, we can upper bound the statistical bias.

### B.6. Generalization of the ECE under the training dataset

Here we show the generalization error bound for the ECE, which is useful when analyzing the recalibration algorithm.

**Theorem 7.** *Assume that $n_{\text{te}} = n_{\text{tr}} = n$. For both binary and multiclass settings, for any fixed prior $\pi$, which is independent of $S_{\text{tr}}$, and posterior distribution $\rho$ over $W$ and any positive constant $\varepsilon \in (0,1)$ and $\lambda > 0$, with probability $1 - \varepsilon$ with respect to $S_{\text{tr}}$, we have*

$$\mathbb{E}_\rho |\mathbb{E}_{S_{\text{te}}} \text{ECE}(f_W, S_{\text{te}}) - \text{ECE}(f_W, S_{\text{tr}})| \leq \frac{\text{KL}(\rho \| \pi) + B \log 2 + \log \frac{1}{\varepsilon} + \frac{4\lambda^2}{n}}{\lambda}.$$

*When Assumption 3 holds, we have*

$$\mathbb{E}_\rho |\mathbb{E}_{S_{\text{te}}} \text{ECE}(f_W, S_{\text{te}}) - \text{ECE}(f_W, S_{\text{tr}})| \leq \frac{\text{KL}(\rho \| \pi) + B \log 2 + \log \frac{1}{\varepsilon} + \frac{\lambda^2}{n}}{\lambda}.$$

*Proof.* To derive the PAC Bayes bound, we transform the loss

$$|\mathbb{E}_{S_{\text{te}}} \text{ECE}(f_W, S_{\text{te}}) - \text{ECE}(f_W, S_{\text{tr}})|$$

$$= \left| \mathbb{E}_{S_{\text{te}}} \sum_{i=1}^B \left| \frac{1}{n} \sum_{(X,Y) \in S_{\text{te}}} [(Y - f_W(X)) \cdot \mathbb{1}_{f_w(X)_C \in I_i}] \right| - \sum_{i=1}^B \left| \frac{1}{n} \sum_{(X,Y) \in S_{\text{tr}}} (Y - f_W(X)) \cdot \mathbb{1}_{f_w(X)_C \in I_i} \right| \right|$$

$$\leq \mathbb{E}_{S_{\text{te}}} \sum_{i=1}^B \left| \frac{1}{n} \sum_{Z \in S_{\text{te}}} l_i(Z) - \frac{1}{n} \sum_{Z \in S_{\text{tr}}} l_i(Z) \right|,$$

where we used the triangle inequality $||a| - |b|| \leq |a - b|$ for the first inequality and set $l_i(z) = (y - f_W(x)) \cdot \mathbb{1}_{f_w(X)_C \in I_i}$.

We then evaluate the exponential moment as

$$\mathbb{E}_{S_{\mathrm{tr}}, \pi} e^{t|\mathbb{E}_{S_{\mathrm{te}}} \mathrm{ECE}(f_W, S_{\mathrm{te}}) - \mathrm{ECE}(f_W, S_{\mathrm{tr}})|} \leq \mathbb{E}_\pi \mathbb{E}_{S_{\mathrm{tr}}} e^{t\mathbb{E}_{S_{\mathrm{te}}} \sum_{i=1}^B \left| \frac{1}{n} \sum_{Z \in S_{\mathrm{te}}} l_i(Z) - \frac{1}{n} \sum_{Z \in S_{\mathrm{tr}}} l_i(Z) \right|}$$

$$\leq \mathbb{E}_\pi \mathbb{E}_{S_{\mathrm{tr}}} \mathbb{E}_{S_{\mathrm{te}}} e^{t \sum_{i=1}^B \left| \frac{1}{n} \sum_{Z \in S_{\mathrm{te}}} l_i(Z) - \frac{1}{n} \sum_{Z \in S_{\mathrm{tr}}} l_i(Z) \right|}.$$

By setting $g(i, S_{\mathrm{tr}}, S_{\mathrm{te}}) := \frac{1}{n} \sum_{Z \in S_{\mathrm{te}}} l_i(Z) - \frac{1}{n} \sum_{Z \in S_{\mathrm{tr}}} l_i(Z)$, we have

$$\mathbb{E}_{S_{\mathrm{tr}}, S_{\mathrm{te}}} e^{t \sum_{i=1}^B |g(i, S_{\mathrm{tr}}, S_{\mathrm{te}})|} = \mathbb{E}_{S_{\mathrm{tr}}, S_{\mathrm{te}}} \prod_{i=1}^B e^{t|g(i, S_{\mathrm{tr}}, S_{\mathrm{te}})|}$$

$$\leq \mathbb{E}_{S_{\mathrm{tr}}, S_{\mathrm{te}}} \prod_{i=1}^B \left( e^{tg(i, S_{\mathrm{tr}}, S_{\mathrm{te}})} + e^{-tg(i, S_{\mathrm{tr}}, S_{\mathrm{te}})} \right)$$

$$\leq \mathbb{E}_{S_{\mathrm{tr}}, S_{\mathrm{te}}} \sum_{v_1, \ldots, v_B = 0, 1} e^{t \sum_{i=1}^B (-1)^{v_i} g(i, S_{\mathrm{tr}}, S_{\mathrm{te}})} \qquad (28)$$

$$= \sum_{v_1, \ldots, v_B = 0, 1} \mathbb{E}_{S_{\mathrm{tr}}, S_{\mathrm{te}}} e^{t \sum_{i=1}^B (-1)^{v_i} g(i, S_{\mathrm{tr}}, S_{\mathrm{te}})}$$

$$= \sum_{v_1, \ldots, v_B = 0, 1} \mathbb{E}_{S_{\mathrm{tr}}, S_{\mathrm{te}}} e^{t \sum_{i=1}^B (-1)^{v_i} \left[ \frac{1}{n} \sum_{Z \in S_{\mathrm{te}}} l_i(Z) - \frac{1}{n} \sum_{Z \in S_{\mathrm{tr}}} l_i(Z) \right]},$$

where $\sum_{v_1, \ldots, v_B = 0, 1}$ is all the combinations that will be generated by expanding $\prod_{i=1}^B$ in Eq. (28) and it has $2^B$ combinations.

We would like to upper bound $\mathbb{E}_{S_{\mathrm{tr}}, S_{\mathrm{te}}} e^{t \sum_{i=1}^B (-1)^{v_i} \left[ \frac{1}{n} \sum_{Z \in S_{\mathrm{te}}} l_i(Z) - \frac{1}{n} \sum_{Z \in S_{\mathrm{tr}}} l_i(Z) \right]}$ using Lemma 1. For that purpose, here we evaluate $c_i$s of Lemma 1. We can upper bound it completely in the same way as Eq. (23). For that purpose, we define $S = S_{\mathrm{te}} \cup S_{\mathrm{tr}}$, $\tilde{g}(S) := t \sum_{i=1}^B (-1)^{v_i} \left[ \frac{1}{n} \sum_{Z \in S_{\mathrm{te}}} l_i(Z) - \frac{1}{n} \sum_{Z \in S_{\mathrm{tr}}} l_i(Z) \right]$. Note that $S$ **consists of** $2n$ **random variables**. We remark that $\mathbb{E}_S[\tilde{g}(S)] = 0$ conditioned on $w$, thus, it satisfies the condition of Lemma 1. Then We define $S'$ in which we replace single $z_m \in S$ with $\tilde{z}_m \in \mathcal{Z}$. Then with Assumption 3, we have

$$\sup_{S, \tilde{z}_m \in \mathcal{Z}} \tilde{g}(S) - \tilde{g}(S') \leq \frac{2t}{n}, \qquad (29)$$

which is followed by a discussion in Eq. (23). Differently from Eq. (23), there exist $4B$ bins, but since we only replace single data point $z_m$, as discussed in Eq. (23), only two indicator function is not zero. This results in the above upper bound.

Combined with Lemma 1, we have that

$$\mathbb{E}_{S_{\mathrm{tr}}} e^{t \sum_{i=1}^B |g(i, Z'_m)|} \leq \sum_{v_1, \ldots, v_B = 0, 1} \mathbb{E}_{S_{\mathrm{tr}}, S_{\mathrm{te}}} e^{\tilde{g}(S)} \leq \sum_{v_1, \ldots, v_B = 0, 1} e^{(t^2/8)(2n)\left(\frac{2}{n}\right)^2} = 2^B e^{\frac{t^2}{n}}.$$

We can derive the upper bound without Assumption 3 in the same way as Eqs. (24) and (29),

$$\sup_{S, \tilde{z}_m \in \mathcal{Z}} \tilde{g}(S) - \tilde{g}(S') \leq \frac{4t}{n},$$

Combined with Lemma 1, we have that

$$\mathbb{E}_{S_{\mathrm{tr}}} e^{t \sum_{i=1}^B |g(i, Z'_m)|} \leq \sum_{v_1, \ldots, v_B = 0, 1} \mathbb{E}_{S_{\mathrm{tr}}, S_{\mathrm{te}}} e^{\tilde{g}(S)} \leq \sum_{v_1, \ldots, v_B = 0, 1} e^{(t^2/8)(2n)\left(\frac{4}{n}\right)^2} = 2^B e^{\frac{4t^2}{n}}.$$

In conclusion, we have

$$f(t, n) := \begin{cases} \log 2^B e^{\frac{4t^2}{n}} & \text{without Assumption 3,} \\ \log 2^B e^{\frac{t^2}{n}} & \text{with Assumption 3.} \end{cases} \qquad (30)$$

Then we can proceed the proof of PAC Bayesian bound in the same way as Eq. (25) and repeat the same derivation. This concludes the proof.

As for the ECE of the multiclass setting, from Eq. (14) and Eq. (15), we can proceed the proof exactly the same way as the binary setting, which results in the same upper bound. □

## B.7. Limitation under the Hölder continuity

Here we discuss the assumption of Lipschitz continuity. It is known that by assuming the higher order smoothness, such as the $\beta$-Hölder continuity, the bias of the nonparametric estimation decreases (Tsybakov, 2008), particularly, the lower bound is $\mathcal{O}(n^{-\frac{\beta}{2\beta+1}})$.

However, in the binning scheme, we cannot improve the optimal order from $\mathcal{O}(n^{-1/3})$ due to the binning bias. This is because that in Eq. (17), which is appearing the proof of the estimation bias, there is the error term $\mathbb{E}|f_w(X) - f_{\mathcal{I}}(X)|$. This term is upper bounded by $1/B$ since we consider that the bins are allocated with equal width. Thus, we cannot improve the estimation bias order owing to this error term, which remains $1/B$ even under the Hölder continuity assumption. Thus, the binning method cannot utilize the smoothness of the underlying data distribution.

## B.8. Proof of Theorem 3 (Bias for the multiclass setting)

The upper bound can be derived similarly to the binary classification setting in Appendix B.3; 1) we first transform the $ECE_K$ and $\mathrm{CE}_K$, 2) decompose the estimation bias to the generalization and binning bias, 3) bound those two terms using the generalization error analysis and the binning definitions.

In this section, we express the L1 distance $\|\cdot\|_1$ of $\mathbb{R}^K$ as $\|\cdot\|$ to simplify the notation.

ECE TRANSFORMATION

Recall the definition

$$\mathrm{ECE}_K(f_w, S_e) := \sum_{i=1}^{B} p_i \|\bar{f}_{i,S_e} - \bar{p}_{i,S_e}\|$$

where $|I_i| := \sum_{m=1}^{n_e} \mathbb{1}_{f_w(x_m) \in I_i}$, $\hat{p}_i := |I_i|/n_e$, $\bar{f}_{i,S_e} := \frac{1}{|I_i|} \sum_{m=1}^{n_e} \mathbb{1}_{f_w(x_m) \in I_i} f_w(x_m)$, and $\bar{p}_{i,S_e} := \frac{1}{|I_i|} \sum_{m=1}^{n_e} \mathbb{1}_{f_w(x_m) \in I_i} \mathbf{e}_{y_m}$.

Using this definition, we first derive the ECE transformation; for the $ECE_K$ under the training dataset, we have

$$\mathrm{ECE}_K(f_w, S_{\mathrm{tr}}) = \sum_{i=1}^{B} \frac{|I_i|}{n} \|\bar{f}_{i,S_{\mathrm{tr}}} - \bar{p}_{i,S_e}\| = \sum_{i=1}^{B} \|\mathbb{E}_{(X,Y)\sim S_{\mathrm{tr}}}(\mathbf{e}_Y - f_w(X)) \cdot \mathbb{1}_{f_w(x_m) \in I_i}\|. \tag{31}$$

Next, we define the conditional expectation as

$$f_{\mathcal{I}}(x) := \sum_{i=1}^{B} f_{I_i}(x) \cdot \mathbb{1}_{f_w(X) \in I_i} = \sum_{i=1}^{B} \mathbb{E}[f_w(X)|f_w(X) \in I_i] \cdot \mathbb{1}_{f_w(X) \in I_i}.$$

Then by repeating the exactly same proof for Eq. (13), the following relation holds

$$\mathrm{CE}_K(f_{\mathcal{I}}) = \sum_{i=1}^{B} \|\mathbb{E}_{(X,Y)\sim\mathcal{D}}(\mathbf{e}_Y - f_w(X)) \cdot \mathbb{1}_{f_w(X) \in I_i}\|. \tag{32}$$

Again, we consider the following decomposition for the estimation bias

$$|\mathrm{CE}_K(f_w) - \mathrm{ECE}_K(f_w, S_{\mathrm{tr}})| \le |\mathrm{CE}_K(f_w) - \mathrm{CE}_K(f_{\mathcal{I}})| + |\mathrm{CE}_K(f_{\mathcal{I}}) - \mathrm{ECE}_K(f_w, S_{\mathrm{tr}})|, \tag{33}$$

We then separately upper bound the above two terms.

GENERALIZATION OF ECE

First we focus on the second term, $|\text{CE}_K(f_{\mathcal{I}}) - \text{ECE}_K(f_w, S_{\text{tr}})|$, and we upper bound it by the PAC Bayes bound;

**Theorem 8.** *Assume that $f_w(X)$ has the probability density over $[0,1]^K$. Then for any fixed prior $\pi$, which is independent of $S_{\text{tr}}$, and posterior distribution $\rho$ over $W$ and any positive constant $\varepsilon \in [0,1]$ and $\lambda > 0$, with probability $1 - \varepsilon$ with respect to $S_{\text{tr}}$, we have*

$$\mathbb{E}_\rho |CE_K(f_{\mathcal{I}}) - \text{ECE}_K(f_W, S_{\text{tr}})| \leq \frac{\text{KL}(\rho\|\pi) + BK\log 2 + \log\frac{1}{\varepsilon} + \frac{K^2\lambda^2}{2n}}{\lambda}.$$

*Proof.* The proof strategy is almost identical to the TCE.

First, we transform the generalization gap by using Eqs. (31) and (32). First note that

$$\text{CE}_K(f_{\mathcal{I}}) = \sum_{i=1}^{B} \|\mathbb{E}_{(X,Y)\sim\mathcal{D}}(\mathbf{e}_Y - f_w(X)) \cdot \mathbb{1}_{f_w(X)\in I_i}\|$$

and

$$\text{ECE}_K(f_w, S_{\text{tr}}) = \sum_{i=1}^{B} \|\mathbb{E}_{(X,Y)\sim\widehat{S}_{\text{tr}}}(\mathbf{e}_Y - f_w(X)) \cdot \mathbb{1}_{f_w(x_m)\in I_i}\|$$

Thus, we have

$$|\text{CE}_K(f_{\mathcal{I}}) - \text{ECE}_K(f_w, S_{\text{tr}})|$$

$$= \left|\sum_{i=1}^{B} \|\mathbb{E}_{(X,Y)\sim\mathcal{D}}(\mathbf{e}_Y - f_w(X)) \cdot \mathbb{1}_{f_w(X)\in I_i}\| - \sum_{i=1}^{B} \|\mathbb{E}_{(X,Y)\sim\widehat{S}_{\text{tr}}}(\mathbf{e}_Y - f_w(X)) \cdot \mathbb{1}_{f_w(x_m)\in I_i}\|\right|$$

$$\leq \sum_{i=1}^{B} \left\|\mathbb{E}_{(X,Y)\sim\mathcal{D}}l_i(Z) - \frac{1}{n}\sum_{Z\in S_{\text{tr}}} l_i(Z)\right\|,$$

where we used the triangle inequality $|\|a\| - \|b\|| \leq \|a - b\|$ for the first inequality and set $l_i(z) = (\mathbf{e}_Y - f_W(x)) \cdot \mathbb{1}_{f_w(X)\in I_i}$. We then evaluate the exponential moment as follows;

$$\mathbb{E}_{S_{\text{tr}},\pi} e^{t|\text{CE}_K(f_{\mathcal{I}}) - \text{ECE}_K(f_w, S_{\text{tr}})|} \leq \mathbb{E}_\pi \mathbb{E}_{S_{\text{tr}}} \mathbb{E}_{S_{\text{te}}} e^{t\sum_{i=1}^{B}\|\mathbb{E}_{(X,Y)\sim\mathcal{D}}l_i(Z) - \frac{1}{n}\sum_{Z\in S_{\text{tr}}} l_i(Z)\|}$$

$$\leq \mathbb{E}_\pi \mathbb{E}_{S_{\text{tr}}} \mathbb{E}_{S_{\text{te}}} e^{t\sum_{i=1}^{B}\sum_{k=1}^{K}|\mathbb{E}_{(X,Y)\sim\mathcal{D}}l_i(Z)_k - \frac{1}{n}\sum_{Z\in S_{\text{tr}}} l_i(Z)_k|}$$

where $l_i(Z)_k$ is the $k$-th dimension of $l_i(z)$.

By setting $g(i, k, S_{\text{tr}}) := \mathbb{E}_{\tilde{Z}}l_i(\tilde{Z})_k - \frac{1}{n}\sum_{Z\in S_{\text{tr}}} l_i(Z)_k$, we have

$$\mathbb{E}_{S_{\text{tr}}} e^{t\sum_{i=1}^{B}\sum_{k=1}^{K}|g(i,k,S_{\text{tr}})|} = \mathbb{E}_{S_{\text{tr}}} \prod_{i=1}^{B}\prod_{k=1}^{K} e^{t|g(i,k,S_{\text{tr}})|}$$

$$\leq \mathbb{E}_{S_{\text{tr}}} \prod_{i=1}^{B}\prod_{k=1}^{K} \left(e^{tg(i,k,S_{\text{tr}})} + e^{-tg(i,k,S_{\text{tr}})}\right)$$

$$\leq \mathbb{E}_{S_{\text{tr}}} \sum_{v_1,\ldots,v_{BK}=0,1} e^{t\sum_{i=1}^{B}\sum_{k=1}^{K}(-1)^{v_i}g(i,k,S_{\text{tr}})}$$

$$= \sum_{v_1,\ldots,v_{BK}=0,1} \mathbb{E}_{S_{\text{tr}}} e^{t\sum_{i=1}^{B}\sum_{k=1}^{K}(-1)^{v_i}g(i,k,S_{\text{tr}})}$$

$$= \sum_{v_1,\ldots,v_{BK}=0,1} \mathbb{E}_{S_{\text{tr}}} e^{t\sum_{i=1}^{B}\sum_{k=1}^{K}(-1)^{v_i}(\mathbb{E}_{\tilde{Z}}l_i(\tilde{Z})_k - \frac{1}{n}\sum_{Z\in S_{\text{tr}}} l_i(Z)_k)}$$

where $\sum_{v_1,\ldots,v_{BK}=0,1}$ is all the combinations that will be generated by expanding $\prod_{i=1}^{BK}$ in Eq. (28) and it has $2^{BK}$ combinations.

We would like to upper bound the exponential moment using Lemma 1. For that purpose, here we evaluate $c_i$s of Lemma 1. We can upper-bound it completely in the same way as the binary case. Then with Assumption that $f_w(x)$ is absolutely continuous, $f_w(x)$ takes the distinct values almost surely. Thus, we have

$$
\sup_{\{z_m\}_{m=1}, \tilde{z}_m \in \mathcal{Z}} \sum_{i=1}^{B} \sum_{k=1}^{K} t(-1)^{v_i} \cdot \left[ \mathbb{E}_{Z'} l_i(Z')_k - \frac{1}{n} \sum_{m=1}^{n} l_i(z'_m)_k \right]
$$
$$
- t(-1)^{v_i} \cdot \left[ \mathbb{E}_{Z'} l_i(Z')_k - \frac{1}{n} \sum_{m \neq m'}^{n} l_i(z'_m)_k - \frac{1}{n} l_i(\tilde{z}_{m'})_k \right]
$$
$$
= \sup_{z'_m, \tilde{z}_{m'} \in \mathcal{Z}} \sum_{i=1}^{B} \sum_{k=1}^{K} \frac{t(-1)^{v_i}}{n} \cdot [-l_i(z'_{m'})_k + l_i(\tilde{z}_{m'})_k]
$$
$$
= \sup_{z'_m, \tilde{z}_{m'} \in \mathcal{Z}} \sum_{k=1}^{K} \frac{t(-1)^{v_1}}{n} \cdot \Big( -\big(((\mathbf{e}_Y)'_{m',k} - f_W(x'_{m'})_k) \cdot \mathbb{1}_{f_W(x'_{m'}) \in I_1}\big) + \big((\hat{\tilde{y}}'_{m',k} - f_W(\tilde{x}'_{m'})_k) \cdot \mathbb{1}_{f_W(\tilde{x}'_{m'}) \in I_1}\big) \Big) +
$$
$$
\vdots
$$
$$
+ \frac{t(-1)^{v_B}}{n} \cdot \Big( -\big(((\mathbf{e}_Y)'_{m',k} - f_W(x'_{m'})_k) \cdot \mathbb{1}_{f_W(x'_{m'}) \in I_B}\big) + \big((\hat{\tilde{y}}_{m',k} - f_W(\tilde{x}_{m'})_k) \cdot \mathbb{1}_{f_W(\tilde{x}_{m'}) \in I_B}\big) \Big) \tag{34}
$$
$$
\leq \sum_{k=1}^{K} \frac{2t}{n} \leq \frac{2Kt}{n},
$$

which is followed by a discussion in Eq. (23). Combined with Lemma 1, we have that

$$
\mathbb{E}_{S_{\mathrm{tr}}} e^{t \sum_{i=1}^{B} \sum_{k=1}^{K} |g(i, S_{\mathrm{tr}})|} \leq \sum_{v_1,\ldots,v_{BK}=0,1} e^{(t^2/8)(n)\left(\frac{2K}{n}\right)^2} = 2^{BK} e^{\frac{K^2 t^2}{2n}}. \tag{35}
$$

Thus, the exponential moment is upper-bounded and we define

$$
f(t, n) := 2^{BK} e^{\frac{K^2 t^2}{2n}}
$$

We then repeat the Chernoff-bounding technique of the proof of the PAC-Bayes bound derivation, and substitute $f(t, n)$ into Eq. (26), we have the following PAC-Bayes bound; for any prior $\pi$ and posterior distribution $\rho$ over $W$ and any positive constant $\varepsilon \in [0, 1]$ and $\lambda > 0$, with probability $1 - \varepsilon$ with respect to $S_{\mathrm{tr}}$, we have

$$
\mathbb{E}_\rho |CE_K(f_{\mathcal{I}}) - \mathrm{ECE}_K(f_w, S_{\mathrm{tr}})| \leq \frac{\mathrm{KL}(\rho \| \pi) + BK \log 2 + \log \frac{1}{\varepsilon} + \frac{K^2 \lambda^2}{2n}}{\lambda}. \tag{36}
$$

$\square$

BIAS ANALYSIS OF THE BINNING METHOD

Next, we upper bound $|\mathrm{CE}_K(f_w) - \mathrm{CE}_K(f_{\mathcal{I}})|$, which is the first term of Eq. (33). By repeating exactly the same procedure in Eq. (16), we can show that

$$
|\mathrm{CE}_K(f_w) - \mathrm{CE}_K(f_{\mathcal{I}})| \leq \mathbb{E}\|\mathbb{E}[\mathbf{e}_Y | f_w(X)] - \mathbb{E}[\mathbf{e}_Y | f_{\mathcal{I}}(X)]\| + \mathbb{E}\|f_w(X) - f_{\mathcal{I}}(X)\|.
$$

We can bound them using the definition of the binning construction; recall that in the current binning construction, we split each dimension of $[0, 1]^K$ with $B'$ bins with equal width. Thus $|f_w(X)_k - f_{\mathcal{I}}(X)_k| \leq 1/B'$ for all $k \in [K]$. This implies that

$$
\mathbb{E}\|f_w(X) - f_{\mathcal{I}}(X)\| \leq \mathbb{E}\Big(\sum_{k=1}^{K} |f_w(X)_k - f_{\mathcal{I}}(X)_k|\Big) \leq \sum_{k=1}^{K} \frac{1}{B'} = \frac{K}{B'}
$$

Then we use the fact that $B = (B')^K$, we have that

$$\mathbb{E}\|f_w(X) - f_{\mathcal{I}}(X)\| \leq \frac{K}{B^{\frac{1}{K}}}$$

Using the Lipschitz continuity for the first, we have

$$|\mathrm{CE}_K(f_w) - \mathrm{CE}_K(f_{\mathcal{I}})| \leq \mathbb{E}|\|\mathbb{E}[\mathbf{e}_Y|f_w(X)] - \mathbb{E}[\mathbf{e}_Y|f_{\mathcal{I}}(X)]\| + \mathbb{E}|f_w(X) - f_{\mathcal{I}}(X)|$$

$$\leq \frac{KL}{B'} + \frac{K}{B'} = \frac{K(1+L)}{B^{\frac{1}{K}}}. \tag{37}$$

CONCLUSION

Finally, taking the expectation by $\rho$ in Eq. (33), we upper bound the first and second term using Eq. (37) and Eq. (36), we obtain the theorem.

As for the total bias

$$\mathrm{Bias}(f_w, S_{\mathrm{te}}, \mathrm{CE}_K) := |\mathrm{CE}_K(f_w) - \mathrm{ECE}_K(f_w, S_{\mathrm{te}})|$$

We can derive this using the above generalization bound for $\mathbb{E}_\rho|CE_K(f_{\mathcal{I}}) - \mathrm{ECE}_K(f_w, S_{\mathrm{tr}})|$ similarly to the derivation in Appendix B.5. The result is simply dropping the KL term in Eq. (36) and obtain the order $\mathrm{Bias}(f_w, S_{\mathrm{te}}, \mathrm{CE}_K) = \mathcal{O}(1/n_{\mathrm{te}}^{1/(K+2)})$ under $B = \mathcal{O}(n_{\mathrm{te}}^{\frac{1}{K+2}})$.

### B.9. Discussion about the assumption of Theorem 3

In Theorem 3, we assumed that $f_w(x)$ has the probability density in $[0,1]^K$. In the binary classification and TCE analysis, this assumption is not required, and if we assume this, our bound results in the improvement of the coefficient, not affect the order of the bound.

On the other hand, when considering the generalization of $ECE_K$, this assumption is inevitable to circumvent the curse of dimensionality especially in the generalization error analysis. Here we discuss how Theorem 3 changes when eliminating this assumption. If we do not assume this assumption, in the proof of Theorem 3, the exponential moment is replaced to

$$\mathbb{E}_{S_{\mathrm{tr}}} e^{t \sum_{i=1}^B \sum_{k=1}^K |g(i,S_{\mathrm{tr}})|} \leq \sum_{v_1,\dots,v_{BK}=0,1} e^{(t^2/8)(n)\left(\frac{2K(2^K)}{n}\right)^2} = 2^{BK} e^{\frac{4^K K^2 t^2}{2n}},$$

which is the surprisingly worse upper bound compared with the upper bound of the exponential moment shown in Eq. (35) with the assumption. Then the obtained PAC Bayesian bound becomes

$$\mathbb{E}_\rho|CE_K(f_{\mathcal{I}}) - \mathrm{ECE}_K(f_W, S_{\mathrm{tr}})| \leq \frac{\mathrm{KL}(\rho\|\pi) + BK \log 2 + \log\frac{1}{\varepsilon} + \frac{4^K K^2 \lambda^2}{2n}}{\lambda}.$$

which has the new coefficient $4^K$ and it is significantly larger than that of Theorem 3.

This is caused by the evaluation of the coefficient of the Lemma 1. In the derivation of Eq. (34), if we do not assume that $f_w$ has the probability density, then there is a possibility that the training data is allocated to the point at the exact grid points of the hypercube constructed by the bins. The grid point is the intersection of $2^K$ small regions, and thus, the $c_i$s of Lemma 1 becomes exponentially large. Owing to this, the coefficient of Eq. (34) is significantly larger compared with the setting when we assume that $f_w$ has the probability density, where we do not need to consider such a worst case. In this way, the order of the bias significantly improves by assuming that $f_w$ has the probability density.

### B.10. Discussion about the selected labels for the ECE

Here we consider the setting that we only focuses on $K' \in [1,K]$ classes. Without loss of generality, we focus on the set of labels defined as $\{0,\dots,K'-1\}$. We then define $f'_w(x) = (f_w(x)_1,\dots,f_w(x)_{K'})$. We also regard $P(Y = 0|f'_w(X)),\dots,P(Y = K'-1|f'_w(X))$ as the $K'$ dimensional vector given $w$ and $x$. We also define the one-hot vector $\mathbf{e}'_Y := ((\mathbf{e}_Y)_1,\dots,(\mathbf{e}_Y)_{K'})$ where $(\mathbf{e}_Y)_k$ is the $k$-the element of $\mathbf{e}_Y$.

Under these notations, we can define

$$CE_{K'}(f_w) := \mathbb{E}\|P(Y|f'_w(X)) - f'_w(X)\|_1,$$

where $\|\cdot\|_p$ is the $L^p$ distance in $\mathbb{R}^{K'}$. The $CE_{K'}$ metric measures the calibration performance of models for only $K'$ classes.

Let us consider the following binning scheme to estimate $CE_{K'}$. Since we want to estimate the $K'$-dimensional conditional probability in $[0,1]^{K'}$ by binning, we split each dimension $[0,1]$ into $B'$ bins of the same width. After doing so, there are $B = (B')^{K'}$ bins, and $[0,1]^{K'}$ is split into $B$ small regions. We refer to these regions as $\mathcal{I} := \{I_i\}_{i=1}^B$. In this case, the ECE of $CE_K$ can be defined as

$$\mathrm{ECE}_{K'}(f_w, S_e) := \sum_{i=1}^B p_i \|\bar{f}'_{i,S_e} - \bar{p}'_{i,S_e}\|_1$$

where $|I_i| := \sum_{m=1}^{n_e} \mathbb{1}_{f'_w(x_m) \in I_i}$, $\hat{p}_i := |I_i|/n_e$, $\bar{f}'_{i,S_e} := \frac{1}{|I_i|}\sum_{m=1}^{n_e} \mathbb{1}_{f'_w(x_m) \in I_i} f'_w(x_m)$, and $\bar{p}_{i,S_e} := \frac{1}{|I_i|}\sum_{m=1}^{n_e} \mathbb{1}_{f'_w(x_m) \in I_i} \mathbf{e}'_{y_m}$.

For these definitions, we can derive the following bias

$$\mathrm{Bias}(f_w, S_{\mathrm{te}}, CE_{K'}) := |CE_{K'}(f_w) - \mathrm{ECE}_{K'}(f_w, S_{\mathrm{te}})|$$

and this can be analyzed exactly in the same way as Appendix B.8. Then we have $\mathrm{Bias}(f_w, S_{\mathrm{te}}, CE_{K'}) = \mathcal{O}(1/n_{\mathrm{te}}^{1/(K'+2)})$ under $B = \mathcal{O}(n_{\mathrm{te}}^{\frac{1}{K'+2}})$. We can obtain the similar generalization bound.

# C. Proofs and discussion of Section 4

We remark that recalibration takes $\eta_V : \Delta^K \to \Delta^K$, and this is clear for the multiclass setting since $f_w : \mathcal{X} \to \Delta^K$. In the case of the binary setting, $f_w : \mathcal{X} \to [0,1]$. So, we consider that the input to $\eta_v$ is $(f_w(x), 1 - f_w(x)) \in \Delta^2$. Then we can treat the recalibration in a unified way.

## C.1. Proofs for Corollary 1 and 2

When we recalibrate the model, we fix $w$. Thus under the $f_w$, the given recalibration data $S_{\mathrm{re}} = \{(X_m, Y_m)\}_{m=1}^{n_{\mathrm{re}}}$, it is transformed into $\tilde{S}_{\mathrm{re}} = \{(f_w(X_m), Y_m)\}_{m=1}^{n_{\mathrm{re}}}$ and samples in $\tilde{S}_{\mathrm{re}}$ are i.i.d by definition. Then we apply our developed PAC-Bayes bounds in Appendix B for this new dataset.

As for Corollary 1, we can prove it exactly in the same way as shown in Appendix B.6. Simply replacing $S_{\mathrm{tr}}$ with $\tilde{S}_{\mathrm{re}}$ in Appendix B.6 and apply it to the standard derivation of PAC-Byaeisan bound shown in Eq. (25). This concludes the proof.

As for Corollary 2, we first provide the complete statement;

**Corollary 3.** *Assume that $n_{\mathrm{te}} = n_{\mathrm{re}} = n$. Under Assumption 2, conditioned on $W = w$, for any fixed prior $\tilde{\pi}$, which is independent of $S_{\mathrm{re}}$, and posterior distribution $\tilde{\rho}$ over $V$ and any positive constant $\varepsilon \in [0,1]$ and $\lambda > 0$, with probability $1 - \varepsilon$ with respect to $S_{\mathrm{re}}$, we have*

$$\mathbb{E}_{\tilde{\rho}}\mathrm{Bias}(\eta_v \circ f_w, S_{\mathrm{re}}, \mathrm{CE}) \leq \frac{1+L}{B} + \frac{\mathrm{KL}(\tilde{\rho}\|\tilde{\pi}) + B\log 2 + \log\frac{1}{\varepsilon} + \frac{2\lambda^2}{n}}{\lambda},$$

$$\mathbb{E}_{\tilde{\rho}}\mathrm{Bias}(\eta_v \circ f_w, S_{\mathrm{re}}, \mathrm{TCE}) \leq \frac{1+L}{B} + \frac{\mathrm{KL}(\tilde{\rho}\|\tilde{\pi}) + B\log 2 + \log\frac{1}{\varepsilon} + \frac{2\lambda^2}{n}}{\lambda}.$$

We can prove this exactly in the same way as shown in Appendix B.3. Simply replacing $S_{\mathrm{tr}}$ with $S_{\mathrm{re}}$ in Appendix B.3. This concludes the proof.

## C.2. Discussion about the conditional KL divergence

In Corollary 1 and 2, we fixed $w$. If we take the expectation with respect to $W$, then the bound of Corollary 1 become as follows;

**Corollary 4.** *Assume that $n_{\text{te}} = n_{\text{re}} = n$. Under Assumption 2, conditioned on $S_{\text{tr}} = s_{\text{tr}}$, for any fixed prior $\tilde{\pi}$, which is independent of $S_{\text{re}}$, and posterior distribution $\tilde{\rho}$ over $V$ and any positive constant $\varepsilon \in [0, 1]$ and $\lambda > 0$, with probability $1 - \varepsilon$ with respect to $S_{\text{re}}$, we have*

$$\mathbb{E}_{\rho(W|s_{\text{tr}})}\mathbb{E}_{\tilde{\rho}}\text{gen}(\eta_v \circ f_w, S_{\text{re}}, \text{ECE}) \leq \frac{\mathbb{E}_{\rho(w|s_{\text{tr}})}[\text{KL}(\tilde{\rho}\|\tilde{\pi})] + B\log 2 + \log\frac{1}{\varepsilon} + \frac{4\lambda^2}{n}}{\lambda}.$$

*where $\mathbb{E}_{\rho(w|s_{\text{tr}})}[\text{KL}(\tilde{\rho}\|\tilde{\pi})]$ represents the conditional KL divergence.*

*Proof.* The proof is almost the same as Appendix B.6. The difference is how we utilize the Chernoff-bounding technique. Here we only show the different parts. After obtaining the upper bound of the exponential moment $f(t, n_{\text{re}})$ as in Eq. (30), we substitute this into Eq. (25), which can be written as follows; for any $w$ and $v$ we have

$$\mathbb{E}_{S_{\text{re}}}e^{t|\text{ECE}(\eta_v \circ f_w, S_{\text{te}}) - \text{ECE}(\eta_v \circ f_w, S_{\text{re}})| - f(t, n_{\text{re}})} \leq 1.$$

Then we take the expectation with respect to the prior $\tilde{\pi}$ and $\rho(w|s_{\text{tr}})$,

$$\mathbb{E}_{\rho(w|s_{\text{tr}})}\mathbb{E}_{\tilde{\pi}}\mathbb{E}_{S_{\text{re}}}e^{t|\text{ECE}(\eta_v \circ f_w, S_{\text{te}}) - \text{ECE}(\eta_v \circ f_w, S_{\text{re}})| - f(t, n_{\text{re}})} \leq 1.$$

Using the Fubini theorem, we change the order of expectations,

$$\mathbb{E}_{\rho(w|s_{\text{tr}})}\mathbb{E}_{S_{\text{re}}}\mathbb{E}_{\tilde{\pi}}\left[e^{t|\text{ECE}(\eta_v \circ f_w, S_{\text{te}}) - \text{ECE}(\eta_v \circ f_w, S_{\text{re}})| - f(t, n_{\text{re}})}\right] \leq 1,$$

and $\rho(w|s_{\text{tr}})$ and $S_{\text{re}}$ are independent thus

$$\mathbb{E}_{S_{\text{re}}}\mathbb{E}_{\rho(w|s_{\text{tr}})}\mathbb{E}_{\tilde{\pi}}\left[e^{t|\text{ECE}(\eta_v \circ f_w, S_{\text{te}}) - \text{ECE}(\eta_v \circ f_w, S_{\text{re}})| - f(t, n_{\text{re}})}\right] \leq 1.$$

Then, use the Donsker–Varadhan lemma, we have

$$\mathbb{E}_{S_{\text{re}}}\mathbb{E}_{\rho(w|s_{\text{tr}})}\left[e^{\sup_{\tilde{\rho}\in\mathcal{P}(\mathcal{W})}\mathbb{E}_{\tilde{\rho}}t|\text{ECE}(\eta_v \circ f_w, S_{\text{te}}) - \text{ECE}(\eta_v \circ f_w, S_{\text{re}})| - \text{KL}(\tilde{\rho}\|\tilde{\pi}) - f(t, n_{\text{re}})}\right] \leq 1.$$

By taking the Jensen inequality, we have

$$\mathbb{E}_{S_{\text{re}}}\left[e^{\mathbb{E}_{\rho(w|S_{\text{tr}})}\sup_{\tilde{\rho}\in\mathcal{P}(\mathcal{W})}\mathbb{E}_{\tilde{\rho}}t|\text{ECE}(\eta_v \circ f_w, S_{\text{te}}) - \text{ECE}(\eta_v \circ f_w, S_{\text{re}})| - \text{KL}(\tilde{\rho}\|\tilde{\pi}) - f(t, n_{\text{re}})}\right]$$
$$\leq \mathbb{E}_{S_{\text{re}}}\mathbb{E}_{\rho(w|S_{\text{tr}})}\left[e^{\sup_{\tilde{\rho}\in\mathcal{P}(\mathcal{W})}\mathbb{E}_{\tilde{\rho}}t|\text{ECE}(\eta_v \circ f_w, S_{\text{te}}) - \text{ECE}(\eta_v \circ f_w, S_{\text{re}})| - \text{KL}(\tilde{\rho}\|\tilde{\pi}) - f(t, n_{\text{re}})}\right] \leq 1.$$

By taking the swap about $\mathbb{E}_{\rho(w|S_{\text{tr}})}$ and sup, we have

$$\mathbb{E}_{S_{\text{re}}}\left[e^{\sup_{\tilde{\rho}\in\mathcal{P}(\mathcal{W})}\mathbb{E}_{\rho(w|S_{\text{tr}})}\mathbb{E}_{\tilde{\rho}}t|\text{ECE}(\eta_v \circ f_w, S_{\text{te}}) - \text{ECE}(\eta_v \circ f_w, S_{\text{re}})| - \text{KL}(\tilde{\rho}\|\tilde{\pi}) - f(t, n_{\text{re}})}\right]$$
$$\leq \mathbb{E}_{S_{\text{re}}}\left[e^{\mathbb{E}_{\rho(w|S_{\text{tr}})}\sup_{\tilde{\rho}\in\mathcal{P}(\mathcal{W})}\mathbb{E}_{\tilde{\rho}}t|\text{ECE}(\eta_v \circ f_w, S_{\text{te}}) - \text{ECE}(\eta_v \circ f_w, S_{\text{re}})| - \text{KL}(\tilde{\rho}\|\tilde{\pi}) - f(t, n_{\text{re}})}\right] \leq 1.$$

After this, we simply follow the derivation of the standard PAC-Bayes derivation in Appendix B.3. $\square$

### C.3. Discussion for Section 4.2

First, the upper bound of the ECE is obtained by

$$\text{ECE}(\eta_v \circ f_w, S_{\text{re}}) \leq \frac{1}{n_{\text{re}}}\sum_{(X,Y)\in S_{\text{re}}}^{n_{\text{re}}}|(\mathbf{e}_Y)_C - \eta_V \circ f_w(X)_C|^2 \leq \frac{1}{n_{\text{re}}}\sum_{(X,Y)\in S_{\text{re}}}^{n_{\text{re}}}\|\mathbf{e}_Y - \eta_V \circ f_w(X)\|_2^2$$

where $(\mathbf{e}_Y)_C$ is the $C$-th dimension of the one-hot encoding of the label $Y$ and the first inequality is followed by Hölder inequality and the Jensen inequality. As for the binary classification, by Hölder inequality and the Jensen inequality, we have

$$\text{ECE}(\eta_v \circ f_w, S_{\text{re}}) \leq \frac{1}{n_{\text{re}}}\sum_{(X,Y)\in S_{\text{re}}}^{n_{\text{re}}}|Y - \eta_V \circ f_w(X)_1|^2,$$

since the recalibrated function is $\eta_V \circ f_w(X) \in \Delta^2$ and its 1st dimension represents the predicted probability of the label is 1.

Then from the estimation bias bound in Eq. (7), we have

$$
\mathbb{E}_{\tilde{\rho}} \mathrm{TCE}(\eta_v \circ f_w)
$$

$$
\leq \mathbb{E}_{\tilde{\rho}} \mathrm{ECE}(\eta_v \circ f_w, S_{\mathrm{re}}) + \frac{1+L}{B} + \frac{\mathrm{KL}(\tilde{\rho}\|\tilde{\pi}) + B \log 2 + \log \frac{1}{\varepsilon} + \frac{4\lambda^2}{n_{\mathrm{re}}}}{\lambda}
$$

$$
\leq \mathbb{E}_{\tilde{\rho}} \frac{1}{n_{\mathrm{re}}} \sum_{(X,Y)\in S_{\mathrm{re}}}^{n_{\mathrm{re}}} \|\mathbf{e}_Y - \eta_V \circ f_w(X)\|_2^2 + \frac{1+L}{B} + \frac{\mathrm{KL}(\tilde{\rho}\|\tilde{\pi}) + B \log 2 + \log \frac{1}{\varepsilon} + \frac{4\lambda^2}{n_{\mathrm{re}}}}{\lambda}, \tag{38}
$$

holds for any $\lambda > 0$. Thus, the objective function of Eq. (8) is the upper bound of the TCE in the PAC-Bayes bound. Clearly, a similar statement holds for $CE$ in the binary setting.

Next, we discuss the objective function of Eq. (10). This objective function can guarantee the performance of test accuracy and TCE simultaneously as follows; Assume that $l_{\mathrm{acc}} : \mathcal{Y} \times \mathcal{Y} \to [0,1]$ and define the training loss as $\hat{L}_{\mathrm{acc}}(V, S_{\mathrm{tr}}) := \frac{1}{n_{\mathrm{tr}}} \sum_{(X,Y)\in S_{\mathrm{tr}}}^{n_{\mathrm{tr}}} l_{\mathrm{acc}}(Y, \eta_V \circ f_w(X))$ and the (expected) test loss as $L_{\mathrm{acc}}(v, \mathcal{D}) := \mathbb{E}_Z l_{\mathrm{acc}}(Y, \eta_V \circ f_w(X))$. Under this setting, we have

**Corollary 5.** *Assume that $n_{\mathrm{te}} = n_{\mathrm{re}} = n$ and . Under Assumption 2, conditioned on $W = w$, for any fixed prior $\tilde{\pi}$, which is independent of $S_{\mathrm{re}}$, and posterior distribution $\tilde{\rho}$ over $V$ and any positive constant $\varepsilon \in [0,1]$ and $\lambda > 0$, with probability $1 - \varepsilon$ with respect to $S_{\mathrm{re}}$, we have*

$$
\mathbb{E}_{\tilde{\rho}}(L_{\mathrm{acc}}(V, \mathcal{D}) + \mathrm{TCE}(\eta_v \circ f_w)) \leq \mathbb{E}_{\tilde{\rho}} \frac{1}{n_{\mathrm{re}}} \sum_{(X,Y)\in S_{\mathrm{re}}}^{n_{\mathrm{re}}} (l_{\mathrm{acc}}(Y, \eta_V \circ f_w(X)) + \|\mathbf{e}_Y - \eta_V \circ f_w(X)\|_2^2)
$$

$$
+ \frac{1+L}{B} + \frac{2\mathrm{KL}(\tilde{\rho}\|\tilde{\pi}) + B \log 2 + \frac{33\lambda^2}{8n_{\mathrm{re}}} + 2 \log \frac{2}{\varepsilon}}{\lambda},
$$

*Proof.* From Eq. (11), which is the standard PAC-Bayes bound for $[0,1]$ bounded loss function, we have

$$
\mathbb{E}_{\tilde{\rho}} L_{\mathrm{acc}}(V, \mathcal{D}) \leq \mathbb{E}_{\tilde{\rho}} \hat{L}_{\mathrm{acc}}(V, S_{\mathrm{tr}}) + \frac{\mathrm{KL}(\tilde{\rho}\|\tilde{\pi}) + \log \frac{1}{\varepsilon} + \frac{\lambda^2}{8n_{\mathrm{re}}}}{\lambda}. \tag{39}
$$

By considering the union bound with Eq. (39) and Eq. (38), setting $\varepsilon$ as $\varepsilon/2$ in each high probability bound, we obtain the result. $\square$

Clearly, a similar statement holds for $CE$ in the binary setting.

Thus, the objective function Eq. (10) optimizes the upper bound of Corollary 5, and thus, it naturally guarantees the test accuracy and TCE simultaneously.

*Table 2.* Datasets used in our experiments

| Dataset | Classes | Train data ($n_{\mathrm{tr}}$) | Recalibration data ($n_{\mathrm{re}}$) | Test data ($n_{\mathrm{te}}$) |
|---|---|---|---|---|
| KITTI (Geiger, 2012) | 2 | 16000 | 1000 | 8000 |
| PCam (Veeling et al., 2018) | 2 | 22768 | 1000 | 9000 |
| MNIST (LeCun et al., 1989) | 10 | 60000 | 1000 | 9000 |
| CIFAR-100 (Krizhevsky, 2009) | 100 | 50000 | 1000 | 9000 |

## D. Details of experimental settings

In this section, we summarize the detail information of our numerical experiments in Section 6. Our CIFAR-100 experiments were conducted on NVIDIA GPUs with 32GB memory (NVIDIA DGX-1 with Tesla V100 and DGX-2). For the other experiments, we used CPU (Apple M1) with 16GB memory.

**Datasets and models:** We show the details of the datasets and the numbers of training, recalibration, and test data in Table 2. For the models, we used XGBoost (Chen and Guestrin, 2016), Random Forests (Breiman, 2001), and a 1-layer neural network (NN) for the KITTI, PCam, and MNIST experiments. For the CIFAR-100 experiments, we used AlexNet (Krizhevsky et al., 2012), WideResNet (Zagoruyko and Komodakis, 2016), DenseNet-BC-190 (Huang et al., 2017), and ResNeXt-29 (Xie et al., 2017). XGBoost, Random Forests, and the 1-layer NN were trained by adapting the code from Wenger et al. (2020) [1]. Models used in the CIFAR-100 experiments were obtained from the GitHub project page: `pytorch-classification` (https://github.com/bearpaw/pytorch-classification).

**Recalibration algorithms:** In the multiclass classification experiments, the standard method of temperature scaling (Guo et al., 2017) and the recently developed GP calibration (Wenger et al., 2020) that has shown good performance were used as baselines. As a baseline method for binary classification, we additionally adopted Platt scaling (Platt, 1999), isotonic regression (Zadrozny and Elkan, 2002), Beta calibration (Kull et al., 2017), and Bayesian binning into quantiles (BBQ) (Naeini et al., 2015). We also adopted the GP calibration with mean approximation (Wenger et al., 2020) in the additional experiments in Appendix E.

Wenger et al. (2020) reported results for Platt scaling (Platt, 1999), isotonic regression (Zadrozny and Elkan, 2002), Beta calibration (Kull et al., 2017), and Bayesian binning into quantiles (BBQ) (Naeini et al., 2015), which were originally developed for binary classification tasks and adapted to the multiclass setting via a one-vs-all approach. However, for a fair comparison, our multiclass classification experiments employed temperature scaling and GP-based recalibration as baseline methods, as these approaches can be directly applied to the multiclass setting.

For our PBR method, we set the Gaussian distribution as both the posterior and prior, as in Wenger et al. (2020). We also followed the settings for training strategy, hyperparameters, and optimizers in Wenger et al. (2020) to ensure a fair comparison. PBR has an optimizable parameter $1/\lambda$ in addition to the posterior parameter; thus, we selected it using grid search from the following candidates: $\{0., 0.01, 0.1, 0.25, 0.5, 0.75, 0.9, 1.0\}$. The mean approximation strategy in Wenger et al. (2020) can be applied for PBR. Thus, we reported the results of PBR with mean approximation (denoted as *PBR app.*) in Appendix E.

**Performance evaluation:** We measured the predictive accuracy and the ECE estimated by binning, with $B = \lfloor n_{\mathrm{re}}^{1/3} \rfloor$, following our theoretical findings in Corollary 2. We conducted 10-fold cross-validation for recalibration function training and reported the mean and standard deviation of these two performance metrics. We also used $J = 100$ Monte Carlo samples from posterior $\tilde{\rho}$ to obtain $V$.

## E. Additional experimental results

Here are additional experimental results that could not be included in the main part of this paper.

We first show all the experimental results regarding the correlation between the KL-divergence and our bound in Figures 3 and 4. We observed that the KL-divergence and generalization gap are well correlated in terms of Pearson's and Kendall's

---

[1] https://github.com/JonathanWenger/pycalib

rank-correlation coefficients, especially in the multiclass classification tasks (MNIST and CIFAR-100). However, for the binary classification experiments, no strong correlation was observed. We conjecture that this is caused by the estimation bias of ECE through binning, which has a very slow convergence rate of $\mathcal{O}(n_{\mathrm{re}}^{1/3})$ under our optimal $B$. Since we use $n_{\mathrm{re}} = 1000$, the noise due to the estimation bias can be $\mathcal{O}(1/10)$, which is larger than the ECE gap values in the binary classification experiments. Therefore, it seems that the correlation is buried in the noise.

We also show the results of comparisons with all baseline methods in Tables 3-6. These results confirm that our method almost consistently improves GP-based recalibration methods.

Finally, we evaluate the effect of using Brier and cross-entropy losses in our recalibration approach defined in Eq. (9). Specifically, we reran the experiments under the same setup described in Appendix D, and re-evaluated our two methods from Eqs. (8) and (9) in terms of ECE, classification accuracy, and cross-entropy loss.

The results are summarized in Tables 7–10. Our first key observation is that both of our methods—PBR and PBR total—consistently improve ECE when minimizing the Brier score. This is expected, as the Brier score upper-bounds ECE. We also observe improvements in accuracy and cross-entropy in some settings, particularly for binary classification tasks on relatively simple datasets such as KITTI and PCam.

However, when recalibration is performed using only the Brier score (with KL regularization), PBR can lead to degradation in both cross-entropy and accuracy, especially for more complex datasets and multi-class tasks. This trend is evident in Table 9 (excluding the XGBoost and Random Forest settings) and Table 10 (excluding AlexNet), and is particularly pronounced in experiments involving deep neural networks.

One possible explanation is as follows: in all experiments, the base models $f_w$ were trained using the cross-entropy loss. When recalibration is subsequently performed using only the Brier score, the optimization objective diverges from the training objective. This mismatch can propagate through the GP-based recalibration process, introducing inconsistencies that ultimately degrade performance in terms of accuracy and cross-entropy. Notably, this degradation is alleviated by incorporating the cross-entropy loss into the recalibration objective, as done in PBR total. This suggests that including cross-entropy helps reduce the misalignment between training and recalibration losses.

In summary, minimizing the Brier score alone does not necessarily improve cross-entropy or accuracy and may in fact worsen them. However, incorporating cross-entropy into the recalibration objective offers a more balanced trade-off, improving calibration while preserving predictive performance.

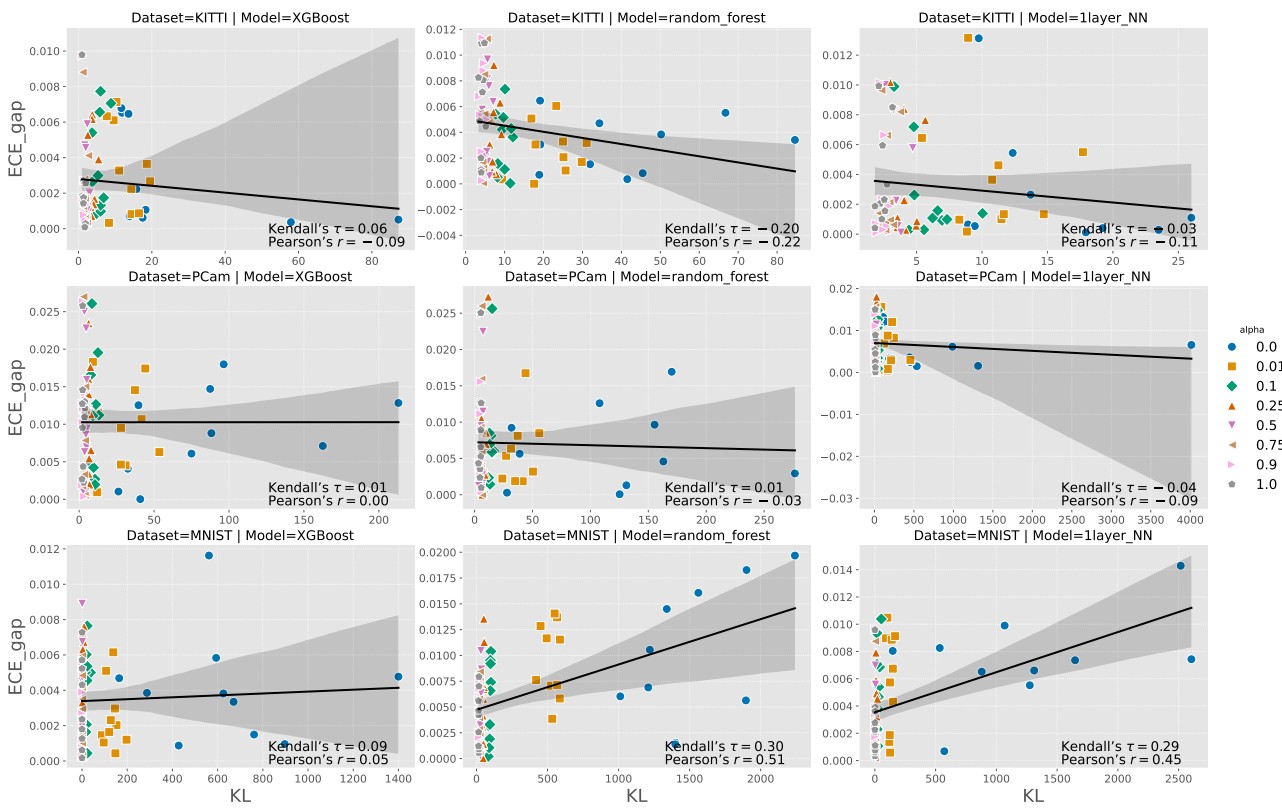

*Figure 3.* KL vs ECE gap on KITTI (binary), PCam (binary), and MNIST (Multiclass) dataset per various regularize parameters ($\alpha$).

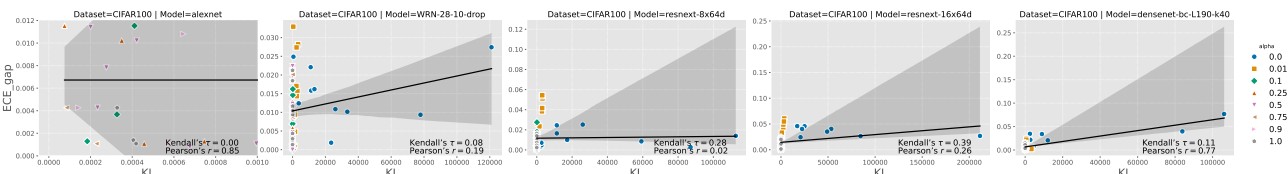

*Figure 4.* KL vs ECE gap on CIFAR-100 (Multiclass) dataset per various regularize parameters ($\alpha$). For the AlexNet experiments, the x-axis range has been limited in the figure to better illustrate the data due to the observation of very large KL values for the case where $\alpha = 0$.

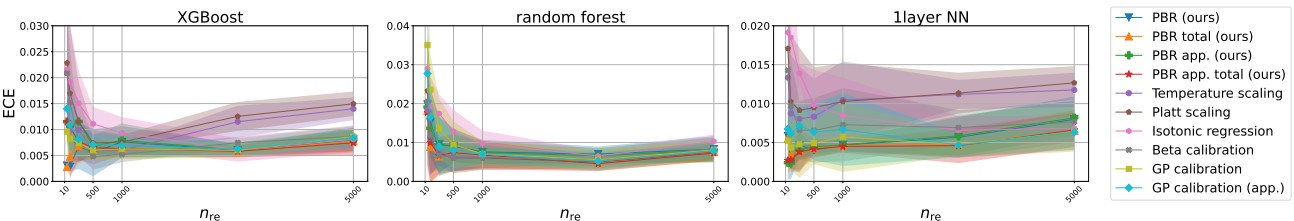

*Figure 5.* $n_{\mathrm{re}}$ vs. ECE on the test dataset for experiments with the KITTI dataset. For details on our methods, we refer to Appendix D.

*Table 3.* KITTI (binary)

| Model | Method | ECE (mean ± std) | Accuracy (mean ± std) |
|---|---|---|---|
| XGBoost | Uncalibration | $0.0113 \pm 0.0004$ | $0.9627 \pm 0.0011$ |
| | GP calibration (Wenger et al., 2020) | $0.0063 \pm 0.0020$ | $0.9633 \pm 0.0007$ |
| | GP calibration (app.) (Wenger et al., 2020) | $0.0079 \pm 0.0033$ | $0.9640 \pm 0.0013$ |
| | PBR (ours) | $0.0076 \pm 0.0030$ | $0.9641 \pm 0.0014$ |
| | PBR total (ours) | $0.0063 \pm 0.0014$ | $0.9636 \pm 0.0013$ |
| | PBR app. (ours) | $0.0079 \pm 0.0033$ | $0.9640 \pm 0.0013$ |
| | PBR app. total (ours) | $0.0063 \pm 0.0012$ | $0.9637 \pm 0.0013$ |
| | Temperature scaling (Guo et al., 2017) | $0.0060 \pm 0.0009$ | $0.9627 \pm 0.0011$ |
| | Platt scaling (Platt, 1999) | $0.0076 \pm 0.0014$ | $0.9621 \pm 0.0011$ |
| | Isotonic regression (Zadrozny and Elkan, 2002) | $0.0093 \pm 0.0030$ | $0.9624 \pm 0.0022$ |
| | Beta calibration (Kull et al., 2017) | $\mathbf{0.0052 \pm 0.0015}$ | $0.9627 \pm 0.0008$ |
| | BBQ (Naeini et al., 2015) | $0.0341 \pm 0.0745$ | $0.9611 \pm 0.0021$ |
| Random Forest | Uncalibration | $0.0736 \pm 0.0014$ | $0.9638 \pm 0.0015$ |
| | GP calibration (Wenger et al., 2020) | $0.0070 \pm 0.0032$ | $0.9628 \pm 0.0018$ |
| | GP calibration (app.) (Wenger et al., 2020) | $0.0074 \pm 0.0029$ | $0.9626 \pm 0.0018$ |
| | PBR (ours) | $0.0076 \pm 0.0031$ | $0.9626 \pm 0.0019$ |
| | PBR total (ours) | $0.0068 \pm 0.0033$ | $0.9628 \pm 0.0018$ |
| | PBR app. (ours) | $0.0074 \pm 0.0029$ | $0.9626 \pm 0.0018$ |
| | PBR app. total (ours) | $0.0068 \pm 0.0037$ | $0.9628 \pm 0.0017$ |
| | Temperature scaling (Guo et al., 2017) | $0.0063 \pm 0.0029$ | $0.9638 \pm 0.0015$ |
| | Platt scaling (Platt, 1999) | $\mathbf{0.0060 \pm 0.0031}$ | $0.9631 \pm 0.0017$ |
| | Isotonic regression (Zadrozny and Elkan, 2002) | $0.0097 \pm 0.0033$ | $0.9612 \pm 0.0024$ |
| | Beta calibration (Kull et al., 2017) | $0.0063 \pm 0.0023$ | $0.9627 \pm 0.0015$ |
| | BBQ (Naeini et al., 2015) | $0.1148 \pm 0.0538$ | $0.8948 \pm 0.0026$ |
| 1-Layer NN | Uncalibration | $0.0049 \pm 0.0008$ | $0.9619 \pm 0.0014$ |
| | GP calibration (Wenger et al., 2020) | $0.0047 \pm 0.0008$ | $0.9619 \pm 0.0014$ |
| | GP calibration (app.) (Wenger et al., 2020) | $0.0047 \pm 0.0008$ | $0.9620 \pm 0.0014$ |
| | PBR (ours) | $0.0047 \pm 0.0008$ | $0.9619 \pm 0.0013$ |
| | PBR total (ours) | $0.0046 \pm 0.0007$ | $0.9621 \pm 0.0014$ |
| | PBR app. (ours) | $0.0047 \pm 0.0008$ | $0.9619 \pm 0.0014$ |
| | PBR app. total (ours) | $\mathbf{0.0045 \pm 0.0008}$ | $0.9620 \pm 0.0014$ |
| | Temperature scaling (Guo et al., 2017) | $0.0104 \pm 0.0050$ | $0.9618 \pm 0.0014$ |
| | Platt scaling (Platt, 1999) | $0.0103 \pm 0.0049$ | $0.9642 \pm 0.0016$ |
| | Isotonic regression (Zadrozny and Elkan, 2002) | $0.0085 \pm 0.0049$ | $0.9672 \pm 0.0024$ |
| | Beta calibration (Kull et al., 2017) | $0.0073 \pm 0.0047$ | $0.9672 \pm 0.0013$ |
| | BBQ (Naeini et al., 2015) | $0.0324 \pm 0.0760$ | $0.9648 \pm 0.0051$ |

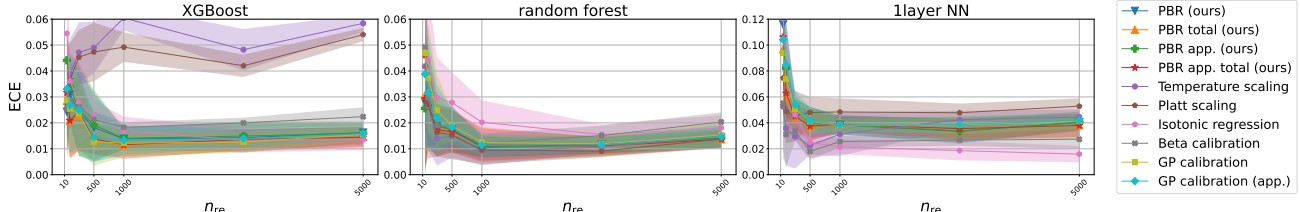

*Figure 6.* $n_{\mathrm{re}}$ vs. ECE on the test dataset for experiments with the PCam dataset. For details on our methods, we refer to Appendix D.

Table 4. PCam (binary)

| Model | Method | ECE (mean ± std) | Accuracy (mean ± std) |
|---|---|---|---|
| XGBoost | Uncalibration | $0.0519 \pm 0.0015$ | $0.8445 \pm 0.0015$ |
| | GP calibration (Wenger et al., 2020) | $0.0122 \pm 0.0053$ | $0.8466 \pm 0.0018$ |
| | GP calibration (app.) (Wenger et al., 2020) | $0.0133 \pm 0.0057$ | $0.8468 \pm 0.0021$ |
| | PBR (ours) | $0.0139 \pm 0.0043$ | $0.8445 \pm 0.0030$ |
| | PBR total (ours) | $\mathbf{0.0113 \pm 0.0043}$ | $0.8464 \pm 0.0024$ |
| | PBR app. (ours) | $0.0140 \pm 0.0042$ | $0.8441 \pm 0.0028$ |
| | PBR app. total (ours) | $0.0116 \pm 0.0055$ | $0.8449 \pm 0.0028$ |
| | Temperature scaling (Guo et al., 2017) | $0.0605 \pm 0.0048$ | $0.8445 \pm 0.0015$ |
| | Platt scaling (Platt, 1999) | $0.0492 \pm 0.0057$ | $0.8474 \pm 0.0011$ |
| | Isotonic regression (Zadrozny and Elkan, 2002) | $0.0170 \pm 0.0053$ | $0.8442 \pm 0.0031$ |
| | Beta calibration (Kull et al., 2017) | $0.0183 \pm 0.0019$ | $0.8475 \pm 0.0012$ |
| | BBQ (Naeini et al., 2015) | $0.0143 \pm 0.0098$ | $0.8412 \pm 0.0065$ |
| Random Forest | Uncalibration | $0.0816 \pm 0.0014$ | $0.8495 \pm 0.0018$ |
| | GP calibration (Wenger et al., 2020) | $0.0124 \pm 0.0027$ | $0.8492 \pm 0.0019$ |
| | GP calibration (app.) (Wenger et al., 2020) | $0.0115 \pm 0.0041$ | $0.8491 \pm 0.0019$ |
| | PBR (ours) | $0.0109 \pm 0.0040$ | $0.8490 \pm 0.0020$ |
| | PBR total (ours) | $0.0116 \pm 0.0068$ | $0.8490 \pm 0.0019$ |
| | PBR app. (ours) | $0.0104 \pm 0.0038$ | $0.8491 \pm 0.0020$ |
| | PBR app. total (ours) | $0.0116 \pm 0.0063$ | $0.8489 \pm 0.0019$ |
| | Temperature scaling (Guo et al., 2017) | $\mathbf{0.0092 \pm 0.0056}$ | $0.8495 \pm 0.0018$ |
| | Platt scaling (Platt, 1999) | $0.0096 \pm 0.0054$ | $0.8496 \pm 0.0014$ |
| | Isotonic regression (Zadrozny and Elkan, 2002) | $0.0202 \pm 0.0085$ | $0.8464 \pm 0.0038$ |
| | Beta calibration (Kull et al., 2017) | $0.0122 \pm 0.0036$ | $0.8488 \pm 0.0024$ |
| | BBQ (Naeini et al., 2015) | $0.0529 \pm 0.0079$ | $0.8063 \pm 0.0082$ |
| 1-Layer NN | Uncalibration | $0.2060 \pm 0.0014$ | $0.5928 \pm 0.0016$ |
| | GP calibration (Wenger et al., 2020) | $0.0385 \pm 0.0073$ | $0.6510 \pm 0.0021$ |
| | GP calibration (app.) (Wenger et al., 2020) | $0.0389 \pm 0.0057$ | $0.6507 \pm 0.0023$ |
| | PBR (ours) | $0.0385 \pm 0.0073$ | $0.6507 \pm 0.0020$ |
| | PBR total (ours) | $0.0384 \pm 0.0060$ | $0.6497 \pm 0.0034$ |
| | PBR app. (ours) | $0.0385 \pm 0.0073$ | $0.6507 \pm 0.0021$ |
| | PBR app. total (ours) | $0.0389 \pm 0.0057$ | $0.6507 \pm 0.0023$ |
| | Temperature scaling (Guo et al., 2017) | $0.0312 \pm 0.0169$ | $0.5928 \pm 0.0016$ |
| | Platt scaling (Platt, 1999) | $0.0484 \pm 0.0105$ | $0.6224 \pm 0.0068$ |
| | Isotonic regression (Zadrozny and Elkan, 2002) | $0.0214 \pm 0.0055$ | $0.6483 \pm 0.0026$ |
| | Beta calibration (Kull et al., 2017) | $0.0256 \pm 0.0044$ | $0.6478 \pm 0.0026$ |
| | BBQ (Naeini et al., 2015) | $\mathbf{0.0179 \pm 0.0029}$ | $0.6467 \pm 0.0039$ |

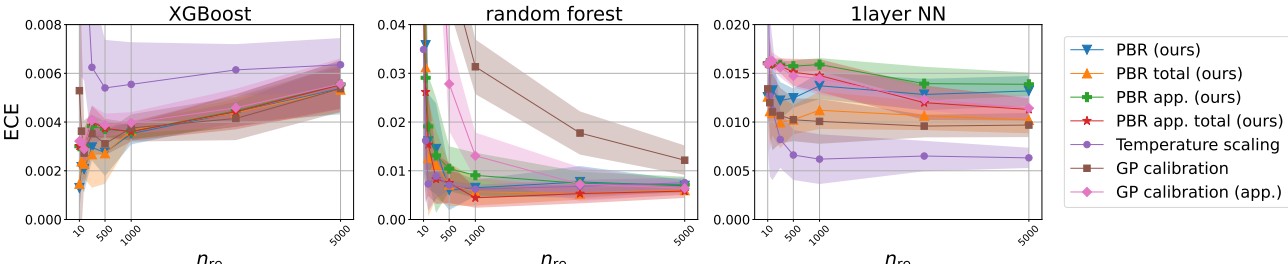

Figure 7. $n_{\mathrm{re}}$ vs. ECE on the test dataset for experiments with the MNIST dataset. For details on our methods, we refer to Appendix D.

*Table 5.* MNIST (Multiclass)

| Model | Method | ECE (mean ± std) | Accuracy (mean ± std) |
|---|---|---|---|
| XGBoost | Uncalibration | $0.0036 \pm 0.0003$ | $0.9785 \pm 0.0005$ |
| | GP calibration (Wenger et al., 2020) | $0.0038 \pm 0.0006$ | $0.9786 \pm 0.0007$ |
| | GP calibration (app.) (Wenger et al., 2020) | $0.0040 \pm 0.0004$ | $0.9785 \pm 0.0007$ |
| | PBR (ours) | $\mathbf{0.0035 \pm 0.0004}$ | $0.9785 \pm 0.0007$ |
| | PBR total (ours) | $0.0037 \pm 0.0004$ | $0.9785 \pm 0.0006$ |
| | PBR app. (ours) | $0.0036 \pm 0.0003$ | $0.9785 \pm 0.0005$ |
| | PBR app. total (ours) | $0.0036 \pm 0.0004$ | $0.9785 \pm 0.0005$ |
| | Temperature scaling (Guo et al., 2017) | $0.0055 \pm 0.0017$ | $0.9785 \pm 0.0005$ |
| Random Forest | Uncalibration | $0.1428 \pm 0.0007$ | $0.9659 \pm 0.0005$ |
| | GP calibration (Wenger et al., 2020) | $0.0313 \pm 0.0056$ | $0.9663 \pm 0.0011$ |
| | GP calibration (app.) (Wenger et al., 2020) | $0.0091 \pm 0.0046$ | $0.9641 \pm 0.0016$ |
| | PBR (ours) | $0.0065 \pm 0.0017$ | $0.9639 \pm 0.0018$ |
| | PBR total (ours) | $0.0057 \pm 0.0028$ | $0.9656 \pm 0.0017$ |
| | PBR app. (ours) | $0.0091 \pm 0.0046$ | $0.9641 \pm 0.0016$ |
| | PBR app. total (ours) | $\mathbf{0.0045 \pm 0.0020}$ | $0.9662 \pm 0.0016$ |
| | Temperature scaling (Guo et al., 2017) | $0.0062 \pm 0.0009$ | $0.9659 \pm 0.0005$ |
| 1-Layer NN | Uncalibration | $0.0164 \pm 0.0004$ | $0.9760 \pm 0.0005$ |
| | GP calibration (Wenger et al., 2020) | $0.0101 \pm 0.0023$ | $0.9740 \pm 0.0008$ |
| | GP calibration (app.) (Wenger et al., 2020) | $0.0159 \pm 0.0007$ | $0.9760 \pm 0.0006$ |
| | PBR (ours) | $0.0137 \pm 0.0014$ | $0.9751 \pm 0.0009$ |
| | PBR total (ours) | $0.0112 \pm 0.0019$ | $0.9740 \pm 0.0016$ |
| | PBR app. (ours) | $0.0159 \pm 0.0007$ | $0.9760 \pm 0.0006$ |
| | PBR app. total (ours) | $0.0148 \pm 0.0017$ | $0.9742 \pm 0.0015$ |
| | Temperature scaling (Guo et al., 2017) | $\mathbf{0.0062 \pm 0.0026}$ | $0.9760 \pm 0.0005$ |

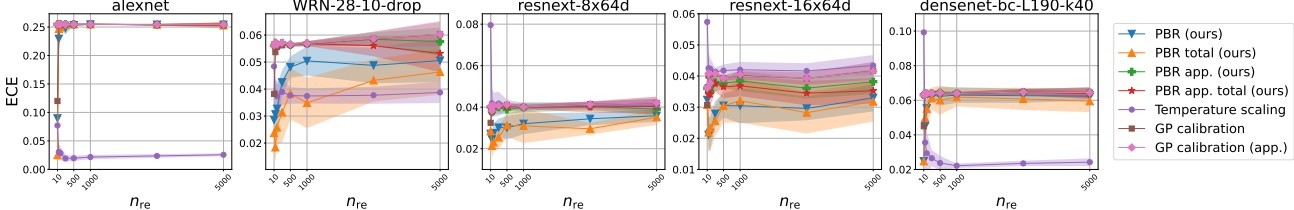

*Figure 8.* $n_{\mathrm{re}}$ vs. ECE on the test dataset for experiments with the CIFAR-100 dataset. For details on our methods, we refer to Appendix D.

Table 6. CIFAR100 (Multiclass)

| Model | Method | ECE (mean ± std) | Accuracy (mean ± std) |
|---|---|---|---|
| alexnet | Uncalibration | $0.2548 \pm 0.0007$ | $0.4374 \pm 0.0008$ |
| | GP calibration (Wenger et al., 2020) | $0.2548 \pm 0.0008$ | $0.4373 \pm 0.0009$ |
| | GP calibration (app.) (Wenger et al., 2020) | $0.2548 \pm 0.0007$ | $0.4374 \pm 0.0008$ |
| | PBR (ours) | $0.2545 \pm 0.0007$ | $0.4374 \pm 0.0008$ |
| | PBR total (ours) | $0.2548 \pm 0.0008$ | $0.4373 \pm 0.0009$ |
| | PBR app. (ours) | $0.2548 \pm 0.0007$ | $0.4374 \pm 0.0008$ |
| | PBR app. total (ours) | $0.2548 \pm 0.0008$ | $0.4374 \pm 0.0008$ |
| | Temperature scaling (Guo et al., 2017) | $\mathbf{0.0216 \pm 0.0037}$ | $0.4374 \pm 0.0008$ |
| WRN-28-10-drop | Uncalibration | $0.0568 \pm 0.0009$ | $0.8131 \pm 0.0011$ |
| | GP calibration (Wenger et al., 2020) | $0.0567 \pm 0.0010$ | $0.8129 \pm 0.0011$ |
| | GP calibration (app.) (Wenger et al., 2020) | $0.0568 \pm 0.0009$ | $0.8130 \pm 0.0011$ |
| | PBR (ours) | $0.0504 \pm 0.0054$ | $0.7878 \pm 0.0049$ |
| | PBR total (ours) | $\mathbf{0.0349 \pm 0.0093}$ | $0.7965 \pm 0.0064$ |
| | PBR app. (ours) | $0.0568 \pm 0.0009$ | $0.8130 \pm 0.0012$ |
| | PBR app. total (ours) | $0.0568 \pm 0.0009$ | $0.8130 \pm 0.0011$ |
| | Temperature scaling (Guo et al., 2017) | $0.0374 \pm 0.0028$ | $0.8131 \pm 0.0011$ |
| resnext-8x64d | Uncalibration | $0.0401 \pm 0.0010$ | $0.8229 \pm 0.0013$ |
| | GP calibration (Wenger et al., 2020) | $0.0400 \pm 0.0010$ | $0.8229 \pm 0.0014$ |
| | GP calibration (app.) (Wenger et al., 2020) | $0.0400 \pm 0.0009$ | $0.8229 \pm 0.0013$ |
| | PBR (ours) | $0.0320 \pm 0.0050$ | $0.8023 \pm 0.0084$ |
| | PBR total (ours) | $\mathbf{0.0310 \pm 0.0082}$ | $0.8123 \pm 0.0080$ |
| | PBR app. (ours) | $0.0399 \pm 0.0024$ | $0.8167 \pm 0.0047$ |
| | PBR app. total (ours) | $0.0401 \pm 0.0010$ | $0.8229 \pm 0.0014$ |
| | Temperature scaling (Guo et al., 2017) | $0.0401 \pm 0.0019$ | $0.8229 \pm 0.0013$ |
| resnext-16x64d | Uncalibration | $0.0405 \pm 0.0012$ | $0.8231 \pm 0.0013$ |
| | GP calibration (Wenger et al., 2020) | $0.0403 \pm 0.0013$ | $0.8230 \pm 0.0014$ |
| | GP calibration (app.) (Wenger et al., 2020) | $0.0405 \pm 0.0014$ | $0.8230 \pm 0.0014$ |
| | PBR (ours) | $\mathbf{0.0305 \pm 0.0056}$ | $0.8014 \pm 0.0065$ |
| | PBR total (ours) | $0.0321 \pm 0.0071$ | $0.8123 \pm 0.0065$ |
| | PBR app. (ours) | $0.0384 \pm 0.0032$ | $0.8118 \pm 0.0068$ |
| | PBR app. total (ours) | $0.0321 \pm 0.0071$ | $0.8123 \pm 0.0065$ |
| | Temperature scaling (Guo et al., 2017) | $0.0420 \pm 0.0025$ | $0.8231 \pm 0.0013$ |
| densenet-bc-L190-k40 | Uncalibration | $0.0639 \pm 0.0008$ | $0.8230 \pm 0.0007$ |
| | GP calibration (Wenger et al., 2020) | $0.0639 \pm 0.0008$ | $0.8229 \pm 0.0007$ |
| | GP calibration (app.) (Wenger et al., 2020) | $0.0639 \pm 0.0008$ | $0.8230 \pm 0.0007$ |
| | PBR (ours) | $0.0630 \pm 0.0045$ | $0.8041 \pm 0.0143$ |
| | PBR total (ours) | $0.0618 \pm 0.0054$ | $0.8195 \pm 0.0078$ |
| | PBR app. (ours) | $0.0639 \pm 0.0008$ | $0.8230 \pm 0.0007$ |
| | PBR app. total (ours) | $0.0639 \pm 0.0008$ | $0.8230 \pm 0.0007$ |
| | Temperature scaling (Guo et al., 2017) | $\mathbf{0.0222 \pm 0.0013}$ | $0.8230 \pm 0.0007$ |

*Table 7.* ECE, accuracy, and cross entropy under our PBR (Eq. (8)) and PBR total (Eq. (10)) on KITTI. Bold font indicates the best result among PBR and PBR total for each metric. If the best values are numerically identical, no bolding is applied. In case of a tie in the mean, the result with a smaller standard deviation is considered better.

| Model | Method | ECE ($\downarrow$) | Accuracy ($\uparrow$) | Cross entropy ($\downarrow$) |
|---|---|---|---|---|
| XGBoost | Uncalibration | $0.011 \pm 0.000$ | $0.963 \pm 0.001$ | $0.110 \pm 0.002$ |
| | PBR (ours) | $0.008 \pm 0.003$ | $0.964 \pm 0.001$ | $0.104 \pm 0.002$ |
| | PBR total (ours) | $\mathbf{0.006 \pm 0.001}$ | $0.964 \pm 0.001$ | $\mathbf{0.104 \pm 0.001}$ |
| Random Forest | Uncalibration | $0.074 \pm 0.001$ | $0.964 \pm 0.001$ | $0.160 \pm 0.003$ |
| | PBR (ours) | $0.008 \pm 0.003$ | $0.963 \pm 0.002$ | $0.114 \pm 0.004$ |
| | PBR total (ours) | $\mathbf{0.007 \pm 0.003}$ | $0.963 \pm 0.002$ | $\mathbf{0.113 \pm 0.005}$ |
| 1-Layer NN | Uncalibration | $0.005 \pm 0.001$ | $0.962 \pm 0.001$ | $0.122 \pm 0.004$ |
| | PBR (ours) | $0.005 \pm 0.001$ | $0.962 \pm 0.001$ | $0.122 \pm 0.004$ |
| | PBR total (ours) | $0.005 \pm 0.001$ | $0.962 \pm 0.001$ | $0.122 \pm 0.004$ |

*Table 8.* ECE, accuracy, and cross entropy under our PBR (Eq. (8)) and PBR total (Eq. (10)) on PCam. Bold font indicates the best result among PBR and PBR total for each metric. If the best values are numerically identical, no bolding is applied. In case of a tie in the mean, the result with a smaller standard deviation is considered better.

| Model | Method | ECE ($\downarrow$) | Accuracy ($\uparrow$) | Cross entropy ($\downarrow$) |
|---|---|---|---|---|
| XGBoost | Uncalibration | $0.052 \pm 0.002$ | $0.845 \pm 0.002$ | $0.356 \pm 0.004$ |
| | PBR (ours) | $0.014 \pm 0.004$ | $0.844 \pm 0.003$ | $0.338 \pm 0.004$ |
| | PBR total (ours) | $\mathbf{0.011 \pm 0.004}$ | $\mathbf{0.846 \pm 0.002}$ | $\mathbf{0.336 \pm 0.003}$ |
| Random Forest | Uncalibration | $0.082 \pm 0.001$ | $0.850 \pm 0.002$ | $0.382 \pm 0.004$ |
| | PBR (ours) | $\mathbf{0.011 \pm 0.004}$ | $0.849 \pm 0.002$ | $0.339 \pm 0.003$ |
| | PBR total (ours) | $0.012 \pm 0.007$ | $0.849 \pm 0.002$ | $0.339 \pm 0.003$ |
| 1-Layer NN | Uncalibration | $0.206 \pm 0.001$ | $0.593 \pm 0.002$ | $0.866 \pm 0.004$ |
| | PBR (ours) | $0.038 \pm 0.007$ | $\mathbf{0.651 \pm 0.002}$ | $0.663 \pm 0.008$ |
| | PBR total (ours) | $\mathbf{0.038 \pm 0.006}$ | $0.650 \pm 0.003$ | $\mathbf{0.661 \pm 0.004}$ |

*Table 9.* ECE, accuracy, and cross entropy under our PBR (Eq. (8)) and PBR total (Eq. (10)) on MNIST. Bold font indicates the best result among PBR and PBR total for each metric. If the best values are numerically identical, no bolding is applied. In case of a tie in the mean, the result with a smaller standard deviation is considered better.

| Model | Method | ECE ($\downarrow$) | Accuracy ($\uparrow$) | Cross entropy ($\downarrow$) |
|---|---|---|---|---|
| XGBoost | Uncalibration | $0.004 \pm 0.000$ | $0.979 \pm 0.000$ | $0.073 \pm 0.002$ |
| | PBR (ours) | $0.004 \pm 0.000$ | $0.978 \pm 0.001$ | $0.073 \pm 0.002$ |
| | PBR total (ours) | $0.004 \pm 0.000$ | $\mathbf{0.979 \pm 0.001}$ | $0.073 \pm 0.002$ |
| Random Forest | Uncalibration | $0.143 \pm 0.001$ | $0.966 \pm 0.001$ | $0.254 \pm 0.001$ |
| | PBR (ours) | $0.007 \pm 0.002$ | $0.964 \pm 0.002$ | $0.134 \pm 0.009$ |
| | PBR total (ours) | $\mathbf{0.006 \pm 0.003}$ | $\mathbf{0.966 \pm 0.002}$ | $\mathbf{0.120 \pm 0.004}$ |
| 1-Layer NN | Uncalibration | $0.016 \pm 0.000$ | $0.976 \pm 0.001$ | $0.124 \pm 0.005$ |
| | PBR (ours) | $0.014 \pm 0.001$ | $\mathbf{0.975 \pm 0.001}$ | $0.107 \pm 0.005$ |
| | PBR total (ours) | $\mathbf{0.011 \pm 0.002}$ | $0.974 \pm 0.002$ | $\mathbf{0.100 \pm 0.006}$ |

*Table 10.* ECE, accuracy, and cross entropy under our PBR (Eq. (8)) and PBR total (Eq. (10)) on CIFAR100. Bold font indicates the best result among PBR and PBR total for each metric. If the best values are numerically identical, no bolding is applied. In case of a tie in the mean, the result with a smaller standard deviation is considered better.

| Model | Method | ECE ($\downarrow$) | Accuracy ($\uparrow$) | Cross entropy ($\downarrow$) |
|---|---|---|---|---|
| alexnet | Uncalibration | $0.255 \pm 0.001$ | $0.437 \pm 0.001$ | $2.952 \pm 0.009$ |
| | PBR (ours) | $0.255 \pm 0.001$ | $0.437 \pm 0.001$ | $2.952 \pm 0.008$ |
| | PBR total (ours) | $0.255 \pm 0.001$ | $0.437 \pm 0.001$ | $2.952 \pm 0.009$ |
| WRN-28-10-drop | Uncalibration | $0.057 \pm 0.001$ | $0.813 \pm 0.001$ | $0.772 \pm 0.005$ |
| | PBR (ours) | $0.050 \pm 0.005$ | $0.788 \pm 0.005$ | $0.897 \pm 0.030$ |
| | PBR total (ours) | $\mathbf{0.035 \pm 0.009}$ | $\mathbf{0.796 \pm 0.006}$ | $\mathbf{0.818 \pm 0.023}$ |
| resnext-8x64d | Uncalibration | $0.040 \pm 0.001$ | $0.823 \pm 0.001$ | $0.704 \pm 0.006$ |
| | PBR (ours) | $0.032 \pm 0.005$ | $0.802 \pm 0.008$ | $0.800 \pm 0.037$ |
| | PBR total (ours) | $\mathbf{0.031 \pm 0.008}$ | $\mathbf{0.812 \pm 0.008}$ | $\mathbf{0.739 \pm 0.023}$ |
| resnext-16x64d | Uncalibration | $0.040 \pm 0.001$ | $0.823 \pm 0.001$ | $0.709 \pm 0.005$ |
| | PBR (ours) | $\mathbf{0.030 \pm 0.006}$ | $0.801 \pm 0.006$ | $0.804 \pm 0.024$ |
| | PBR total (ours) | $0.032 \pm 0.007$ | $\mathbf{0.812 \pm 0.007}$ | $\mathbf{0.741 \pm 0.018}$ |
| densenet-bc-L190-k40 | Uncalibration | $0.064 \pm 0.001$ | $0.823 \pm 0.001$ | $0.708 \pm 0.007$ |
| | PBR (ours) | $0.063 \pm 0.005$ | $0.804 \pm 0.014$ | $0.799 \pm 0.074$ |
| | PBR total (ours) | $\mathbf{0.062 \pm 0.005}$ | $\mathbf{0.820 \pm 0.008}$ | $\mathbf{0.721 \pm 0.028}$ |

