# OpenReview forum: "PAC-Bayes Analysis for Recalibration in Classification"
_ICML.cc/2025/Conference — ICML 2025 poster_

### Official Review · Reviewer_pWL9 · 2025-03-07

**Overall Recommendation:** 3

**Summary:**

In this paper, the PAC-Bayesian framework is used to analyze recalibration in a multiclass classification setting. Specifically, the bias of estimators of the calibration error is bounded, taking both binning and statistical effects into account. The resulting bounds are used as the basis for new recalibration algorithms, which yield improved performance over state-of-the-art approaches for certain settings.

## update after rebuttal

I thank the authors for their responses. I have updated my evaluation upward accordingly, but the contributions still seem somewhat incremental relative to utami and Fujisawa (2024).

**Claims And Evidence:**

Yes — the theoretical derivations appear sound, and the claimed performance improvements (mixed depending on setting) are not beyond what is actually demonstrated

**Essential References Not Discussed:**

N/A

**Ethical Review Flag:**

Flag this paper for an ethics review.

**Experimental Designs Or Analyses:**

I did not study the experimental results beyond the main paper. A minor point that does not seem fully supported is that the theorems involve a “Gibbs error” (i.e. parameters are drawn randomly from posterior for each use) while the practical algorithm deploys an averaged parameter.

**Methods And Evaluation Criteria:**

Yes

**Other Comments Or Suggestions:**

1. “the expectation of a random variable $X$ as $E_X$”: this notation seems to be actually used with distributions in subscript rather than random variables
2. Line 152: “We expect…” should be with $S_{re}$ rather than $S_{te}$?
3. Theorem 1: $\lambda$ is not defined in theorem statement
4. Eq. (6): Should be $\eta_V$?
5. Line 227, right column: “required” — as the obtained result is only an upper bound, it cannot be used to argue that regularization is required for preventing overfitting
6. Line 309: “Eq. (10)” from appendix is referred to without context
7. Line 294, right column: “Equation. (9)”

**Other Strengths And Weaknesses:**

It appears to me that the key strength of the paper is the proposed algorithm, and the fact that this leads to improvements in certain settings, while a key weakness is the similarity of the theoretical analysis compared to Futami and Fujisawa (2004).

**Questions For Authors:**

1. Can you clarify how the theoretical analysis differs from that of Futami and Fujisawa (2004)? If there are key aspects that I have missed, this may affect my evaluation.
2. It is noted in the paper that the proposed algorithm PBR performs worse than temperature-based approaches for, e.g., settings where the underlying classifier has low accuracy. What may this depend on? As the key contribution of the paper appears to be a potential algorithmic advance, clarifying why and how it is useful may affect evaluation.
3. In line 249 and onwards, the use of focusing on $K’$ classes is discussed. Do these classes need to be identified in advance or is, e.g., top-$K’$ classes okay? (Just curious)

**Relation To Broader Scientific Literature:**

The contributions of this paper give a foundation for many approaches to recalibration, for which theoretical analyses may be lacking. The most closely related work to this is Futami and Fujisawa (2004), for which similar bounds on recalibration error are derived. The key difference appears to be that the present paper derives high-probability PAC-Bayesian bounds, while the work of Futami and Fujisawa obtains CMI-based bounds in expectation. However, the proof techniques are very similar, and obtaining the results of this paper seems to essentially rely on using a different change of measure in Donsker-Varadhan. The concentration inequalities that are used are essentially identical, based on a bounded differences argument [1]. The attractiveness of the PAC-Bayesian approach, as highlighted in this paper, is that it enables explicit optimization, potentially leading to improved algorithms. While the present paper also discusses multi-class classification, this analysis appears very similar to the binary case, as noted in the paper.

[1]: e.g., lines 878-908 are identical with the argument on p. 15 of Futami and Fujisawa without explicit reference. This casts some doubt on the "main contribution [being] the derivation of new concentration inequalities"

**Theoretical Claims:**

I skimmed all proofs and did not notice any issues, but did not go through details beyond App. B.4.

---

> ### Author Rebuttal · Authors · 2025-03-31
>
> We sincerely thank you for your feedback.
>
> ## Experimental Designs Or Analyses
>
> ### Q.1: Regarding the Gibbs error
>
> Step 5 of Algorithm 1 shows that we take the average over $J$ posterior samples. In contrast, the theory defines bias as the expectation over the hypothesis distribution outside the absolute value, while the algorithm takes it inside. By Jensen’s inequality, the inside expectation is smaller, so the theoretical bound—specifically, Corollary 5—still holds for the algorithm.
>
> ## Relation To Broader Scientific Literature
>
> ### Q.2: Regarding the relationship between this study and Futami and Fujisawa (2024)
>
> Please refer to our response to [Reviewer mK1Q’s Q.1](https://openreview.net/forum?id=eJzZryJfri&noteId=4CcGzbpkT0). In short, our main technical contribution is the decomposition of bias into approximation and estimation errors, and the development of a bounded-difference concentration inequality enabling finite-sample ECE bias analysis in multi-class settings. This extends the analytical framework of Futami and Fujisawa (2024) beyond binary classification.
>
> ## Other Comments and Suggestions
>
> ### Q.3: Regarding the notation of expectation
>
> We apologize for the inconsistent use of $\mathbb{E}\_X$ and $\mathbb{E}\_{p(X)}$. This reflects common conventions in different contexts: PAC-Bayes often uses $p(X)$, while bias analysis refers to $X$. We will revise the notation for better consistency and clarity.
>
> ### Q.4: Regarding our explanation around line 152
>
> We revised the explanation as follows:
>   - “The output after recalibration, $\eta_v\circ f_w$, is expected to yield a sufficiently small $\mathrm{ECE}(\eta_v\circ f_w, S_{\mathrm{re}})$. From a generalization perspective, it is important to theoretically investigate conditions under which $\eta_v\circ f_w$ also achieves low $\mathrm{ECE}(\eta_v\circ f_w,S_{\mathrm{te}})$. To this end, we define the following error term, resembling the standard generalization error typically defined via a loss function.”
>
> ### Q.5 and 6: Regarding $\lambda$ in Theorem 1, Eq. (6)
>
> We added the assumption $\lambda > 0$ to Theorem 1 and corrected it to $\eta_{V}$.
>
> ### Q.7: Regarding our explanation around line 227
>
> We apologize for the overstatement and have revised the text as follows:
>   - “This result highlights the importance of KL regularization in the parameter space—similar to the standard PAC-Bayes bound over $S_{\mathrm{tr}}$ (McAllester, 2003; Alquier et al., 2016)—in preventing overfitting and improving generalization.”
>
> ### Q.8: Regarding our explanation around line 309
>
> We revised the explanation as follows:
>   - “Since this objective function is derived from the PAC-Bayes bound for $l_{\textrm{acc}}$ (Theorem 4 in Appendix B) and Corollary 2 via a union bound, it remains within the generalization error bound (see Corollary 5 in Appendix C.3 for details).”
>
> ### Q.9: Regarding our explanation around line 294
>
> This has been corrected. We apologize for any confusion.
>
> ## Questions
>
> ### Q.10: Regarding the difference from Futami and Fujisawa (2024)
>
> Please see our responses under Relation to Broader Scientific Literature and Weaknesses, and [Reviewer mK1Q’s Q.1](https://openreview.net/forum?id=eJzZryJfri&noteId=4CcGzbpkT0) for clarification.
>
> ### Q.11: Regarding the limitation of our PBR
>
> We added the following to the fifth paragraph of Section 6.2: GP-based recalibration methods, including PBR, construct the GP prior using outputs of the trained model $f_w$ as inducing points. If $f_w$ misclassifies training data, the prior may be misaligned with the true distribution, and the posterior will be regularized toward this inappropriate prior, degrading performance.
>
> ### Q.12: Regarding clarification on class identification
>
> We apologize for the lack of clarity. Our intention was to consider the Top-$K'$ classes—those with the highest predicted probabilities—as a natural extension of TCE. Your comment prompted us to reconsider pre-specifying a particular set of $K'$ classes. We found this also theoretically valid and practically relevant when focusing on calibration for specific target classes. This discussion was added to the final paragraph of Section 3.3.
>
> ### References
> - [Futami & Fujisawa, 2024](https://proceedings.neurips.cc/paper_files/paper/2024/file/9961e42624a6c083279303767c73269d-Paper-Conference.pdf)
> - [Alquier et al., 2016](https://jmlr.org/papers/v17/15-290.html)
> - [McAllester, 2003](https://link.springer.com/chapter/10.1007/978-3-540-45167-9_16)

---

> > ### Comment · Reviewer_pWL9 · 2025-04-02
> >
> > Thank you for the response. I have a remaining question regarding the Gibbs error.
> >
> > If I understood correctly, Step 5 means that the _parameter_ $V$ is the average of several samples of the posterior. So, it's not just that the algorithm does $\|E_V[ E_{S_{te}}[ECE(\eta_V ...) - ECE(\eta_V ... )] ] \| $ (i.e., gen as defined in Lines 159-160 with an expectation inside the absolute value). But in fact, it does $\|E_{S_{te}}[ECE(\eta_{E_V[ V]} ...) - ECE(\eta_{E_V[ V ]} ... )] ] \| $, which would not be as straight-forward to relate to the actual bound using Jensen. Or does the relation still follow?
> >
> > I may have misunderstood something -- a clarification would be appreciated. Thank you!

---

> > > ### Author Response · Authors · 2025-04-03
> > >
> > > Dear Reviewer pWL9,
> > >
> > > We apologize for the lack of clarity in our previous explanation regarding this point.
> > > The intended correspondence between our bound and Jensen’s inequality is as follows.
> > >
> > > As stated around line 275 in Section 4.2, the goal of our algorithm is to obtain a recalibration model that achieves a smaller TCE. Therefore, at the beginning of Section 4.2, we should have referred to Corollary 2, which directly addresses TCE, rather than Corollary 1. We have corrected this in the revised version.
> > >
> > > **Corollary 2** provides a bound on the bias between TCE and ECE under the posterior $\tilde{\rho}$ (as defined in the right-hand side of lines 111-113):
> > > - $\mathbb{E}\_{\tilde{\rho}}[\mathrm{Bias}(\eta_v\circ f_w,\mathrm{S}_{\mathrm{re}},\mathrm{TCE})] = \mathbb{E}\_{\tilde{\rho}}[|\mathrm{TCE}(\eta_v \circ f_w) -  \mathrm{ECE}(\eta_v \circ f_w, \mathrm{S}\_{\mathrm{re}})|] \leq \textrm{(KL term)}$.
> > >
> > > Here, the “KL term” refers to the right-hand side of Corollary 2, including constant factors.
> > >
> > > By applying the triangle inequality and noting that **both TCE and ECE are non-negative**, we obtain:
> > > - $\mathbb{E}\_{\tilde{\rho}}[\mathrm{TCE}(\eta_v \circ f_w)] \leq \mathbb{E}\_{\tilde{\rho}}[\mathrm{ECE}(\eta_v \circ f_w, \mathrm{S}\_{\mathrm{re}})] + \text{(KL term)}$.
> > >
> > > This result is also discussed in Corollary 5 in Appendix C.3.
> > >
> > > Now, **using Jensen’s inequality** under the posterior $\tilde{\rho}$, we can **further bound the left-hand side of the above** as follows:
> > > - $\mathrm{TCE}(\mathbb{E}\_{\tilde{\rho}}[\eta_v \circ f_w]) \leq \mathbb{E}\_{\tilde{\rho}}[\mathrm{TCE}(\eta_v \circ f_w)] \leq \mathbb{E}\_{\tilde{\rho}}[\mathrm{ECE}(\eta_v \circ f_w, \mathrm{S}\_{\mathrm{re}})] + \text{(KL term)}$.
> > >
> > > **This chain of inequalities justifies the operation in Step 5 of our algorithm, where we estimate $\mathbb{E}\_{\tilde{\rho}}[\eta_v \circ f_w]$ using $J$ i.i.d. samples from $\tilde{\rho}$**. This corresponds to the standard computation of the predictive distribution in Bayesian inference.
> > >
> > > We now recognize that this connection was not clearly explained in the original submission. To clarify, we will add a brief explanation of the above reasoning toward the end of the paragraph beginning at line 311, where Algorithm 1 is introduced.
> > > We hope our explanation addresses your question. Please feel free to let us know if any part requires further clarification.
> > >
> > > Sincerely,
> > >
> > > --Authors

---

### Official Review · Reviewer_ZBrY · 2025-03-12

**Overall Recommendation:** 2

**Summary:**

The paper presents the PAC-Bayes based analysis of generalisation in recalibration of predictors. Recalibration of predictors to minimise calibration errors is a common task, however to the best of my knowledge, I haven't seen formal PAC-Bayes results on the generalisation aspect of it. Some arguments that I have seen are based on the usual concentration results. In that sense, the paper makes useful contribution. The paper also studies the bias in estimating top-label calibration with an estimator of the form of expected calibration error---a popular metric for measuring calibration. The paper also presents a recalibration based approach that is directly informed by the PAC-Bayes analysis. Experiments support the generation gap dependence on the PAC-Bayes terms, and further experiments on recalibration approaches suggest mixed insights.

**Claims And Evidence:**

1. The goals of the paper are clearly stated, and the paper also accomplishes it convincingly.
2. There are connections also mentioned to similar reported results in the literature (results from Tsybakov in Section 3.1, GP-based recalibration by Wenger et al. in Section 4.2).

3. One clarity that I can get is in Equation 8 and Equation 9: Equation 8 suggests to minimise a regularised form of Brier score, and it is known (check Chapter 3 here: https://www.cis.upenn.edu/~aaroth/uncertainty-notes.pdf#page=25.58) that minimising it should also fix the accuracy, or if we fix the calibration error, the predictor's performance improve. So is there some additional motivation for directly adding a loss function in Equation 9?

**Essential References Not Discussed:**

None.

**Experimental Designs Or Analyses:**

The paper suggests that temperature scaling can cause overfitting when recalibration data is small. However, I'm assume GP based inference can also suffer from that. Furthermore, when the dataset is large, then GP based inference can be computationally expensive. I'm aware there are methods to deal with these issues in the GP literature, but I'd appreciate if this can be highlighted.

**Methods And Evaluation Criteria:**

The methods and evaluation are justified to a certain extent. However, the paper needs to explore alternate approaches to estimating TCE / ECE, like the kernel based methods or the smooth calibration error. Experimentally, alternative recalibration methods like beta calibration, isotonic regression can also be investigated. This could help inform more exhaustive insights.

**Other Comments Or Suggestions:**

See above.

**Other Strengths And Weaknesses:**

The paper can also be improved in terms of writing. The paper currently presents theoretical results one after the other with little intuition or motivation over the significance of the results. The paper also assumes significant familiarity with the PAC-Bayes machinery, and could be made more accessible by a gentle introduction. For example, lines 191-192 state that $f_w$ and $S_{tr}$ are dependent, however for someone who is not familiar with the PAC-Bayes would not get the importance of such statements.

**Questions For Authors:**

Check above.

**Relation To Broader Scientific Literature:**

The paper's result informs the generalisation of recalibration approaches and the trade-offs therein. While the results could be useful, they don't seem very revealing in terms of what one should expect. The literature on ECE is familiar with the trade-off between the number of bins and estimation bias. PAC-Bayes generalisation bounds also are intuitive, and are expected in terms of what is known in the standard PAC-Bayes bounds. I'd appreciate if authors can further help me understanding the implications of the presented results.

**Theoretical Claims:**

I haven't verified the proofs of the presented theoretical claims in details, but they follow the similar machinery as the typical PAC-Bayes results, and in that sense, I do agree with the presented claims.

---

> ### Author Rebuttal · Authors · 2025-04-01
>
> Thank you very much for your valuable suggestions. All proposed changes have been incorporated into the main text. Due to the word limit, we cannot provide all revision details here. Please feel free to contact us during the discussion period if you'd like more information.
>
> ## Claim and Evidence
>
> ### Q.1: Motivation for Eq. (9)
> As you noted, the Brier score is a proper scoring rule and can improve accuracy (e.g., Kull et al., 2015; Perez-Lebel et al., 2018). However, our setting involves recalibration, where $f_w$ is trained with cross-entropy loss. Using a Brier-based recalibration may lead to inconsistent per-sample losses, particularly since squared loss can behave erratically for noise in the dataset.
> To mitigate this, we evaluated both Eq. (8) (Brier) and Eq. (9) (Brier + cross-entropy). Empirically, Eq. (9) tended to perform better.
>
> ## Methods and Evaluation Criteria
>
> ### Q.2: Motivation for using UWB and method comparison
> - (a) We focus on ECE via uniform-width binning (UWB), a widely used estimator in prior work. As noted in Section 7, alternatives like Kernel CE or Smoothed CE may improve stability, but our work is the first to theoretically analyze the instability of binning-based ECE in the multiclass case.
> - (b) Binary methods (e.g., Beta calibration, isotonic regression) can be extended via one-vs-all but calibrate only the top class. Hence, we compare only methods designed for multiclass outputs.  We revised Section 6.2 to clarify this, and Appendix E contains full comparisons.
>
> ## Experimental Designs or Analyses
>
> ### Q.3: GP-based methods and overfitting
> GP-based methods may face instability with limited recalibration data. However, their kernel-based smoothness prior acts as a natural regularizer based on the principle of Occam’s razor (e.g., Rasmussen & Williams, 2006; Bishop, 2006; MacKay, 1998), making them more robust to overfitting than parametric methods like temperature scaling. We added this to Section 6.2.
>
> ### Q.4: GP computational complexity
> We revised Section 4.2 to clarify:
> - We use the outputs of $f_w$ on $S_{\mathrm{re}}$ as $M$ inducing points for a data-dependent GP prior $\tilde{\pi}$, which maintains independence from $S_{\mathrm{tr}}$.
> - Following (Wenger et al. , 2020), we reduce the cost from $\mathcal{O}(N^3)$ to $\mathcal{O}(N^2M)$ (here, $N$ corresponds to the number of $S_{\mathrm{re}}$), which is feasible as the sample size of $S_{\mathrm{re}}$ is often smaller than that of $S_{\mathrm{tr}}$.
>
> ## Relation to Broader Scientific Literature
>
> ### Q.5: Novelty and context
> Prior work on ECE bias (e.g., Gupta & Ramdas, 2021; Sun et al., 2023; Futami & Fujisawa, 2024) mostly focuses on binary settings and does not address generalization (except for Futami & Fujisawa, 2024). Our work extends this to multiclass classification, with an optimizable PAC-Bayes bound and a novel recalibration algorithm.
> While some findings (e.g., optimal bin size) are consistent with binary results, this cross-setting agreement itself shows a novel knowledge obtained via our study.
> Multiclass calibration metrics have been discussed (e.g., Zhang et al., 2020; Gruber & Buettner, 2022), but without generalization or recalibration—key aspects of our study.
>
> We revised Section 5 accordingly:
> - Discussed how our PAC-Bayes analysis differs from conventional i.i.d.-based approaches.
> - Clarified that our bound is optimizable, enabling practical algorithm design.
> - Highlighted how our Theorem 3 formally explains the curse of dimensionality in $\mathrm{CE}\_K$, unlike prior work.
>
> ## Other Strengths and Weaknesses
>
> ### Q.6: Writing improvements
> Thank you for your comment. To enhance clarity, especially for readers less familiar with PAC-Bayes theory, we added proof sketches after Theorems 1 and 2 (see our response to [Reviewer FaKB Q.1](https://openreview.net/forum?id=eJzZryJfri&noteId=UqauLyR5uq)). We also removed a potentially confusing explanation around lines 191–192.
> Due to the word limit, we were unable to list all specific revision details here. If you are interested, we would be happy to provide them during the discussion period—please feel free to reach out.
>
> ### References
> - [Kull et al., 2015](https://link.springer.com/content/pdf/10.1007/978-3-319-23528-8_5.pdf)
> - [Perez-Lebel et al., 2018](https://arxiv.org/pdf/2210.16315)
> - [Rasmussen & Williams, 2006](https://gaussianprocess.org/gpml/chapters/RW.pdf)
> - [Bishop, 2006](https://link.springer.com/book/9780387310732)
> - [MacKay, 1998](https://core.ac.uk/download/pdf/216127203.pdf)
> - [Wenger et al. , 2020](https://proceedings.mlr.press/v108/wenger20a.html)
> - [Gupta & Ramdas, 2021](https://arxiv.org/abs/2105.04656)
> - [Sun et al., 2023](https://arxiv.org/abs/2305.10886)
> - [Futami & Fujisawa, 2024](https://proceedings.neurips.cc/paper_files/paper/2024/file/9961e42624a6c083279303767c73269d-Paper-Conference.pdf)
> - [Zhang et al., 2020](https://arxiv.org/pdf/2003.07329)
> - [Gruber & Buettner, 2022](https://arxiv.org/abs/2203.07835)

---

> > ### Comment · Reviewer_ZBrY · 2025-04-04
> >
> > thanks for the clarification. I'd appreciate if the authors can further clarify the answer to question Q1. Is there an empirical demonstration to this inconsistent per-sample losses? My comment is more on the lines that if we actively eliminate the calibration error, it should also improve the overall loss (due to the decomposition of risk into calibration and sharpness). Obviously, there are practical considerations to this statement, but I'm just curious about recalibration approaches using two losses (Brier + cross-entropy, both of which have properness).

---

> > > ### Author Response · Authors · 2025-04-05
> > >
> > > Dear Reviewer ZBrY,
> > >
> > > Thank you very much for your thoughtful comments. We apologize for having provided a response that did not fully address your intent regarding Q1.
> > >
> > > Based on your feedback, we re-ran the experiments under the same settings and re-evaluated our methods by measuring not only ECE and classification accuracy, but also cross-entropy. The results are presented below. Here, bold font indicates the best result among PBR and PBR total for each metric. If the best values are numerically identical, no bolding is applied. In case of a tie in the mean, the result with a smaller standard deviation is considered better. (Please note that due to re-running the experiments, some numerical values may slightly differ from those reported in the original submission.)
> > > - https://drive.google.com/file/d/1Dlvl9a1Gu76TG0gAuGYLJSaKsak7VSSF/view?usp=sharing
> > > - https://drive.google.com/file/d/1eNGLvt8oMnd3EA-TZ3mGlRh6hMx0uK2F/view?usp=sharing
> > >
> > > From these updated results, the first clear observation is that our two methods—PBR and PBR total—consistently improve ECE by minimizing the Brier score. This is an expected outcome, considering that the Brier score is upper-bound of the ECE. Regarding cross-entropy and accuracy, we also observe improvements in some settings, particularly in binary classification tasks on relatively simple datasets such as KITTI and PCam.
> > >
> > > On the other hand, when minimizing the Brier score alone (still including KL regularization), PBR can lead to degradation in both cross-entropy and classification accuracy, especially on complex datasets and in multi-class classification tasks. This tendency is evident in Table 9 (excluding the XGBoost/Random Forest settings) and Table 10 (excluding the AlexNet experiment), and is particularly pronounced in experiments involving deep neural network models (please see the second link we provided).
> > >
> > > One possible reason for this behavior is as follows:
> > > In all of our experiments, the deep neural network models $f_w$ are originally **trained using the cross-entropy loss**. When recalibration is then performed only using a Brier score-based objective, the model is **recalibrated according to a loss function that differs from the one used during training**. Such mismatches in the optimization objectives can propagate through the GP-based recalibration process and result in serious inconsistencies, which may manifest as a decline in accuracy and an increase in cross-entropy.
> > >
> > > The results also show that this degradation can be alleviated by incorporating cross-entropy into the recalibration objective, as done in PBR total. This suggests that the inclusion of cross-entropy helps mitigate the effect of the mismatch caused by using a recalibration loss function that is misaligned with the original training objective.
> > >
> > > In summary, minimizing the Brier score alone does not necessarily improve cross-entropy or accuracy and may, in some cases, worsen them. However, by including cross-entropy as part of the recalibration objective, we can achieve a better balance between calibration and predictive performance, improving ECE while avoiding deterioration in accuracy.
> > >
> > > Thanks to your insightful comment, we were able to conduct a deeper analysis of the behavior of our two proposed methods. We have included the discussion and the new experimental results in Appendix E. We hope this response adequately addresses your concerns.
> > >
> > > Sincerely,
> > >
> > > --Authors

---

### Official Review · Reviewer_FaKB · 2025-03-13

**Overall Recommendation:** 3

**Summary:**

This paper provides a PAC-Bayesian analysis of recalibration in multiclass classification, particularly focusing on evaluating and controlling the bias and generalization error of the expected calibration error (ECE) viewed as an estimator of the top-label calibration error (TCE; infeasible to evaluate); see Section 3. The authors introduce a general theoretical framework using PAC-Bayes analysis to derive optimizable upper bounds for the generalization error and bias of the ECE, which are subsequently used to propose a novel recalibration algorithm (PAC-Bayes Recalibration, or PBR; see Section 4.2). Theoretical results include optimal bin-size choices and associated convergence rates. Empirical evaluations demonstrate the effectiveness of the proposed method, showing consistent performance improvements in recalibration compared to other existing methods (e.g., Gaussian process recalibration, temperature scaling).

**Claims And Evidence:**

The main claims that PAC-Bayes bounds can effectively quantify generalization and estimation biases in multiclass recalibration scenarios are generally well-supported by solid theoretical analyses (Theorems 1, 2, and 3) and comprehensive numerical experiments (Section 6).

**Essential References Not Discussed:**

The discussion of related works seems generally adequate.

**Experimental Designs Or Analyses:**

The experimental setups and analyses appear generally sound, utilizing standard benchmarks such as MNIST and CIFAR-100 datasets, and appropriate baseline methods (temperature scaling, GP-based recalibration). However, clearer connections between empirical experiments and the theoretical results (e.g., verifying optimal bin-size choices empirically) could help readers better appreciate the results in this paper as well as further enhance the paper’s contribution.

**Methods And Evaluation Criteria:**

The proposed methods and evaluation criteria—particularly the use of standard benchmarks (MNIST, CIFAR-100) and established baselines (temperature scaling, GP recalibration)—are relevant and appropriate for this problem.The choice of calibration metrics (TCE, ECE) aligns well with existing literature.

**Other Comments Or Suggestions:**

This paper offers noteworthy contributions, yet it could be significantly strengthened by improving clarity and organization. In particular, introducing key concepts and theoretical motivations earlier—and in a more structured manner—would help preempt many reader questions and confusions. Below, I elaborate on specific suggestions:

**1. Manuscript organization and clarification:**
It would be beneficial to address anticipated reader questions more directly and to summarize theoretical insights earlier. For example, in the introduction (lines 76–86), the authors raise the concern that some bins may remain empty when applying PAC-Bayes analysis to ECE. However, they do not specify that either they will use equal-width binning or this challenge applies primarily to equal-width binning---prompting the reader to wonder about "what if we use uniform-mass binning instead?" Another instance arises from the presentation of TCE as the central recalibration metric in early sections (e.g., Section 1 and Section 2.2), along with the implication that ECE is crucial for analyzing TCE. This leads to questions about why TCE is the only focus and whether analyzing ECE is indeed the best way to manage TCE. Although these issues are addressed in later sections, introducing the rationale and framing them clearly from the outset would greatly enhance readability.

**2. Explicit definitions and explanations:**
Clearly defining core ideas---such as total bias (pre- and post-calibration) and the generalization error of ECE---would help readers follow the arguments more easily. Placing these definitions in a dedicated “Definition” environment could make them more prominent and accessible. Additionally, when comparing Equations (8) and (9), it should be highlighted more explicitly that the only change is the added classification loss term $l_{\mathrm{acc}}$, instead of merely presenting two expressions. A more explicit mention of this difference, possibly alongside visual cues or parentheses to clarify the scope of the expectation in Equation (9), would be helpful.

**3. Positioning of the “Related Work” section:**
It might be advantageous to move Section 5 (“Related Work”) closer to the introduction. While I understand that the section also serves to discuss the results presented later, having this material earlier could frame the research questions and contributions more clearly. Doing so would likely address some of the aforementioned reader concerns about TCE, ECE, and binning choices right from the start.


I have some additional minor comments below:

**4.** In Section 6, the authors write "... we observe a correlation between the KL divergence and the ECE generalization gap, as confirmed by Pearson’s and Kendall’s rank-correlation coefficients, supporting the validity of our bound."  However, I don't clearly see (1) if the correlation is sufficiently significant, and (2) how the observed trend supports theoretical findings.  It may be worth adding a few more sentences for a more detailed and systematic assessment.

**5.** In Section 2.3, the authors define the recalibration map as a *parametric* function $\eta_V: \Delta^K \to \Delta^K$ with parameter $V \in \mathbb{R}^{d'}$. Meanwhile, the abstract and other parts of the paper emphasize nonparametric binning. Either revising the definition of the recalibration map or clarifying how a parametric approach to recalibration aligns with the stated focus on nonparametric binning would help avoid confusion and strengthen the paper’s coherence.

**Other Strengths And Weaknesses:**

Strengths:
* Solid theoretical contributions that provide novel guarantees for ECE using PAC-Bayes bounds, enhancing methodological rigor.

* Broad relevance, particularly with practical applications in multiclass recalibration scenarios.

* Extensive empirical validation that convincingly demonstrates the effectiveness of the proposed recalibration method.

Weaknesses:
* Presentation can be significantly improved by clarifying key ideas earlier in the text and summarizing the theoretical contributions more explicitly and concisely, enhancing readability and accessibility.

**Questions For Authors:**

Could you clarify or comment on the practical tightness of the upper bounds presented in Theorems 1, 2, and 3? While I appreciate the arguments regarding the asymptotic minimax rate with respect to sample size, I am curious about two additional aspects:

**1. Dependence on other factors.** How tight is the dependence on parameters such as $K, B, \lambda$, and the KL divergence? It would be informative to see either a theoretical discussion or numerical experiments (e.g., with synthetic toy datasets) that illustrate how these factors affect the bounds.

**2. Potentially favorable properties in practical datasets.** Are there particular properties in real-world datasets (or their underlying distributions) that might lead to faster rates, even if the general minimax rates are pessimistic? Demonstrating such scenarios would further highlight the practical significance and relevance of your results.

**Relation To Broader Scientific Literature:**

The authors effectively position their contributions within existing literature on calibration methods, PAC-Bayes theory, and binning-based calibration error estimation. Their work clearly distinguishes itself from recent recalibration literature by providing a PAC-Bayes theoretical framework for recalibration, particularly emphasizing the multiclass setting compared to previous binary-focused works.

**Theoretical Claims:**

I briefly checked the main theoretical results (Theorems 1, 2, and 3) and their proofs provided in the appendix. They appear technically sound and are based on standard PAC-Bayes and concentration inequality arguments. However, the main text could greatly benefit from including clearly summarized high-level intuitions and sketches of key proofs (particularly for Theorem 1). I believe presenting the theoretical insights more explicitly and earlier could improve readability and comprehension.

---

> ### Author Rebuttal · Authors · 2025-04-01
>
> Thank you very much for your valuable suggestions. All proposed changes have been incorporated into the main text. Due to the word limit, we cannot provide all revision details here. Please feel free to contact us during the discussion period if you'd like more information.
>
> ## Theoretical Claims and Weaknesses
>
> ### Q.1: Proof Sketch of Theorems 1 and 2
> We added a proof sketch to clarify our theoretical contributions. The bias is decomposed into binning approximation error and finite-sample estimation bias:
> - We define $f_{\mathcal{I}}(x) = \sum_{i=1}^B \mathbb{E}[f_w(X) \mid f_w(X) \in I_i] \cdot 1_{f_w(x) \in I_i}$, representing the expected label frequency in each bin.
> - The bias is bounded as:  $\mathrm{Bias}(f_w, S_{\text{te}}, \mathrm{CE}) \leq |\mathrm{CE}(f_w) - \mathrm{CE}(f_{\mathcal{I}})| + |\mathrm{CE}(f_{\mathcal{I}}) - \mathrm{ECE}(f_w, S_{\text{te}})|$
> - The first term uses the Lipschitz assumption; the second is bounded via McDiarmid’s inequality with binwise conditioning.
> - In Theorem 2, only the second term remains and is bounded via PAC-Bayes using the Donsker–Varadhan inequality.
>
> These techniques generalize the binary setting [Futami & Fujisawa, 2024](https://proceedings.neurips.cc/paper_files/paper/2024/file/9961e42624a6c083279303767c73269d-Paper-Conference.pdf) to the multiclass case and form a core novelty of our work.
>
> ### Q.2: Clarifying Contributions in the Introduction
> We revised the sixth paragraph of the Introduction to concisely state our contributions, including the decomposition and concentration inequality enabling analysis of binning-based ECE with UWB.
>
> ## Experimental Design or Analyses
>
> ### Q.3: Empirical Validation of Theoretical Results
> We extended the bound verification experiment from [Zhang et al., 2020](https://arxiv.org/pdf/2003.07329) and (Futami & Fujisawa, 2024) to the multiclass case. Using synthetic data with analytically computable TCE, we confirmed:
> - The TCE–ECE gap decreases as $\mathcal{O}(1/n^{1/3})$ at the theoretical bin size.
> - The empirically optimal bin size also follows $\mathcal{O}(n^{1/3})$.
>
> Results are shown [here](https://drive.google.com/file/d/1YBM1El-FtxYh1sRg0OjA0tw7CItwOnuQ/view?usp=sharing).
> This discrepancy arises because the theoretical bin size minimizes the **upper bound**, not necessarily the actual TCE gap. We added this discussion to Section 6.
>
> ## Other Comments and Suggestions
>
> ### S.1-1: Clarifying Use of UWB
> We clarified early in the Introduction that our analysis focuses on ECE estimated using UWB. In Section 7, we also note that extending the theory to estimators like uniform-mass binning is an important future direction.
>
> ### S.1-2: Justifying TCE and ECE Focus
> We chose TCE and ECE because (1) TCE is theoretically well-founded, (2) ECE is its standard estimator in practice, and (3) their bias and generalization in multiclass settings remain underexplored.
>
> ### S.2: Clarifying Eqs. (8) and (9)
> We added definitions for key terms (e.g., total bias, generalization error, top-$K$ CE) and clarified the distinction between Eqs. (8) and (9).
>
> ### S.3: Related Work Positioning
> To improve clarity, we placed Related Work after introducing the main definitions. A pointer to Section 5 was added at the end of the Introduction.
>
> ### S.4: Correlation Analysis
> Following prior work (e.g., [Jiang et al., 2019](https://arxiv.org/abs/1912.02178); [Kawaguchi et al., 2023](https://arxiv.org/pdf/2305.18887)), we confirmed a positive correlation between the KL term in Eq. (6) and the ECE gap through PBR-based experiments. This supports the validity of the bound. We made this clearer in Section 6.1.
>
> ### S.5: Parametric vs. Nonparametric Clarification
> We revised the abstract and Section 1 to clarify the use of nonparametric methods (e.g., binning for ECE estimation) and parametric ones (e.g., recalibration via GP). We also clearly stated the two main objectives.
>
> ## Questions
>
> ### Q.4: Class Size and Prior Choices
> Theorems 1 and 2 focus on Top-1 accuracy and are unaffected by class size $K$, but Theorem 3 is. Our bound decreases slower than $\mathcal{O}(\sqrt{K/n})$ ([Morvant et al., 2013](https://arxiv.org/pdf/1202.6228)), which is an important direction for future work.
> Our bin size is minimax-optimal and aligns with asymptotic trends. $\lambda$ and the prior are tunable; we used a GP prior, but tighter (non-optimizable) bounds from marginal or IT-based priors (e.g., Futami & Fujisawa, 2024) are also possible.
>
> ### Q.5: Fast Rates under Low-Noise Conditions
> Under Bernstein or Tsybakov noise ([Alquier et al., 2016](https://jmlr.org/papers/v17/15-290.html)), PAC-Bayes can achieve fast rates. Our setting differs structurally, but exploring whether fast-rate bounds can be incorporated into ECE analysis is an interesting future direction.

---

> > ### Comment · Reviewer_FaKB · 2025-04-04
> >
> > I thank the reviewers for their response, and maintain my positive evaluation rating.

---

> > > ### Author Response · Authors · 2025-04-08
> > >
> > > Dear Reviewer FaKB,
> > >
> > > Thank you very much for taking the time to carefully read our rebuttal, especially during this busy period.
> > > We truly appreciate your thoughtful feedback, which has helped us improve the quality of our paper.
> > >
> > > As the discussion period is coming to a close, we would be grateful if you could briefly comment on why the score remained unchanged in light of our rebuttal.
> > > If there are any remaining issues or aspects that were insufficiently addressed, we would be more than happy to make further revisions.
> > >
> > > Thank you again for your time and consideration.
> > >
> > > Sincerely,
> > >
> > > --Authors

---

### Official Review · Reviewer_mK1Q · 2025-03-25

**Overall Recommendation:** 4

**Summary:**

Existing recalibration methods either lack theoretical analysis or are limited to binary classification. To address this, the authors first analyze the generalization error and estimation bias of the ECE in multiclass classification, deriving non-asymptotic bounds and identifying the practical optimal bin size. They then conduct a PAC-Bayes analysis for recalibration and propose a new generalization-aware recalibration algorithm based on the PAC-Bayes bound. Numerical experiments demonstrate that the proposed algorithm outperforms the performance of Gaussian process-based recalibration across various benchmark datasets and models

**Claims And Evidence:**

Yes.

**Essential References Not Discussed:**

No.

**Experimental Designs Or Analyses:**

Yes, the experiments seem convincing and discussion is detailed

**Methods And Evaluation Criteria:**

Yes. The definition of generalization error is standard, and the datasets are appropriate.

**Other Comments Or Suggestions:**

See weaknesses.

**Other Strengths And Weaknesses:**

Strengths:
1. The writing is clear, with well-stated theorems and a strong takeaway message.
2. The theoretical results appear robust, tight, and technically innovative.
3. Practical algorithms are derived from the theoretical analysis, demonstrating applicability.

Weaknesses:
1. Some results may not be particularly surprising, given the results of Futami and Fujisawa (2024).
2. The assumption of Lipschitz continuity may be too strong, as the argmax function itself is not Lipschitz. A small perturbation in x could lead to significant changes in the conditional probability.

**Questions For Authors:**

1. Could you elaborate on the technical innovations of this paper compared to Futami and Fujisawa (2024)?
2. The assumption of Lipschitz continuity appears somewhat strong, given that the argmax function itself is not Lipschitz. A small perturbation in x can potentially cause a large change in the conditional probability. Could you clarify why this assumption is still reasonable in this context?

**Relation To Broader Scientific Literature:**

Conceptually, this paper primarily extends the results of Futami and Fujisawa (2024) from binary classification to the multiclass setting. However, it employs different mathematical techniques, which contribute to its originality and innovation.

**Theoretical Claims:**

I didn't verify the proofs in the appendix in detail but reviewed the proof outline, which appears to be correct, but I could be wrong because I am not very familiar with the PAC-Beyas theory.

---

> ### Author Rebuttal · Authors · 2025-03-31
>
> We sincerely thank you for your feedback.
>
> ### Q.1: Regarding the key technical innovations of this paper relative to Futami and Fujisawa (2024)
>
> The key difference from Futami and Fujisawa (2024) lies in extending the analysis from binary to multi-class classification. In the binary setting, the change in ECE when replacing a single sample can be easily bounded, enabling a straightforward application of McDiarmid’s inequality via bounded differences. In contrast, our multi-class setting requires bounding the difference over simplex-structured binning, which is significantly more complex. Establishing a proof technique to handle this is a core technical contribution of our work.
> The effectiveness of extending the proof techniques to the multiclass setting is most clearly shown in the analysis of the Top-$K$ calibration metric in Theorem 3 and the corresponding optimal bin size. Our analysis suggests that the derived optimal bin size is closely related to the nonparametric estimation of conditional probabilities in $K$-dimensional spaces. This insight generalizes the findings of Futami and Fujisawa (2024), which focused on the binary classification case, essentially a one-dimensional problem involving the predicted probability of a single label.
> Moreover, while their information-theoretic analysis yields non-vacuous bounds via mutual information, it does not lead to optimizable objectives, making it unsuitable for deriving generalization-aware recalibration algorithms. In contrast, our PAC-Bayes-based analysis provides a KL-divergence upper bound that is optimizable, naturally leading to a variational Bayes-style recalibration method.
>
>
> ### Q.2: Regarding the justification of the Lipschitz continuity assumption
>
> We guess this question concerns Assumption 1, so we first clarify its meaning. Assumption 1 states that, for a neural network $f_w(x)$ with a final layer mapping into the probability simplex (e.g., via softmax), the conditional probability $\mathbb{E}[Y \mid f(x)]$ of the top predicted class is Lipschitz continuous with respect to the input $x$, under fixed parameters $w$. Notably, the Lipschitz condition is imposed only on the input space, not on the computation of the top class index $C = \arg\max_{k} f_{w}(x)_k$. That is, the assumption only requires that there exists a constant $L$ such that $\left|\mathbb{E}[Y \mid f(x)] - \mathbb{E}[Y \mid f(x')]\right| \leq L \|x - x'\|$, where the output components are fixed. Thus, the argmax operation does not influence this assumption.
> Moreover, this Lipschitz continuity is a mild and standard assumption in nonparametric estimation, including binning and kernel-based ECE estimation. Since ECE estimates a conditional expectation, some form of smoothness—such as Lipschitz or Hölder continuity—is required. Without it, even a small change in input could cause drastic label changes, making estimation from finite samples infeasible. It has been shown that such smoothness is necessary for consistency in conditional expectation estimation (see Li et al., (2021)). This assumption is directly related to estimation bias: without it, ECE does not converge to TCE, even as the number of samples increases.
>
> ### References
> - [Futami & Fujisawa, 2024](https://proceedings.neurips.cc/paper_files/paper/2024/file/9961e42624a6c083279303767c73269d-Paper-Conference.pdf)
> - [Li et al., 2021](https://arxiv.org/pdf/2103.07095)

---

> > ### Comment · Reviewer_mK1Q · 2025-04-08
> >
> > I would like to thank the reviewers for addressing my questions. I will maintain my score. Regarding the response to Q2, I understand that the Lipschitz assumption is not define on top of the argmax. However, in that case, should the notation be revised? More generally, I believe the notation throughout the paper could be revised to improve readability.

---

> > > ### Author Response · Authors · 2025-04-09
> > >
> > > Thank you very much for your valuable comment, which will greatly help improve the readability of our paper.
> > >
> > > To clarify the meaning, we will add the following note under Assumption 2:
> > > - “We consider the Lipschitz continuity of $f_w(X)_k$ with respect to $X$, **given a fixed label** $C = \arg\max_k f_w(X)_k$. We note that the argmax operation used to compute $C$ is not included in the Lipschitz condition.”

---

### Decision · Program_Chairs · 2025-05-01

**Decision:**

Accept (poster)

**Comment:**

The reviewers agreed that the paper offers one of the rare theoretically grounded contributions to classifier recalibration. This is also an original contribution to the PAC-Bayes framework, as the studied metric differs from the ones classically studied when bounding the generalization error. The PAC-Bayesian treatment allows the conception of an efficient recalibration algorithm based on bound optimization.

Importantly, the reviewers diligently suggested clarification improvements during the discussion. I strongly encourage the authors to integrate them into the revised version of their paper conscientiously.